



# Surface composition of debris-covered glaciers across the Himalaya using linear spectral unmixing of Landsat 8 OLI imagery

**Adina E. Racoviteanu[1], Lindsey Nicholson[2], and Neil F. Glasser[1]**

[1]Department of Geography and Earth Sciences, Aberystwyth University, TS1 , UK
[2]Department of Atmospheric and Cryospheric Sciences, University of Innsbruck, TS2 , Austria

**Correspondence:** Adina E. Racoviteanu (adr18@aber.ac.uk, racovite@gmail.com)

**Abstract.** CE1 CE2 TS3 The Himalaya mountain range is characterized by highly glacierized, complex, dynamic topography. The ablation area of glaciers often features a highly heterogeneous debris mantle comprising ponds, steep and shallow slopes of various aspects, variable debris thickness, and exposed ice cliffs associated with differing ice ablation rates. Understanding the composition of the glacier surface is essential for a proper understanding of glacier hydrology and glacier-related hazards. Until recently, efforts to map debris-covered glaciers from remote sensing focused primarily on glacier extent rather than surface characteristics and relied on traditional whole-pixel image classification techniques. Spectral unmixing routines, rarely used for debris-covered glaciers, allow decomposition of a pixel into constituting materials, providing a more realistic representation of glacier surfaces. Here we use linear spectral unmixing of Landsat 8 Operational Land Imager (OLI) images (30 m) to obtain fractional abundance maps of the various supraglacial surfaces (debris material, clean ice, supraglacial ponds, vegetation) across the Himalaya around the year 2015. We focus on the debris-covered glacier extents as defined in the supraglacial debris cover database. The spectrally unmixed surfaces are subsequently classified to obtain maps of composition of debris-covered glaciers across sample regions.

We test the unmixing approach in the Khumbu region of the central Himalaya, and we evaluate its performance for supraglacial pond by comparison with independently mapped ponds from high-resolution Pléiades (2 m) and PlanetScope imagery (3 m) for sample glaciers in two other regions with differing topo-climatic conditions. Spectral unmixing applied over the entire Himalaya mountain range (a supraglacial debris cover area of 2254 km$^2$) indicates that at the end of the ablation season, debris-covered glacier zones comprised 60.9 % light debris, 23.8 % dark debris, 5.6 % clean ice, 4.5 % supraglacial vegetation, 2.1 % supraglacial ponds, and small amounts of cloud cover (2 %), with 1.2 % unclassified areas. Supraglacial ponds were more prevalent in the monsoon-influenced central-eastern Himalaya (up to 4 % of the debris-covered area) compared to the monsoon-dry transition zone (only 0.3 %) and in regions with lower glacier elevations. Climatic controls (higher average temperatures and more abundant precipitation), coupled with higher glacier thinning rates and lower average glacier velocities, further favour pond incidence and the development of supraglacial vegetation. The spectral unmixing performed satisfactorily for the supraglacial pond and vegetation classes (an F score of ∼ 0.9 for both classes) and reasonably for the debris classes (F score of 0.7). With continued advances in satellite data and further method refinements, the approach presented here provides avenues towards achieving large-scale, repeated mapping of supraglacial features.

## 1 Introduction

High relief orogenic belts such as the Himalaya are characterized by glacierized, complex, dynamic topography and the presence of a continuous cover of rock debris across the lowest part of the ablation zone of glaciers (Kirkbride, 2011). Globally, supraglacial debris cover accounts for ∼ 7 % of the total glacierized area (Scherler et al., 2018; Herreid and Pellicciotti, 2020). In high-mountain environments, high denudation rates and mass-wasting processes such as rockfalls and rockslides from the steep valley sides supply abundant

rock debris to the glacier surface (Kirkbride, 2011; Shroder et al., 2000; Evatt et al., 2015). This results in highly heterogeneous surfaces, consisting of debris material of various lithologies and grain sizes (sand and silt to boulders), forming debris cones on variable but mostly shallow slopes. Some of the most notable features of such surfaces are the supraglacial ponds and exposed ice cliffs, which have gained interest in recent years for several reasons. First, they influence the surface energy receipts of the supraglacial debris surface and the efficiency with which atmospheric energy can be transferred to the underlying ice and cause glacier ice ablation. While ice ablation beneath debris cover of more than a few centimetres thick is strongly reduced (Østrem, 1959; Nicholson and Benn, 2006; Reid and Brock, 2010), ice cliffs and supraglacial ponds are local hot spots for glacier downwasting due to enhanced energy absorption at the surface of these features (Ragettli et al., 2016; Miles et al., 2016; Sakai et al., 2002; Buri et al., 2016; Steiner et al., 2015). Understanding their spatial distribution is essential for a proper assessment of glacier hydrology, notably to simulate glacier-wide ablation rates and meltwater production. Second, the current distribution and fluctuation of proglacial lakes and supraglacial pond extents is of interest for assessing glacier-related hazards. Recent studies have reported an increase in pro- and supraglacial lake area and number in the Himalaya and worldwide as a response to climatic changes (Shugar et al., 2020; Nie et al., 2017; Shukla et al., 2018). Some of the supraglacial ponds coalesce and form larger supraglacial lakes, which may evolve into fully formed proglacial ice or moraine-dammed lakes (Benn et al., 2012; Thompson et al., 2012), with enhanced potential for producing hazards such as glacier lake outburst floods (Benn et al., 2012; Komori, 2008; Richardson and Reynolds, 2000; Reynolds, 2014; GAPHAZ, 2017). Increasing trends of pond development of 17 % to 52 % per year were reported in the Khumbu region (2000 to 2015) (Watson et al., 2016), with a 3-fold increase in pond area over three decades (1989 to 2018) (Chand and Watanabe, 2019). Quantifying the number/area of supraglacial ponds and their evolution (Miles et al., 2017 TS4; Liu et al., 2015; Watson et al., 2016) is important for assessing which ones might represent conditioning factors for hazards (Sakai and Fujita, 2010; Reynolds, 2000). Third, understanding the fluctuations of these surface characteristics, in particular supraglacial vegetation, is important since vegetation expansion on debris-covered surfaces may indicate the transition from a debris-covered glacier to a rock glacier in a context of climate change (Shroder et al., 2000; Jones et al., 2019 TS5; Knight et al., 2019; Monnier and Kinnard, 2017; Kirkbride, 1989).

Our understanding of the regional variability in glacier mass balance of both clean and debris-covered glaciers in the Himalaya has improved over the last years (Dehecq et al., 2019; Brun et al., 2017; Shean et al., 2020), and the role of glacier morphology in controlling glacier behaviour and changes has been demonstrated in recent studies (Salerno et al., 2017; Brun et al., 2019). However, a comprehensive assessment of the surface geomorphology, supraglacial pond coverage, moraine characteristics and supraglacial vegetation at various temporal scales is still needed over the entire Himalaya. Until recently, efforts to map debris-covered glaciers focused primarily on their extent rather than the surface characteristics. This was achieved at regional scales using a combination of digital elevation models (DEMs), various spectral band ratios and terrain curvature (Shukla et al., 2010; Bolch et al., 2007; Kamp et al., 2011; Bishop et al., 2001; Paul et al., 2004). Attempts to improve the accuracy of debris-covered glacier mapping included the use of thermal data, i.e. temperature differences between debris underlined by glacier ice and the surrounding non-ice moraines (Taschner and Ranzi, 2002; Bhambri et al., 2011a; Racoviteanu and Williams, 2012; Alifu et al., 2016) or the use of glacier velocity (Smith et al., 2015 TS6). Considerable improvements in monitoring capacity due to recent satellite developments and cloud-computing platforms such as Google Earth Engine allowed exploitation of large amounts of Landsat and Sentinel-2 data. This has resulted in two recent global datasets of supraglacial debris (Scherler et al., 2018; Herreid and Pellicciotti, 2020). While these global datasets represent an important development in advancing the understanding of the distribution of debris-covered glaciers at a large scale, they can suffer from the use of inconsistent methods and different temporal coverage between and/or within regions. Supraglacial debris in these databases was mapped within the bounds of the Randolph Glacier Inventory (RGI) (Pfeffer et al., 2014), which has varying analysis dates and accuracy. While these issues were partially mitigated in a revised dataset based on semi-automated assessments of Landsat imagery (Herreid and Pellicciotti, 2020), improvements were limited to glaciers larger than 1 km$^2$ and were not applied repeatedly at the global scale.

Supraglacial ponds and ice cliffs are currently not represented either in existing supraglacial debris cover datasets or in the updated, publicly available regional glacier lake inventories (Wang et al., 2020; Shugar et al., 2020; Chen et al., 2021). The latter tend to focus primarily on the representation of proglacial lakes and their decadal changes. A database of supraglacial ponds at several time periods is desirable in order to complement the existing supraglacial debris and lake databases, as the distribution of these surface features on debris-covered glacier tongues remains limited to a handful of glaciers in the Himalaya (Watson et al., 2016, 2017a, 2018; Steiner et al., 2019). For example, regional studies on seasonal dynamics and evolution of supraglacial ponds and ice cliffs tend to be biased towards the well-studied Khumbu and Langtang areas of Nepal Himalaya (Watson et al., 2016, 2017a; Miles et al., 2017; Steiner et al., 2019). More studies are needed in other regions in order to assess the spatial differences in their occurrence as well as to infer the long-term changes of these features.

The increased availability of high-resolution (0.5 to 5 m) remotely sensed data from Pléiades, SPOT and Quick-Bird satellites, complemented by RapidEye, PlanetScope and SkySat images from Planet, has offered new opportunities for characterizing the surface of debris-covered glaciers in more detail. Supraglacial ponds and ice cliffs have been mapped using a combination of manual digitization on high-resolution multi-spectral imagery (1–3 m) or directly on Google Earth (Brun et al., 2018; Watson et al., 2018, 2017a, 2016; Steiner et al., 2019). Semi-automated mapping methods include adaptive binary thresholding (Anderson et al., 2021), band ratios and/or morphological operators (Miles et al., 2017; Liu et al., 2015), the normalized difference water index (NDWI) (Watson et al., 2018; Gardelle et al., 2011; Miles et al., 2017; Kneib et al., 2020; Liu et al., 2015; Wessels et al., 2002; Narama et al., 2017), feature extraction via decision trees and/or object-based image analysis (OBIA) (Liu et al., 2015; Kraaijenbrink et al., 2016; Panday et al., 2011), or thermal imagery (Suzuki et al., 2007; Foster et al., 2012). Other methods include the use of very-high-resolution topographic models generated using terrestrial structure-from-motion techniques (Westoby et al., 2014; Rounce et al., 2015; Herreid and Pellicciotti, 2018; Westoby et al., 2020) or the use of unmanned aerial vehicle (UAV) data (Kraaijenbrink et al., 2016). Synthetic aperture radar overcomes the limitations of optical remote sensing in areas with frequent cloud cover (i.e. the eastern Himalaya) and has been used to map supraglacial ponds and track their dynamics (e.g. Strozzi et al., 2012; Wangchuk and Bolch, 2020; Zhang et al., 2021). Despite methodological developments, a robust and transferable method for mapping ice cliffs and ponds in a systematic manner using these high-resolution datasets does not yet exist, and current methods remain computationally intensive. Understanding how the surface composition of the debris-covered tongues upscales in coarser-resolution imagery such as Landsat is still needed at regional scales. For example, large differences were shown between UAV-derived ponds and RapidEye-derived ponds in other studies (CE3 cf. Kraaijenbrink et al., 2016).

Even with the increased availability of high-resolution imagery, medium resolution data from archive Landsat series (30 m spatial resolution) remain a valuable data source for various regional-scale mapping applications due to their large swath width (185 km), free accessibility and acquisition time spanning four decades. One of the limitations in using these medium-resolution data is that most studies rely on traditional whole-pixel image classification techniques. While these classification techniques are advantageous for some applications, they does not reveal the constituent surfaces of image pixels on the ground or their proportions (Keshava and Mustard, 2002). Spectral unmixing routines, initially described by Atkinson (1997, 2004)TS7 and Foody (2004)TS8, allow decomposition of a given pixel into constituting materials, providing their fractional abundance and thus generating a more realistic representation of complex surfaces (Ke-

shava and Mustard, 2002). These have been used in glaciology to retrieve snow grain size and derive fractional snow-covered areas from MODIS or Landsat (Painter, 2003 TS9; Painter et al., 2009; Sirguey et al., 2009; Veganzones et al., 2014; Rosenthal and Dozier, 1996) and to map clean glacier areas or snow (Painter et al., 2012; Cortés et al., 2014), lakes (Zhang et al., 2004), and vegetation (Ettritch et al., 2018; Song, 2005; Xie et al., 2008). A small number of studies used spectral unmixing to characterize the mineral composition of debris-covered glaciers (Casey and Kääb, 2012; Casey et al., 2012); to characterize lake colour, turbidity and suspended sediments (Matta et al., 2017; Giardino et al., 2010); and more recently to map ice cliffs (Kneib et al., 2020), but the potential of sub-pixel mapping for debris-covered glaciers has not been fully exploited.

In this study, we use spectral unmixing of Landsat 8 Operational Land Imager (OLI) imagery to detect the surface characteristics of supraglacial debris cover across the Himalaya, with a particular emphasis on quantifying the supraglacial pond coverage and vegetation. We first apply and validate the spectral unmixing in the well-studied Khumbu region of the central Himalaya. Using the spectra and spectral unmixing parameters that were derived from the Khumbu, we infer the composition of supraglacial debris cover for the entire Himalaya spatial domain. We validate the pond results by comparing the supraglacial pond areas derived from spectral unmixing with those obtained using OBIA on high-resolution imagery for selected glaciers at three different sites. We use the results to assess the composition of the debris-covered glacier tongues in regions with differing topo-climatic conditions to evaluate the distribution of supraglacial ponds and vegetation across the mountain range in relation to geographic location, climate, topographic characteristics, glacier mass balance and surface velocity, and we discuss the potential relationship between these features and the temporal evolution of these glaciers.

## 2 Data sources and methods

### 2.1 Study area

Our study area comprises various spatial domains (Fig. 1). The larger Himalaya domain is defined here as the region spanning ∼ 1500 km (∼ 76 to 92° longitude and ∼ 26 to 34° latitude), covering areas from the Himachal/Jammu and Kashmir border in the west to the Bhutan Himalaya in the east (Fig. 1). Glaciers in this area have been in a state of negative mass balance in the last decades, with accelerating trends in the 2000 to 2010 decade (Bolch et al., 2019; Brun et al., 2017; Kääb et al., 2012; Maurer et al., 2019). We developed our method in the glacierized Khumbu region of Nepal, which we refer to hereafter as the "Khumbu domain", although it also includes glaciers north of the divide (Fig. 2). Glaciers in the Khumbu have been well studied in terms of

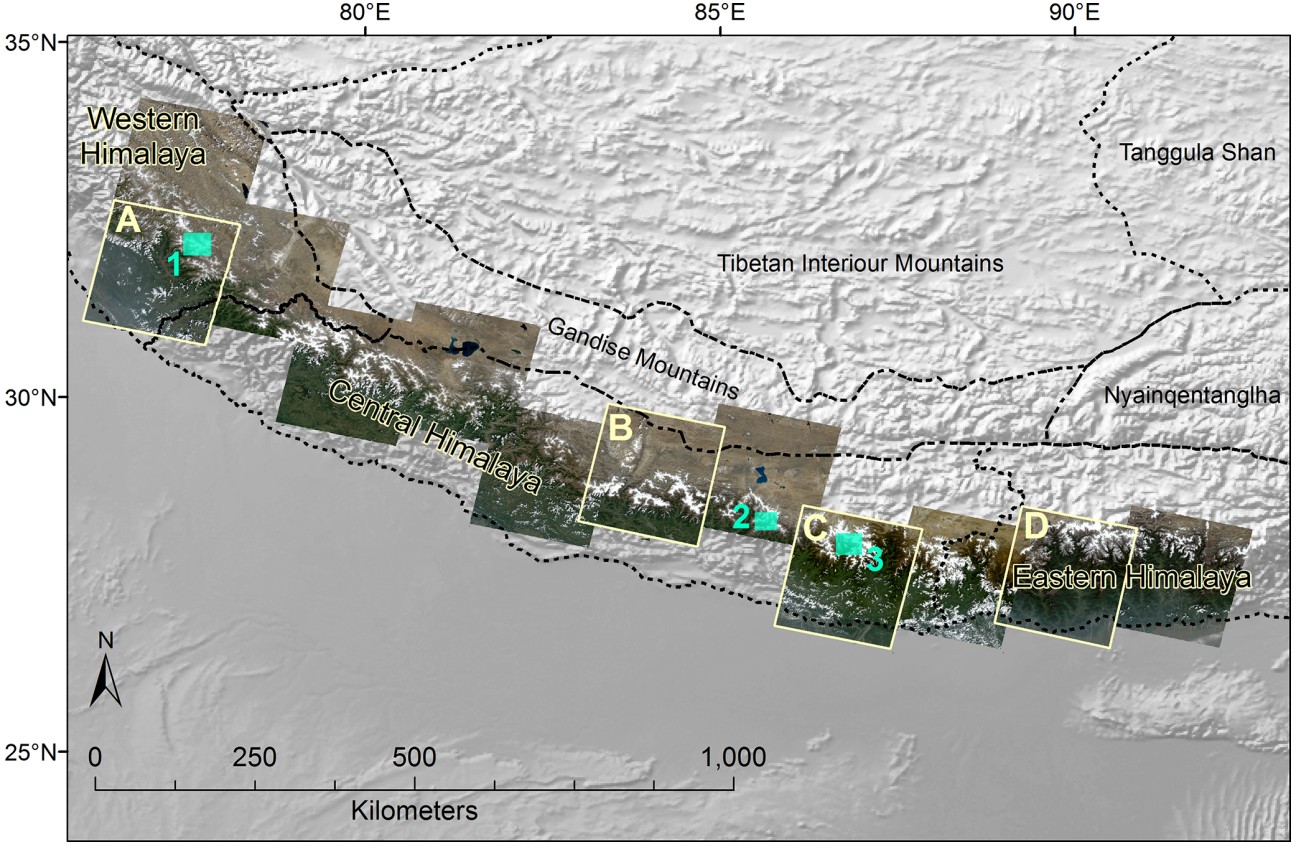

**Figure 1.** Himalaya study domain showing the large climatic regions from Bolch et al. (2019) as dotted black lines and the studied regions (western, central and eastern). The figure also shows the selected domains across the monsoonal gradient discussed in the text, shown as light-yellow outlines and labelled as follows: A, Lahaul–Spiti in the monsoon-arid transition zone of the western Himalaya; B, Manaslu; C, Khumbu and parts of eastern Tibet in the central Himalaya; D, Bhutan in the eastern Himalaya. Turquoise boxes represent the pond validation sites: 1, Lahaul–Spiti glaciers; 2, Langtang glaciers; 3, Khumbu glaciers. Image footprints are the true colour composite of Landsat 8 OLI (bands 4,3,2) scenes used in this study and described in Table 1.

glacier mass balance using the traditional glaciologic method (Wagnon et al., 2013), the geodetic method (Bolch et al., 2008; Nuimura et al., 2012; Brun et al., 2017; Bolch et al., 2011; Rieg et al., 2018), energy balance models (Rounce and McKinney, 2014; Rounce et al., 2015; Kayastha et al., 2000), debris cover characteristics (Iwata et al., 1980; Watanabe et al., 1986; Nakawo et al., 1999; Iwata et al., 2000; Casey et al., 2012; Yukari et al., 2000) and surface velocity (Quincey et al., 2009). Rates of change of the debris-covered glacier areas in the Khumbu vary from $-0.12 \pm 0.05\,\%\,a^{-1}$ from 1962 to 2005 (Bolch et al., 2008) to $-0.27 \pm 0.06\,\%\,a^{-1}$ from 1962 to 2011 (Thakuri et al., 2014). Supraglacial ponds cover $\sim 0.3\,\%$ to 7 % of the glacierized area in the Khumbu based on high-resolution Pléiades data (Watson et al., 2017a; Kneib et al., 2020; Salerno et al., 2012); ice cliffs cover between 1 % and 9.2 % of the glacier areas (Brun et al., 2018; Watson et al., 2017a; Kneib et al., 2020).

To examine and highlight regional differences in the composition of the debris-covered surfaces, we use four sub-regions selected across monsoonal gradients as de-fined in the literature, corresponding to the Landsat scenes ($\sim 32\,919\,km^2$) shown on Fig. 1 (Bookhagen and Burbank, 2010; Thayyen and Gergan, 2010; Barros and Lang, 2003). The Lahaul–Spiti region in the western Himalaya is in the monsoon-arid transition zone, characterized by monsoon precipitation during the summer and precipitation from the westerlies in the winter (Thayyen and Gergan, 2010). The Manaslu and Khumbu regions in the central Himalaya, and the Bhutan region in the eastern Himalaya, are all under the influence of the Indian summer monsoon, which brings large amounts of precipitation during the summer months (June to September) (Barros and Lang, 2003; Bookhagen and Burbank, 2006) (Fig. 1).

To validate the performance of the spectral unmixing as a basis for estimating pond coverage, we used debris-covered glacier zones at three validation sites (700–1150 km²), selected across the wider Himalaya domain from the Khumbu, Langtang and Lahaul–Spiti regions (Fig. 1). Supraglacial ponds on these glaciers were mapped using OBIA methods on high-resolution imagery (Sect. 2.6).

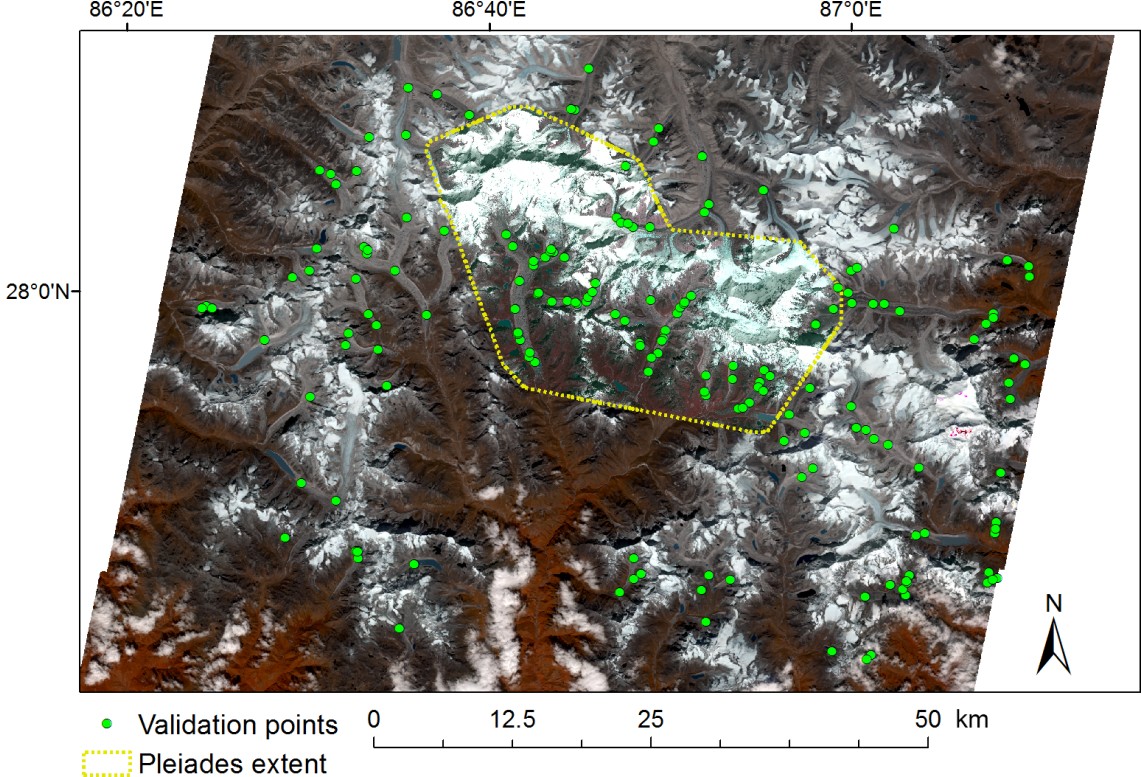

**Figure 2.** The Khumbu test region in Nepal showing the RapidEye image of 9 October 2015 (bands 5, 4 and 3) and the Pléiades image of 7, 19 and 20 October 2015 (bands 4, 3 and 2) (yellow dotted outline). Vegetation appears in dark red/brown; ponds display various shades of turquoise. Green dots represent the ground truth points digitized on the high-resolution images and used for the accuracy assessment of the linear spectral unmixing.

## 2.2 Remote sensing data

The satellite data used for spectral unmixing comprise of 13 Landsat 8 OLI images covering the Himalaya domain (Fig. 1). Characteristics of these images are given in Table 1. These were top-of-atmosphere registered, radiometrically calibrated and orthorectified imagery (level L1TP - T1), available at 30 m spatial resolution in the visible to short-wave infrared since 2013 (Wulder et al., 2019; USGS, 2015). We selected scenes from the post-monsoon period only (September to November) in order to minimize cloud and snow cover occurrence (Bookhagen and Burbank, 2006). In addition, Landsat scenes across the domain were selected around the same date as much as possible to minimize seasonal differences in surface conditions, notably seasonal changes in pond occurrence (Miles et al., 2017). All chosen images were acquired around the same time of the day (05:00 UTC time), with similar solar azimuth ($\sim 143°$) and zenith angle ($\sim 30°$). This is important to ensure that differences in surface conditions were minimal. Where the 2015 images had too much cloud or snow, we selected images for the same season in 2014 and 2016 (Table 1). We acknowledge that this choice may introduce some uncertainties due to the temporal difference, which we discuss later (Sect. 4.6).

The Landsat 8 OLI scene from Khumbu (30 September 2015) was chosen as reference for method development and testing. We also performed a second spectral unmixing on an additional 2016 Landsat 8 OLI scene for Lahaul–Spiti in the western Himalaya (Table 1) in order to have an analysis that was coincident with the high-resolution data used to validate the supraglacial pond mapping within this region.

For calibration and validation of the spectral unmixing products at specific locations, we used a combination of high-resolution optical imagery from Pléiades and Planet (Table 1). The Pléiades 1A satellite sensor acquires tri-stereo high-resolution data (0.5 m spatial resolution in the panchromatic band and 2 m in the multispectral bands, blue to near-infrared), with 20 km image swath at nadir (Table 1). Three Pléiades scenes from 2015 (7, 19 and 20 October) covered the north, north-east, and south-east parts of Khumbu (Rieg et al., 2018) and offered the closest match to the date of the reference Landsat image (30 September 2015) (Fig. 1); these Pléiades scenes were cloud-free and snow-free over the debris-covered part of the glaciers. The scenes were provided as three sets of triplets of primary data (1A) and were orthorectified in the Leica Photogrammetry Suite in ERDAS Imagine 2013 (ERDAS, 2010 TS10) using the Pléiades Rational Polynomial Coefficient model and the Pléiades DEM

**Table 1.** Satellite imagery used in this study.

| Sensor | Path/row | Product | Date | Bands | Cell size (m) | Swath width (km) | Usage |
|---|---|---|---|---|---|---|---|
| Landsat 8 OLI | 137/41<br>138/41<br>139/41<br>140/41<br>141/40<br>142/40<br>143/40<br>144/39<br>145/39<br>146/38<br>147/37<br>147/38<br>147/38 | L1TPT1 | 25 Nov 2014<br>19 Nov 2015<br>9 Oct 2015<br>30 Sep 2015<br>7 Oct 2015<br>1 Nov 2016<br>5 Oct 2015<br>10 Sep 2015<br>3 Oct 2015<br>8 Sep 2015<br>15 Sep 2015<br>15 Sep 2015<br>19 Oct 2016 | Band 1 Visible<br>0.43–0.45 μm<br>Band 2 Visible<br>0.450–0.51 μm<br>Band 3 Visible<br>0.53–0.59 μm<br>Band 4 Red<br>0.64–0.67 μm<br>Band 5 Near-IR<br>0.85–0.88 μm<br>Band 6 SWIR 1<br>1.57–1.65 μm<br>Band 7 SWIR 2<br>2.11–2.29 μm | 30 | 185 | Spectral unmixing |
| Pléiades | – | Level 1A | 7 Oct 2015<br>19 Oct 2015<br>20 Oct 2015 | Blue 430–550 nm<br>Green 490–610 nm<br>Red 600–720 nm<br>Near IR 750–950 nm | 2 | 20 | Visual checking of Landsat endmembers; pond validation (Khumbu area) |
| RapidEye | | Level 3A | 9 Oct 2015 | Green 520–590 nm<br>Red 630–685 nm<br>Red edge 690–730 nm<br>Near-IR 760–850 nm | 5 | 77 | Visual checking of Landsat endmembers (Khumbu area) |
| PlanetScope | | Level 3A | 19 Oct 2016<br>20 Oct 2016 | Blue 455–515 nm<br>Green 500–590 nm<br>Red 590–670 nm<br>Near IR 780–860 nm | 3 | 24.6 × 16.4 | Additional pond validation (Lahaul–Spiti area) |

(1 m) previously generated using semi-global matching (Rieg et al., 2018). The individual image scenes were mosaicked to a single image using nearest neighbour at 2 m spatial resolution. In addition, a RapidEye level 3A analytic ortho-tile from 9 October 2015 from Planet (Planet_Team, 2017 TS11) was used in addition to Pléiades in the Khumbu in order to cover a wider region to better overlap the Landsat scene. This RapidEye scene consists of orthorectified, surface reflectance data at 5 m spatial resolution and five multispectral bands, projected to UTM coordinates. A PlanetScope ortho-tile from 19 October 2016 (3 m spatial resolution, 4 multispectral bands) was used in the Lahaul–Spiti area to validate the ponds resulting from unmixing the 2016 Landsat 8 scene for this region (Table 1). Both RapidEye and PlanetScope tiles obtained from Planet were mosaicked to single scenes using nearest neighbour. These have a stated positional accuracy of $< 10$ m, reported as root mean square error, RMSE (Planet_Labs, 2021 TS12).

We co-registered all high-resolution images and the corresponding Landsat 8 OLI images using the Co-registration of Optically Sensed Images and Correlation (COSI-Corr) routine (Leprince et al., 2007) implemented in ENVI 5.5 Classic (L3Harris Geospatial, Boulder CO). For the Pléiades image, after co-registration with 20 tie points and a second-order polynomial transformation (RMSE = 1.3 m),

image displacements were $-0.16$ m in the E/W direction and 0.12 m in the N/S direction. The Planet RapidEye and PlanetScope scenes were co-registered on the Landsat 8 OLI with 15 and 10 tie points (RMSE = 5 and 1.6 m, respectively), yielding offsets of $\sim 1.1$ to 1.7 m in the E/W direction and 0.09 to 0.5 m in the N/S direction after co-registration. These offsets were below the spatial resolution of all scenes (2–5 m).

## 2.3 Atmospheric and topographic corrections

All Landsat 8 OLI scenes were corrected to minimize atmospheric effects due to scattering or absorption from atmospheric gases, aerosols and clouds. We used the open-source Atmospheric and Radiometric Correction of Satellite Imagery (ARCSI v 3.1.6) routine based on the 6S algorithm (Vermote et al., 1997). We applied the STDSREF option in ARCSI with the shadow option, which provided standardized surface reflectance products for all the scenes where deep shadows were masked out as NoData. ARCSI allows for global and local viewing and solar geometries using physically based illumination and reflectance corrections based on topographic data (Shepherd and Dymond, 2003), a specified atmospheric profile, an aerosol optical thickness (AOT) value and sensor geometry. These settings are impor-

tant for minimizing differences in surface conditions among the various scenes. The AOT value was automatically derived in ARCSI by a numerical inversion of the surface reflectance on an image basis using the simple dark object subtraction technique (DOS) from the blue band, yielding an AOT of 0.05 for the 30 September 2015 Khumbu scene. To validate the performance of the DOS technique for the atmospheric profile representation in our study area for this date, we validated the estimated AOT against level 1.5 data at reference wavelength of $\lambda = 500$ nm aerosol size from AERONET (https://aeronet.gsfc.nasa.gov/, last access: TS13) (Giles et al., 2019) and against daily forecast global reanalysis of total optical depth at multiple wavelengths from the Copernicus Atmospheric Monitoring Service (CAMS) (https://atmosphere. copernicus.eu/catalogue#/, last access: TS14). The AOT values obtained using the DOS method (0.05) were consistent with the ones calculated from AERONET and CAMS (0.07 and 0.05, respectively). In the Himalaya, we can generally assume relatively clean atmospheres and thus consider that low AOT values are reasonable (P. TS15 Bunting, Aberystwyth Univ., personal communication, February 2021). Our choice of a constant AOT value in high environments is in line with findings from other studies (Gillingham et al., 2013; Matta et al., 2017). Surface topography used for the atmospheric and topographic corrections was based on the ALOS Global Digital Surface Model (AW3D30 version 2.2, at 30 m) (JAXA, 2019 TS16), constructed from data acquired from 2006 to 2011. The vertical accuracy of $\sim 10$ m in eastern Nepal (Tadono et al., 2014) is superior to that of Shuttle Radar Topography Mission (SRTM) DEM (23.5 m, reported by Mukul et al., 2017), because it contains fewer data voids and provides better shadow rendering in our area.

## 2.4 Supraglacial debris cover data

In this study, we constrained our analysis over supraglacial debris surfaces, extracted from the database of global distribution of supraglacial debris cover (Scherler et al., 2018) and referred to hereinafter as the "SDC". Debris-covered glacier outlines in this dataset were derived from Landsat 8 OLI and Sentinel-2 data using automated approaches on Google Earth Engine by excluding clean ice and snow from glacier areas within the limits of the Randolph Glacier Inventory (RGI v.6) (RGI_Consortium, 2017 TS17). Outlines span the period 1998 to 2001 for the central and eastern Himalaya, the year 2002 for the western Himalaya (monsoon-dry transition zone) and mostly the year 2010 for glaciers in China. In this study, the outlines obtained from the SDC dataset required pre-processing because supraglacial ponds along with other surfaces such as nunataks were represented as "holes" in this dataset. This caused "NULL geometry" errors due to unclosed polygons, duplicated vertices, etc. We fixed these errors in the SDC polygons using the Repair Geometry command in ArcGIS v10.8., in order to "fill" the holes so that these were included in the SDC polygons. For the test

Khumbu area, we removed supraglacial debris polygons with an area less than 0.01 km$^2$, which proved to be erroneous areas upon visual examination, i.e. sliver polygons or isolated bare land pixels. Such unwanted small polygons typically result from polygon overlays and do not represent a physical entity on the ground (Delafontaine et al., 2009).

## 2.5 Spectral unmixing background and set-up

In remote sensing, the reflectance spectrum of any image pixel represents an average of the materials on the ground, present in various proportions within that pixel (Keshava and Mustard, 2002). These "mixed pixels" are a common occurrence and are especially a concern in low- to medium-resolution imagery, including Landsat. In the case of debris-covered glacier tongues, constituent materials include various types of rock debris and/or ice cliffs, supraglacial ponds, and vegetation in various proportions (Rounce et al., 2018). Spectral unmixing techniques serve to quantify mixed spectra and to decompose each pixel into its constituent materials based on their characteristic, distinct spectral signatures. These materials are referred to as "pure" endmembers (Painter et al., 2009; Keshava and Mustard, 2002) and are either extracted from the image itself before unmixing using unsupervised techniques or supplied by the user using a priori knowledge (Painter et al., 2009; Keshava and Mustard, 2002; Dixit and Agarwal, 2021). The relationship between the fractional abundance of each material and its spectra is most often defined as a linear combination of the spectral reflectance of the distinct constituent materials. This is implemented as linear mixing models (LMMs), used for example to distinguish among vegetation, rock or different snow grain sizes (Painter et al., 2009). LMMs are easy to implement and are therefore widely used (Dixit and Agarwal, 2021; Keshava and Mustard, 2002). In contrast, nonlinear mixing models take into account multiple scattering between surfaces and are used in forested areas where canopy height or particulate mineral mixtures are in close association (Roberts et al., 1993). They are more realistic but are also more difficult to implement (Dixit and Agarwal, 2021).

To yield physically meaningful results, fractions obtained from spectral unmixing should ideally comply with two major constraints: (a) the non-negativity (or positivity) constraint (i.e. fractions should not be negative) and (b) the sum to unity (i.e. for each pixel, fractions should add up to 1) (Keshava and Mustard, 2002). The non-negativity condition is recommended because negative reflectance values have no physical meaning, and the sum-to-unity constraint is recommended when very dark endmembers such as shadows are targeted or for unmixing radiance or thermal infrared emissivity. Models that comply with both conditions (called "fully constrained models") are difficult to achieve because they require perfect knowledge of the system, which is rarely feasible. Furthermore, fully constrained models have been shown to produce unrealistic fractions in poorly defined areas or ar-

Please note the remarks at the end of the manuscript.

eas of low illumination (Cortés et al., 2014). In this study, we applied a LMM with endmembers extracted from the Landsat 8 OLI image itself, and we constrained our analysis over the supraglacial debris cover only to reduce model complexity. We used the LMM implementation in the ENVI 5.5 software (L3Harris Geospatial, Boulder CO).

### 2.5.1 Endmember selection and spectral signatures

The selection of endmembers is crucial in determining the accuracy and reliability of the spectral unmixing (Song, 2005; Dixit and Agarwal, 2021), and it requires some trial and error as well as a priori knowledge. We selected the endmembers within the debris-covered areas in the Khumbu domain, based on the reference Landsat 8 OLI scene (30 September 2015). Prior to this, we performed a forward minimum noise fraction transform on the Landsat scene (Green et al., 1988), which consists of a linear transformation of the data based on principal component analysis and allows us to estimate noise in the bands. All bands had eigenvalues $> 1$, so we determined the dimensionality of the Landsat data as $n = 7$. We used the unsupervised pixel purity index routine in ENVI to find pure pixels in an automated manner. This routine outputs a data cloud where the value of each point indicates the number of times each pixel was marked as extreme, thus representing pixels with the highest occurrence in the image. We optimized the pure pixel extraction using various numbers of iterations (20 000 to 50 000) with thresholds ranging from 2 to 3 (i.e. 2 to 3 times the noise level in the data) until all pure pixels were detected. Larger thresholds identify more extreme pixels, but they are less likely to be pure endmembers. Pure pixels were identified on the Landsat 8 OLI scene as corresponding to six endmembers: clean ice, dry vegetation, clouds, light debris, dark debris and turbid water (Fig. 3). These were checked against co-registered Pléiades and RapidEye false colour composites in the Khumbu in order to minimize any occurrence of mixed pixels.

The spectra of the six endmembers (Fig. 4a) were statistically separable based on the Jeffries–Matusita and transformed divergence separability measures (Richards, 2013) (values $> 1.9$–2.0). We defined both light and dark debris endmembers on the basis of their spectral differences (Fig. 4a), also noted in other studies (Casey et al., 2012; Kneib et al., 2020). We visually compared these spectral signatures with those we acquired previously in the field on Mer de Glace (French Alps) using an SVC HR-1024 spectrometer (350 to 2500 nm) (Racoviteanu and Arnaud, 2013 TS19) (Fig. 4b), as well as with supraglacial debris spectra from other papers (Naegeli et al., 2015, 2017; Casey and Kääb, 2012). To minimize the number of endmembers, we made several choices: (a) we did not consider any snow; (b) we assumed the supraglacial ponds to be mostly of turbid type, i.e. those containing larger quantities of suspended sediments. We based this choice on results from Matta et al. (2017),

who reported 52 % of ponds in the Himalaya to have grey waters and 24 % blueish waters; the water spectra in Fig. 4a corresponds well with field-based spectra for other turbid lakes in the Khumbu, such as Chola Lake, reported in their study; (c) based on our field observations of high-altitude vegetation in the Khumbu (Fig. 3d), we defined the vegetation endmember as "dry vegetation", whose spectral signature (a) corresponds roughly to the graminoid shrubs or overgrown vegetation with a grass-like appearance typically found at high altitudes (Wehn et al., 2014); (d) deep shadows were previously removed during the topographic corrections with ARCSI and assigned to NoData so they were not considered as an endmember. We ran the LMM for various combinations and numbers of endmembers (three to six endmembers) and recorded the model RMSE for each combination. We examined the residuals (RMSE band) provided from the unmixing to determine areas of missing or incorrect endmembers; when this contained distinct features, it indicated poorly defined endmembers. We excluded the endmembers one by one and ran the LMM until we obtained a "salt and pepper" CE4 with no distinct features, indicating that no endmembers were missing or misidentified.

### 2.5.2 Surface classification from fractional maps

LMM routines result in a multi-band raster containing pixel-by-pixel fractional cover values for each class, which ideally range from 0 to 1. When we obtained negative values for a class, we assumed that the material was missing and forced these values to zero. Positive values were normalized by dividing each endmember fraction by the sum of the endmembers, so that the sum of the fractions of the various materials in each pixel added up to 1. This is a common procedure suggested by previous studies (Rosenthal and Dozier, 1996; Quintano et al., 2012; Cortés et al., 2014) when the sum-to-one condition is not satisfied.

For further analysis, we require maps of the surfaces rather than just a numerical value of area, so we classified the 30 m fractional maps by applying a threshold $\alpha$ to produce binary maps for each class. Previous studies used a minimum threshold of $\alpha = 0.4$ or 0.5; i.e. a pixel was assigned to a class if it contained a fraction of 40 %–50 % to 100 % of that constituent material (Hall, 2002 TS20). The thresholds varied by class, because any pixel contains a mixture of materials in various proportions (Sect. 3.1). Pixels which satisfy two different thresholds are categorized as "unclassified". For the supraglacial ponds in the Khumbu, we defined the water threshold quantitatively based on comparison of the LMM-derived pond areas against those derived from Pléiades for seven glaciers (Sect. 2.6), and we evaluated the sensitivity of the chosen water threshold. For the other classes, the thresholds were adjusted carefully based on visual interpretation against the Pléiades and RapidEye images in the Khumbu. The thresholds established for the Khumbu were applied over the entire Himalaya domain.

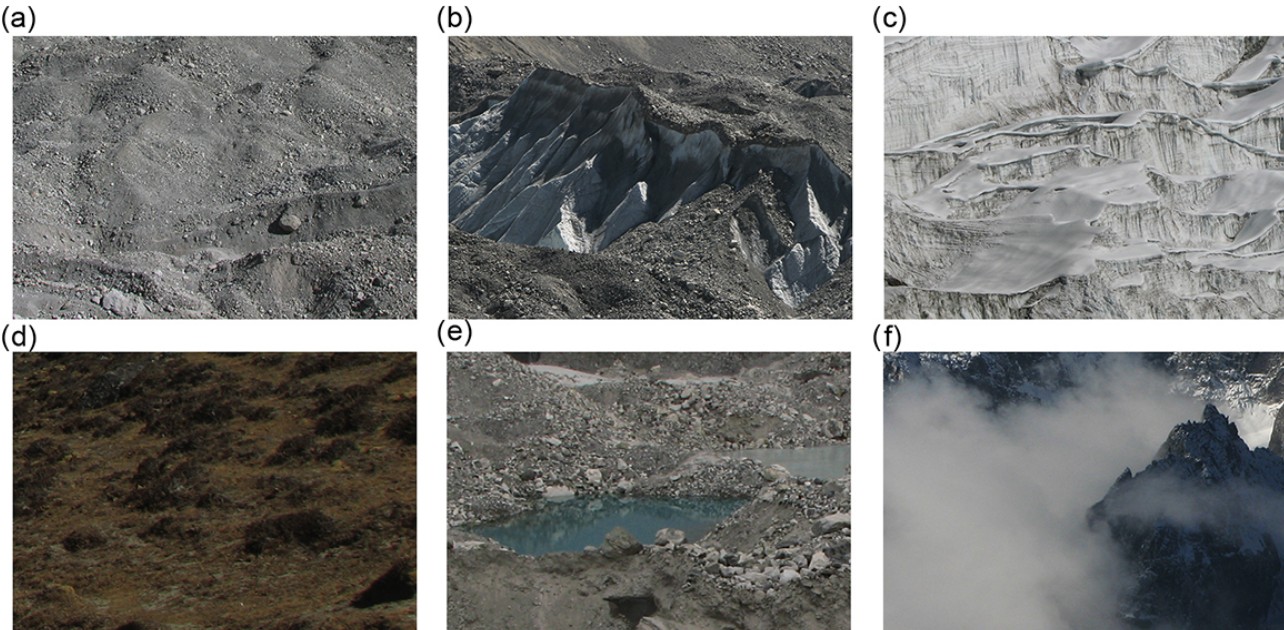

**Figure 3.** Types of surfaces present in the study area: **(a)** light debris cover (quartz, feldspar); **(b)** darker schistic debris with ice cliff; **(c)** clean ice with crevasses in the glacier ablation area; **(d)** graminoid shrub type vegetation (dry); **(e)** supraglacial lakes with different turbidity levels; **(f)** valley clouds. All photos were taken in the Khumbu region. Photo credit: Adina E. TS18 Racoviteanu.

**Table 2.** Summary of accuracy metrics per class for the Khumbu area, calculated based on the confusion matrix, including true positives (TP), false positives (FP), false negatives (FN) and true negatives (TN).

| Class | TP | FP | FN | TN | Recall | Precision | F score |
|---|---|---|---|---|---|---|---|
| Clean ice | 1 | 0 | 13 | 112 | 0.07 | 1.00 | 0.13 |
| Water (turbid) | 32 | 2 | 6 | 81 | 0.84 | 0.94 | 0.89 |
| Debris (dark) | 29 | 23 | 0 | 84 | 1.00 | 0.56 | 0.72 |
| Debris (light) | 21 | 8 | 9 | 62 | 0.70 | 0.72 | 0.71 |
| Clouds | 5 | 3 | 5 | 92 | 0.50 | 0.63 | 0.56 |
| Vegetation (dry) | 25 | 2 | 5 | 88 | 0.83 | 0.93 | 0.88 |

### 2.5.3 Accuracy assessment

The performance of the LMM was assessed both qualitatively (on the basis of visual interpretation and comparison with surfaces visible on the high-resolution Pléiades and RapidEye) and quantitatively (using established measures, i.e. RMSE, fraction value abnormalities and the residual band output in the LMM) (Gillespie et al., 1990). To quantitatively assess the ground accuracy of the LMM, we manually digitized 151 test pixels covering all six classes (10–38 pixels per class) on false colour composites of the Pléiades and RapidEye images in the Khumbu using a simple random sampling strategy. The reference points were chosen so that they were well distributed across the Khumbu (Fig. 2) and were taken to represent ground truth. The predicted class was compared to the ground truth at each pixel to generate a confusion matrix and to compute the overall accuracy of the model (percent pixels classified correctly). We also re-port class-specific metrics as true positives (number of pixels correctly classified and found in a class, TP), true negatives (number of correctly classified pixels that do not belong to a class, TN), false positives (number of pixels that were incorrectly assigned to a class, FP) and false negatives (number of pixels that were omitted from a class, FN) (Table 2). We calculated three metrics which are suitable for multi-class classification routines (Sokolova and Lapalme, 2009) as follows (Eqs. 1–3):

$$\text{Precision} = \frac{\text{TP}}{\text{TP} + \text{FP}}, \tag{1}$$

$$\text{Recall} = \frac{\text{TP}}{\text{TP} + \text{FN}}, \tag{2}$$

$$\text{F score} = \frac{2\text{TP}}{2\text{TP} + \text{FP} + \text{FN}}. \tag{3}$$

Precision measures the agreement between ground data and classified data, i.e. the probability that a pixel classified as

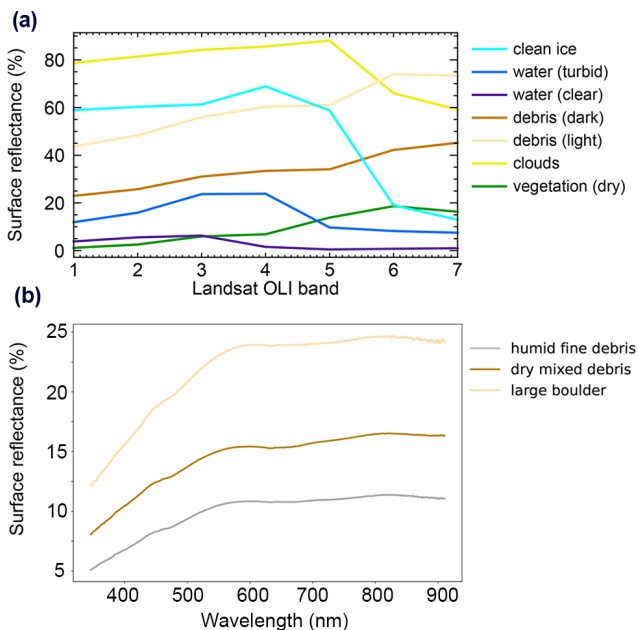

**Figure 4. (a)** Spectral signatures of endmembers extracted from Landsat 8 OLI bands 1 to 7 (30 September 2015 Khumbu image) after the atmospheric and topographic corrections; **(b)** field spectra from the debris cover part of Mer de Glace Glacier (France) shown for comparison purposes only.

water is indeed water on the ground. Recall measures the effectiveness of the classifier to identify a pixel in the class of interest, i.e. the percentage of results correctly classified by the algorithm. F score balances precision and recall as the harmonic means of the two and measures the relation between the pixels on the ground and those classified, i.e. the model accuracy for each class. For all metrics, a poor score is 0.0 and a perfect score is 1.0.

### 2.6 Validation of supraglacial ponds with high-resolution data

We validated the performance of the spectral unmixing for supraglacial pond areas on the basis of high-resolution imagery for 6 to 7 debris-covered glacier extents at each of the three sites shown in Fig. 1. For the Khumbu and Lahaul–Spiti glaciers, supraglacial pond areas were mapped from Pléiades and PlanetScope imagery, respectively (Table 1), using OBIA techniques (Blaschke et al., 2014) implemented in the ENVI Feature Extraction Module (Harris_Geospatial, 2017). In the Khumbu, the Pléiades images were acquired several weeks apart from the date of the Landsat scene in some parts of the region (see Table 1), but we assume minimal lateral expansion between the two dates, as discussed by Watson et al. (2018). For the Langtang region, we validated our LMM-derived pond areas with those reported for seven glaciers based on SPOT7 satellite imagery in Steiner et al. (2019). The OBIA method used for Khumbu and Lahaul–Spiti con-

sisted in a segmentation-only extraction workflow on the visible bands of Pléiades and/or PlanetScope, with an edge algorithm (to delineate the pond segments), a fast lambda setting (to merge adjacent segments with similar colours and borders) and a texture kernel size of 3 pixels (suitable for segmenting small areas). The scale and merge levels were adjusted against colour composites to prevent over-segmenting and to combine different segments into one pond. The resulting polygons were further manually corrected (split, merged or digitized) for any missing and/or shaded areas beneath ice cliffs as described in Watson et al. (2017a). Our aim was not to construct a sophisticated OBIA classification scheme but rather to use the feature extraction module as a time-saving strategy and to add objectivity to the manual digitization.

### 2.7 Auxiliary region-wide datasets

We explored the dependency of the resulting supraglacial pond cover incidence on topographic variables: elevation bands above the termini, slope and aspect of the debris cover areas. These were calculated over the debris-covered parts of the glaciers on the basis of the AW3D30 DEM (30 m). Only glacier polygons with area larger than $1\,km^2$, resulting in a subset of 408 glaciers, were selected from the SDC database over the Himalaya domain for an in-depth glacier-by-glacier analysis. The area threshold was applied in order to remove spurious small bare land patches or isolated debris pixels present in the SDC database. While the vast majority of glaciers in the Himalaya are smaller than $1\,km^2$, these are mostly clean glaciers (Racoviteanu et al., 2015). In addition to the glacier-by-glacier basis analysis, we also binned the topographic variables, i.e. 100 m elevation, 2° slope and 45° aspect, and summarized the pond incidence in each bin.

We explored spatial patterns in the pond incidence and supraglacial vegetation with respect to regional climate gradients, average glacier mass balance and average surface velocity. Climate data (total precipitation and average temperature) were obtained from ERA5-Land, which provides gridded monthly average means at 0.1° × 0.1° of land surface properties (Copernicus Climate Change Service, 2019 TS21) (Muñoz-Sabater, 2019 TS22). Gridded glacier thickness change at 30 m resolution for the period 2000–2019 was obtained from Shean et al. (2020). Glacier surface velocities for the period 2013–2015 based on Landsat data were obtained from Dehecq et al. (2015). All topo-climatic variables were binned and averaged over a 1° × 1° grid as used in other studies (e.g. Brun et al., 2017; Dehecq et al., 2019) to explore the topo-climatic controls on spatial trends in pond and vegetation incidence.

## 3 Results

### 3.1 Fractional maps

Here we present results of the unconstrained LMM, because this had a lower RMSE (0.6 %) compared to the partially constrained model run (RMSE of 1.5 %). The normalized fractional maps of the six surface types are presented in Fig. 5; fractional values ranged from 0.004 to 1. Fractional water values greater than 0.5 correspond to supraglacial ponds, visible for example at the termini of Ngozumpa and Khumbu glaciers (Fig. 6a and b). Light and dark debris was identified with a threshold of 0.25 and 0.40, respectively, defined visually on the basis of the Pléiades image. Dry vegetation patches generally exhibited pixel fractions greater than 0.65. Pixels with abnormally high positive fractional vegetation values were found in areas of healthy green vegetation and/or bare terrain, which should not be part of the debris-covered tongues, as will be discussed later (Sect. 4.5). Cloud pixels display fractional values greater than 0.45, although some pixels were mixed with debris, particularly at cloud shadow areas. For clean ice, fractional values were rather low (0.20) and ranged from 0 (areas which might have some degree of dirty, dark ice with a lower albedo) to 1 (small number of clean ice pixels found in the upper areas of supraglacial debris).

### 3.2 Accuracy of the LMM-based classification for the Khumbu

Accuracy measures presented in Table 2 for the Khumbu domain show that errors were not evenly distributed among classes. For the water and vegetation classes, recall score was 0.83 to 0.84, respectively, with a precision of 0.94 and 0.93, respectively (Table 2). For these classes, the LMM achieved a balance of precision and recall metrics, with a high F score of $\sim 0.9$ indicating an accurate model. For the debris classes, the model was reasonable but not outstanding, with an F score of $\sim 0.7$ and lower precision score for dark debris (0.56) compared to light debris (0.72) (Table 2). This suggests that in the case of dark debris, the LMM model was less accurate than light debris, because pixels from other classes (clean ice, water and light debris) got mistakenly assigned to this class. Clouds were classified with low precision and low recall scores (F score of $\sim 0.5$), which means that the LMM performed relatively poorly for this class and it also missed 50 % of the cloud pixels. There was confusion between clean ice and cloud pixels, i.e. clean ice pixels were mistakenly included in the cloud class. Clean ice was the most poorly classified, with a recall score close to 0 and F score of 0.13; one ice pixel was correctly identified, but other surfaces were confounded with ice. We attribute this to the poorly defined ice class in the model data (i.e. limited number of pure ice pixels used to extract the spectral signature). Based on these measures, we note that overall

**Table 3.** Sensitivity analysis of the supraglacial pond area for the seven reference glaciers in the Khumbu domain, obtained using various thresholds applied to the fractional water maps.

| Glacier | Surface area (km$^2$) | | |
| --- | --- | --- | --- |
| | Fractional water > 0.4 | Fractional water > 0.45 | Fractional water > 0.5 |
| Khumbu | 0.45 | 0.32 | 0.20 |
| Lhotse | 0.07 | 0.06 | 0.05 |
| Lhotse Nup | 0.03 | 0.03 | 0.02 |
| Ngozumpa | 0.79 | 0.66 | 0.50 |
| Nuptse | 0.09 | 0.05 | 0.03 |
| Changri Nup | 0.25 | 0.19 | 0.09 |
| Gaunara | 0.16 | 0.12 | 0.07 |
| Total pond coverage | 1.8 | 1.4 | 1.0 |

the LMM most accurately classified the water and vegetation classes, with reasonable performance for the light debris class but poor performance for clean ice and clouds. The overall accuracy of the LMM-based classification of the six surfaces was 75 %; however, this is a rather coarse metric, and it does not indicate the specific performance of the model for each class, so we do not use this here as evaluation of the accuracy.

### 3.3 Supraglacial pond thresholds and validation

The sensitivity analysis of the pond areas obtained from LMM fractional maps with various thresholds (Table 3) indicates that there was up to 40 % variability in total pond area when compared to Pléiades-based ponds, depending on the glacier. A threshold of 0.5 applied to the water class (fractional water $> 0.5 =$ supraglacial ponds) yielded the best agreement with the total pond areas for the seven glaciers, obtained from OBIA mapping on the Pléiades image (1.0 km$^2$ compared to 1.1 km$^2$ for the total coverage, respectively, or a 9 % difference) (Table 4). For the Khumbu Glacier, LMM with a threshold of 0.5 yielded a pond area of 0.20 km$^2$ versus 0.23 km$^2$ from Pléiades (Table 4), which is in agreement with the area reported by Watson et al. (2017b) (0.24 km$^2$) using the same Pléiades image (7 October 2015).

In the Lahaul–Spiti region, for the seven glaciers we investigated, LMM yielded a total pond area of 0.14 km$^2$ (0.31 % of the total debris-covered area of the glaciers). The area mapped from PlanetScope image from the same date (19 October 2016) using OBIA yielded 0.10 km$^2$ (0.22 % of the debris-covered area) (Table 4).

In the Langtang region, for the six glaciers investigated in Steiner et al. (2019), our LMM-derived pond areas yielded a total of 0.17 km$^2$ pond area (0.64 % of the debris-covered area). Steiner et al. (2019) obtained a total pond area of 0.21 km$^2$ (0.86 % of the debris-covered area) for the same glaciers based on manual digitization by multiple analyses

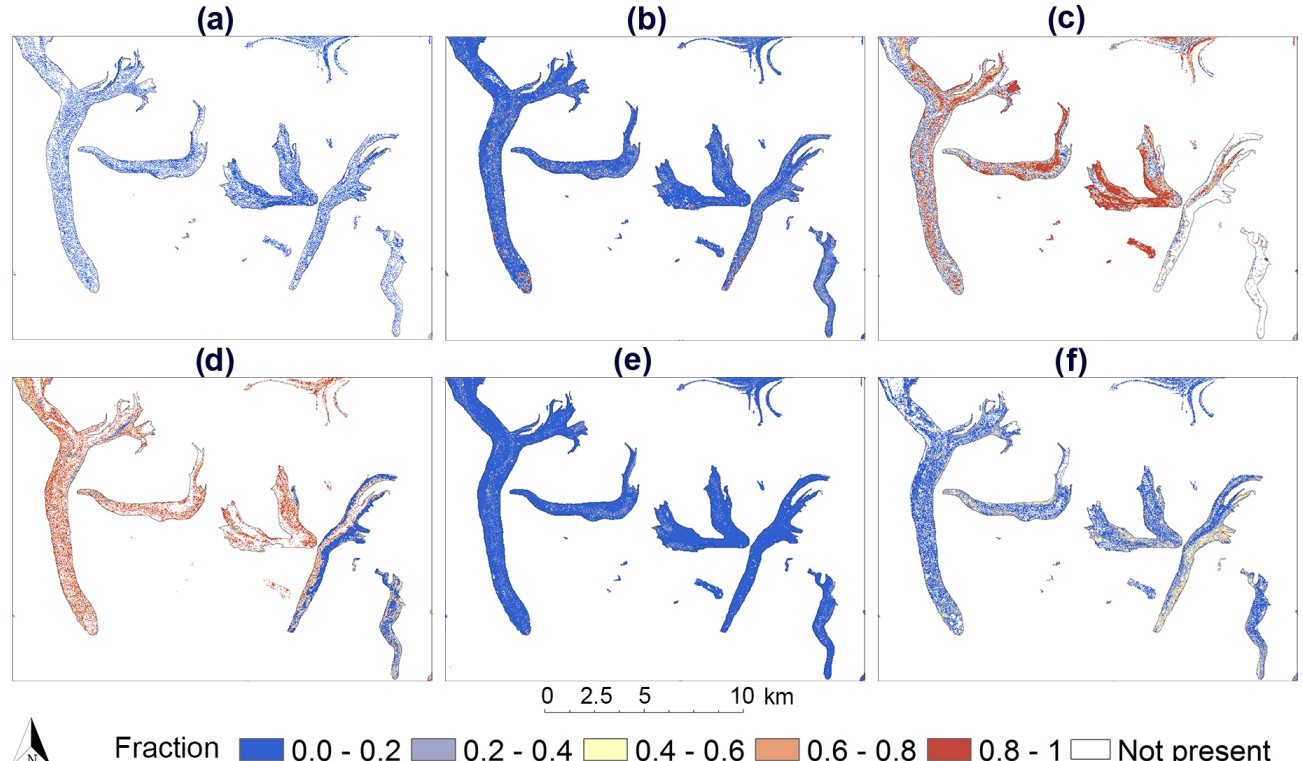

**Figure 5.** Fractional maps obtained from the LMM routine for a subset of the Khumbu area. Colour bars show the percentage covered by each type of material on a pixel-by-pixel basis: **(a)** clean ice; **(b)** turbid water; **(c)** dark debris; **(d)** light debris; **(e)** clouds; **(f)** dry vegetation.

from SPOT7 data for the same date as the Landsat. LMM underestimated the pond area by 0.05 km$^2$ (19 %), which is within the uncertainty range (21 %) reported for the ponds in the Langtang area by Steiner et al. (2019).

Visually, the spectrally unmixed pond pixels correspond well with the validation dataset, although there is a difference in the representation of the pond surfaces due to the spatial resolution (30 m Landsat vs. 2 m Pléiades) (Fig. 6). For Lhotse Glacier, the supraglacial pond area was slightly underestimated compared to Pléiades (Table 5) as can be seen on Fig. 6. This is perhaps due to the predominance of darker debris type on this glacier, some of which was confused with water, as shown by the accuracy metrics (Table 2). Similarly, in the Lahaul–Spiti region, locations of the supraglacial ponds correspond well between LMM and PlanetScope on Bara Shigri Glacier (Fig. 6c), but the small ponds are not identified using the water threshold of 0.5, which assumes that more than 50 % of the pixel area is covered by water.

### 3.4 Application to regional non-glacier lake databases

While supraglacial ponds are the focus of this study, we mention that LMMs can also be parameterized to map other lakes, by masking the debris-covered glacier areas and replacing the turbid water endmember with the clear water endmember, which has a lower spectral signature (Fig. 4a). This

is beyond the purpose of this study, but we provide an illustration of such an output for the terminus of Ngozumpa Glacier (Fig. 7). We present the ponds and lakes on the debris cover and outside it for comparison with two existing glacial lake databases constructed from the same year (2015 Landsat): the HMA v.1 lake dataset, derived using a normalized difference water index (Shugar et al., 2020), and HI-MAG constructed using a modified NDWI and manual corrections (Chen et al., 2021). A comparison with other global databases such as the Global Surface Water dataset (Pekel et al., 2016) was not undertaken here, as this has already been shown to underestimate the water occurrence over most of the Himalaya by Chen et al. (2021). With regards to HMA v.1 and HI-MAG datasets, Fig. 7 shows that the lake outlines obtained from spectral unmixing for the supra-glacier ponds at the terminus of Ngozumpa Glacier and the Gokyo Lakes outside the glaciers are outperforming both of the existing databases in this area. Our lake extents are consistent with the HMA v.1 lake extents outside debris cover (Fig. 7), and the surface area estimates agree quite well; for example, we calculated a difference of 5 % in the summed pond area over the three Gokyo Lakes (1.15 km$^2$ in our estimates vs. 1.09 km$^2$ in HMA v.1). The slight underestimate in the latter is due to simplification of the raster edges in the vector conversion process, visible in the lake extents. With regards to supraglacial ponds, for example Spillway Lake at

**Table 4.** Validation of the Landsat spectral unmixing for supraglacial pond coverage at selected glaciers at three sites across the Himalaya domain, shown in Fig. 1.

| Khumbu | | Landsat 8 spectral unmixing | | | Pléiades OBIA | | |
|---|---|---|---|---|---|---|---|
| Glacier | Debris area (km$^2$) | Pond area (km$^2$) | % coverage | Date | Pond area (km$^2$) | % coverage | Date |
| Khumbu | 7.50 | 0.20 | 2.80 | | 0.21 | 2.70 | |
| Lhotse | 5.20 | 0.05 | 0.90 | | 0.08 | 1.70 | |
| Lhotse Nup | 1.50 | 0.02 | 1.00 | | 0.02 | 1.60 | |
| Ngozumpa | 19.40 | 0.50 | 2.70 | 30 Sep 2015 | 0.59 | 3.00 | 7 Oct 2015 |
| Nuptse | 2.90 | 0.03 | 0.90 | | 0.03 | 1.00 | |
| Changri Nup & Shar | 7.30 | 0.09 | 1.30 | | 0.11 | 1.50 | |
| Gaunara | 5.20 | 0.07 | 1.40 | | 0.09 | 1.70 | |
| Total | 49.00 | 1.00 | 2.04 | | 1.10 | 2.24 | |
| Langtang | | Landsat 8 spectral unmixing | | | SPOT 7 manual digitization (from Steiner et al., 2019) | | |
| Lirung | 1.44 | 0.00 | 0.00 | | 0.00 | 2.70 | |
| Ghana[CE5] | 0.69 | 0.00 | 0.00 | | 0.00 | 1.70 | |
| Langshisha | 4.46 | 0.01 | 0.20 | | 0.01 | 1.60 | |
| Langtang | 16.17 | 0.15 | 0.92 | 7 Oct 2015 | 0.18 | 3.00 | 6 Oct 2015 |
| Sabalchum[CE6] | 3.44 | 0.01 | 0.33 | | 0.02 | 1.00 | |
| Lirung | 1.44 | 0.00 | 0.00 | | 0.00 | 1.50 | |
| Total | 26.20 | 0.17 | 0.64 | | 0.21 | 0.86 | |
| Lahaul–Spiti | | Landsat 8 spectral unmixing | | | PlanetScope OBIA | | |
| Yichu[CE7] | 5.7 | 0.002 | 0.000 | | 0.001 | 0.000 | |
| Dibi Ka[CE8] | 5.6 | 0.004 | 0.000 | | 0.009 | 0.000 | |
| Bara Shigri | 21.3 | 0.126 | 0.027 | | 0.076 | 0.016 | |
| Sara Umga | 7.8 | 0.007 | 0.001 | 19 Oct 2016 | 0.012 | 0.001 | 19 Oct 2016 |
| G077666E32079N | 0.7 | 0.000 | 0.000 | | 0.000 | 0.000 | |
| G077559E32106N | 3.2 | 0.000 | 0.000 | | 0.000 | 0.000 | |
| G077698E32078N | 1.2 | 0.001 | 0.000 | | 0.000 | 0.000 | |
| Total | 45.5 | 0.14 | 0.31 | | 0.10 | 0.22 | |

**Table 5.** Composition of the seven debris-covered tongues in Khumbu, expressed as percent coverage of each material with respect to the debris-covered zones of each glacier.

| Glacier | Clean ice (%) | Water turbid (%) | Debris dark (%) | Debris light (%) | Cloud (%) | Vegetation dry (%) |
|---|---|---|---|---|---|---|
| Khumbu | 0.4 | 2.8 | 17.2 | 79.3 | 0.0 | 0.3 |
| Lhotse | 0.2 | 0.9 | 91.1 | 7.5 | 0.0 | 0.4 |
| Lhotse Nup | 0.7 | 1.0 | 69.1 | 29.2 | 0.0 | 0.0 |
| Ngozumpa | 0.4 | 2.7 | 54.2 | 42.2 | 0.1 | 0.5 |
| Nuptse | 0.3 | 0.9 | 2.7 | 95.8 | 0.0 | 0.3 |
| Changri Nup | 1.4 | 1.3 | 76.0 | 20.9 | 0.0 | 0.5 |
| Gaunara | 0.9 | 1.4 | 65.6 | 30.5 | 0.0 | 1.6 |
| Average | 0.6 | 1.6 | 53.7 | 43.6 | 0.0 | 0.5 |

the terminus of Ngozumpa Glacier, our spectral unmixing technique maps most of these lakes, while both HMA v.1 and the HI-MAG datasets fail to detect all the supraglacial ponds. The HI-MAG detects more of the surface of Spillway Lake compared to HMA v.1, but the outlines are simplified and lack precision with respect to Landsat pixels (Fig. 7).

We did not simplify the lake and pond polygons, as this can introduce significant area errors.

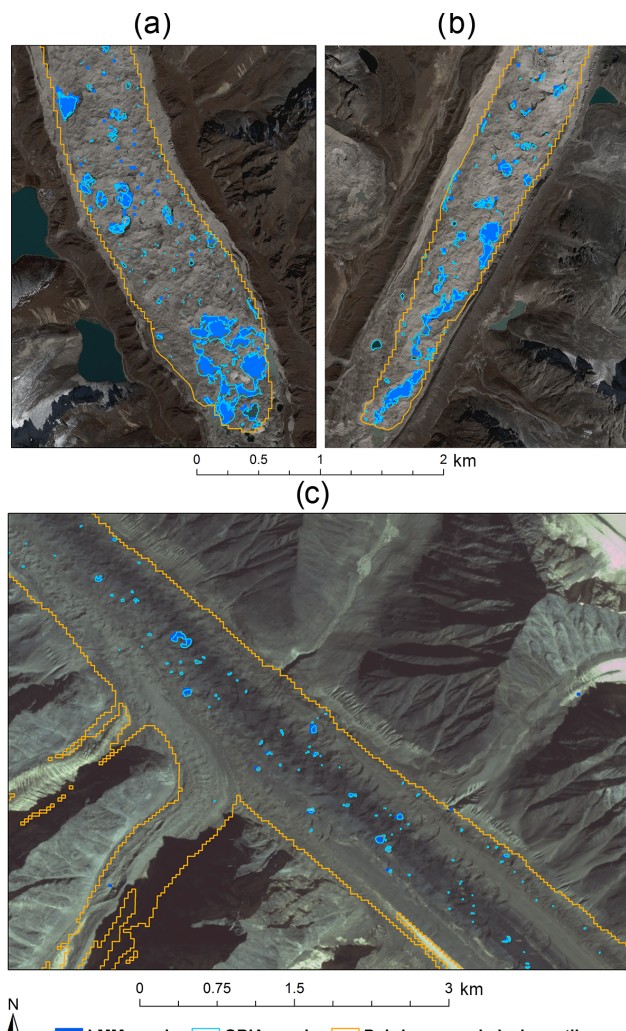

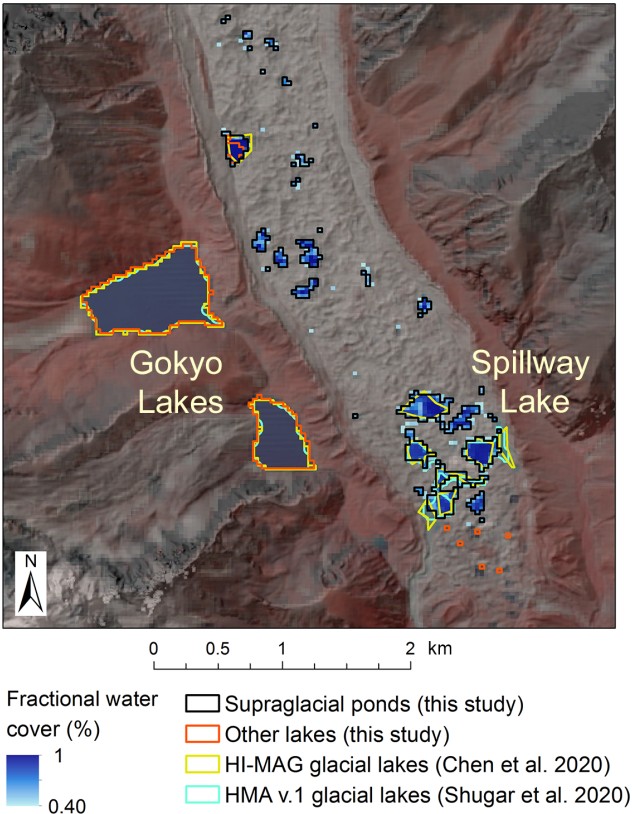

**Figure 7.** Comparison of the fractional ponds from this study with two recent lake datasets based on 2015 Landsat imagery (same as our study) for the Spillway Lake at the terminus of Ngozumpa Glacier and the Gokyo Lakes, with the Landsat colour composite of bands 5, 4 and 3 overlaid on shaded relief.

**Figure 6.** Comparison of the Landsat sub-pixel classified fractional ponds (dark blue) with OBIA pond outlines (light blue) based on high-resolution data for the termini of three glaciers: **(a)** Ngozumpa Glacier, **(b)** Khumbu Glacier and **(c)** Bara Shigri Glacier. The background images are colour composites (bands 1,2,3) of Pléiades imagery **(a, b)** and PlanetScope imagery **(c)**. Glacier outlines are from the SDC dataset (Scherler et al., 2018).

## 3.5 Composition of the debris cover glacier tongues: glacier to regional scale

### 3.5.1 Khumbu domain

For the seven debris-covered glacier tongues in the Khumbu (Fig. 8), the most prevalent materials detected using the LMM were dark and light debris, with an average of 53.7 % and 43.6 % of the supraglacial debris area, respectively (Table 5). The dark and light debris areas exhibit variable distribution patterns by glacier. For example, the debris-covered tongue of Nuptse Glacier in Khumbu is mostly covered by light debris (> 95 % of its area), while the opposite is true

for Lhotse Glacier, which is mostly composed of dark debris (> 91 %) (Table 5). Other glaciers in the eastern part of Khumbu, i.e. Kangshung Glacier, exhibit alternating bands of light and dark debris, where darker bands represent medial moraines (Fig. 8).

Exposed ice was detected in small quantities in the Khumbu, ranging from 0.2 % (Lhotse) to 1.4 % (Changri Nup) with an average of 0.6 % of the debris-covered areas (Table 5 and Fig. 9). Patches of supraglacial vegetation ranged from ∼ 0 % (Lhotse Nup Glacier) to 1.6 % (Gaunara Glacier), with an average of 0.5 % over the seven tongues (Table 5). Vegetation patches were found for several pixels corresponding to the lateral moraine of Ngozumpa Glacier, or larger patches at the terminus of Labeilong and Kazhenpu glaciers in China (Figs. 8. and 10). The supraglacial pond area in the Khumbu in 2015 ranged from 0.9 % (Lhotse and Nuptse glaciers) to ∼ 3 % of the debris-covered area (Ngozumpa and Khumbu glaciers), with an average of 1.6 % over the seven debris-covered glacier tongues and glacier-by-glacier variability (Table 5). The larger water coverage for Ngozumpa and Khumbu glaciers is consistent with the pres-

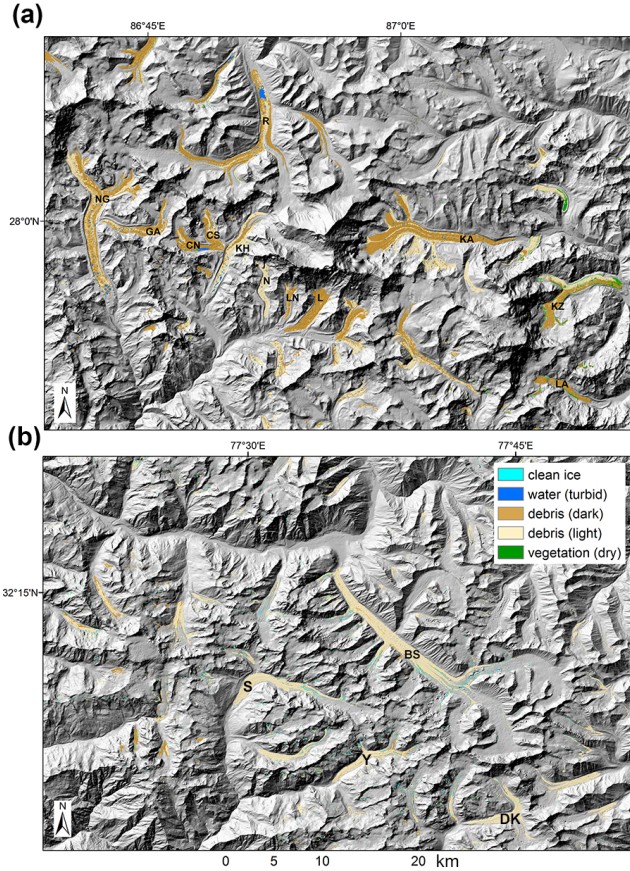

**Figure 8.** Composition of debris-covered glacier tongues shown for two of the domains showing glaciers discussed in the text: **(a)** subset of the Khumbu domain (NG: Ngozumpa Glacier; GA: Gaunara Glacier; CN: Changri Nup Glacier; CS: Changri Shar Glacier; KH: Khumbu Glacier; N: Nuptse Glacier; LN: Lhotse Nup Glacier; L: Lhotse Glacier; KA: Kangshung Glacier; KZ: Kazhenpu Glacier `CE9`; LA: Labeilong Glacier `CE10`) and **(b)** subset of the Lahaul–Spiti area (BS: Bara Shigri Glacier; S: Sara Umga Glacier; Y: Yichu Glacier; DK: Dibi Ka Glacier). Surfaces are shown on shaded relief from the AW3D30 DEM, with debris cover glaciers from the SDC dataset (Scherler et al., 2018). Note that the extent of Changri Nup incorrectly includes the inactive part of the glacier in this global dataset.

ence of large supraglacial ponds at the terminus of these two glaciers shown on Fig. 6.

### 3.5.2 Himalaya domain

Here we consider patterns across the whole analysed mountain range and also compare and contrast conditions in the four regions highlighted in Fig. 1. Light debris is prevalent over the entire Himalayan domain, comprising almost 3 times the extent of dark debris (60.9 % vs. 23.8 %, respectively). There is a slight regional variability in the occurrence of light debris, but all regions exhibit similar patterns in terms of the proportion of light and dark debris (Table 6). Glaciers

in the western part of the Himalaya are mostly composed of supraglacial light debris, which presumably reflects the nature of the underlying bedrock geology here (Searle et al., 1987).

We detected a higher percent coverage of clean ice/snow within the debris-covered area for the entire range (5.6 % of the debris) with respect to the reference Khumbu domain (0.6 % on average) (Table 6). At the date of the analysis (September to October 2015), some of the debris-covered glaciers in the eastern part (Bhutan) exhibited snow on the upper parts of the supraglacial debris, perhaps due to early snowfalls common in this area at this time of the year.

Cloud coverage amounted to 45 km$^2$ (2.0 % of the debris-covered area) over the entire range, with less coverage in Lahaul–Spiti and Khumbu (1.6 % and 0.6 %, respectively) compared to the Bhutan domain (6 %).

Supraglacial vegetation covered a total of 4.5 % of the debris-covered parts of glaciers over the Himalaya domain, with less coverage in the western part (Lahaul–Spiti, 1.6 % of the debris cover) than in the central and eastern Himalaya regions (Khumbu and Bhutan domains, at $\sim$ 3 %). We show examples of the vegetation maps obtained from the LMM on Kazhenpu Glacier in China in Fig. 10a. On other glaciers, such as Labeilong Glacier (Fig. 10b), these values might be slightly overestimated because the SDC dataset included patches of healthy vegetation as part of the debris cover.

The supraglacial pond dataset over the Himalaya domain consists of a total of 18 325 ponds ranging in area from 0.0009 to 0.002 km$^2$. Ponds accounted for an area of 47 km$^2$ (2.1 % of the total supraglacial debris cover), with marked regional variability among western Himalaya (Lahaul–Spiti: 0.3 % of the supraglacial debris), central Himalaya (Khumbu: 1.6 % and Manaslu: 2.6 %) and eastern Himalaya (Bhutan: 4.9 %) (Table 6).

### 3.6 Glacier-by-glacier pond and vegetation coverage

The 408 debris-covered glacier tongues selected from the SDC dataset for the in-depth analysis (cf. Sect. 2.7) ranged in area from 1 to 37 km$^2$, with an average area of 3.9 km$^2$ and a mean slope of 12.7°. The supraglacial pond and vegetation coverage of these glaciers shows heterogeneous patterns (Fig. 11a and b). Both supraglacial ponds and vegetation cover a relatively small percent of the debris-covered glacier areas in the western Himalaya (0 % to 2.5 %) compared to the central and eastern parts. We note some clusters of higher percentage occurrence of both ponds and vegetation in these two regions (7.5 %–10 % for ponds and 20 %–40 % for vegetation, respectively) (Fig. 11a and b). The glacier-by-glacier analysis of pond coverage with respect to minimum debris-covered glacier elevation did not yield a clear trend, suggesting that ponds do not occur necessarily on glaciers situated at lower altitudes. Similarly, supraglacial vegetation coverage did not display significant dependencies on either average slope or minimum elevation of the debris-covered tongues.

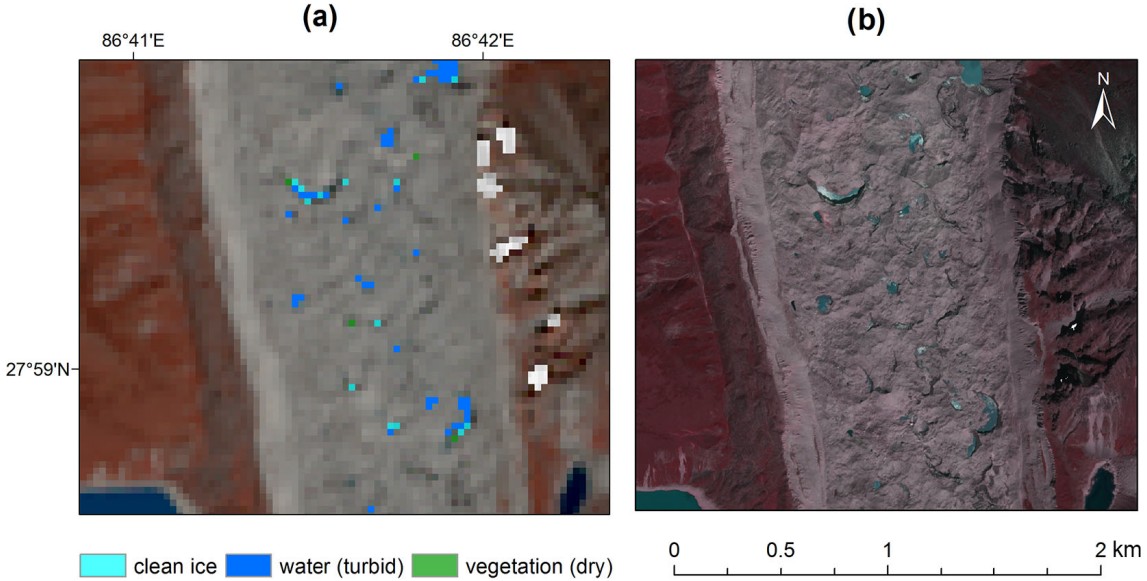

**Figure 9.** Ice pixels detected by the LMM at the surface of Ngozumpa Glacier in the Khumbu region. **(a)** Landsat 8 OLI false colour composite bands 5, 4, 3 and unmixing results for ice, water and vegetation classes only; **(b)** Pléiades colour composite (bands 4, 3, 2) shown for comparison, with vegetation shown as red shades. Ice cliffs display the typical crescent moon shape. White pixels in panel **(a)** correspond to NoData in areas of topographic shadows, resulting from the topographic correction routine.

**Table 6.** Composition of the debris-covered glaciers over the entire Himalaya domain and four selected sub-domains along the monsoonal gradient and for the entire domain, listed from west to east. Debris-covered glacier areas are based on the SDC dataset (Scherler et al., 2018).

|  | Lahaul–Spiti | | Manaslu | | Khumbu | | Bhutan | | Entire domain | |
|---|---|---|---|---|---|---|---|---|---|---|
|  | Area (km$^2$) | % | Area (km$^2$) | % | Area (km$^2$) | % | Area (km$^2$) | % | Area (km$^2$) | % |
| Clean ice | 10.2 | 5.0 | 7.1 | 6.9 | 2.7 | 0.9 | 10.1 | 7.8 | 126.5 | 5.6 |
| Clouds | 3.3 | 1.6 | 2.9 | 2.8 | 0.6 | 0.2 | 7.8 | 6.0 | 45.0 | 2.0 |
| Debris (dark) | 26.6 | 13.1 | 14.9 | 14.6 | 148.1 | 48.9 | 19.5 | 15.0 | 535.4 | 23.8 |
| Debris (light) | 151.4 | 74.4 | 70.1 | 68.6 | 130.2 | 43.0 | 83.1 | 64.1 | 1371.0 | 60.9 |
| Turbid water | 0.6 | 0.3 | 2.7 | 2.6 | 4.9 | 1.6 | 5.2 | 4.0 | 47.0 | 2.1 |
| Vegetation (dry) | 3.3 | 1.6 | 4.5 | 4.4 | 9.6 | 3.2 | 4.1 | 3.1 | 101.7 | 4.5 |
| Unclassified | 8 | 4 | 16 | 16 | 6.9 | 6 | 0.0 | 0.0 | 26.0 | 1.2 |
| Total debris cover | 204 | 100 | 118 | 100 | 303 | 100 | 130 | 100 | 2253 | 100 |

The analysis of pond coverage per 100 m elevation bands over the entire range, however, shows clearer patterns than the glacier-by-glacier results: 77 % of the pond area coverage occurs within 10 % elevation from the glacier termini, and then pond density decreases exponentially towards the upper part of the debris-covered tongues (0.1 % of pond coverage at 75 % elevation upwards from the termini) (Fig. 12a). We note from Fig. 12a that the largest concentration of ponds does not occur directly at the glacier termini but rather within 2 % of the elevation from the terminus, i.e. within 100 m above the minimum elevation. The exponential fit shown in Fig. 12a could have useful predictive power but misses the peak pond coverage that is typically found near the glacier terminus, where ponds coalesce into large terminal lakes.

This implies that the exponential fit is useful for capturing the ponds perched found above the terminus on thicker ice but likely does not capture the water level representing the hydrological base level in the depressions found in thinner ice at the terminus (cf. Benn et al., 2012; Miles et al., 2017b).

The analysis of pond incidence with regards to slope (Fig. 12b) shows that 38 % of the total pond area occurs on 0 to 10° slope bins, with the maximum pond area coverage found at slope bins averaging 4 to 6° (9 % of the pond area). The pond incidence increases initially and then drops on slopes > 8° (Fig. 12b), which is to be expected because at steeper slopes meltwater can drain away (Reynolds, 2000). This is consistent with findings from a previous study (Scherler et al., 2011), which found that slope areas with gradients

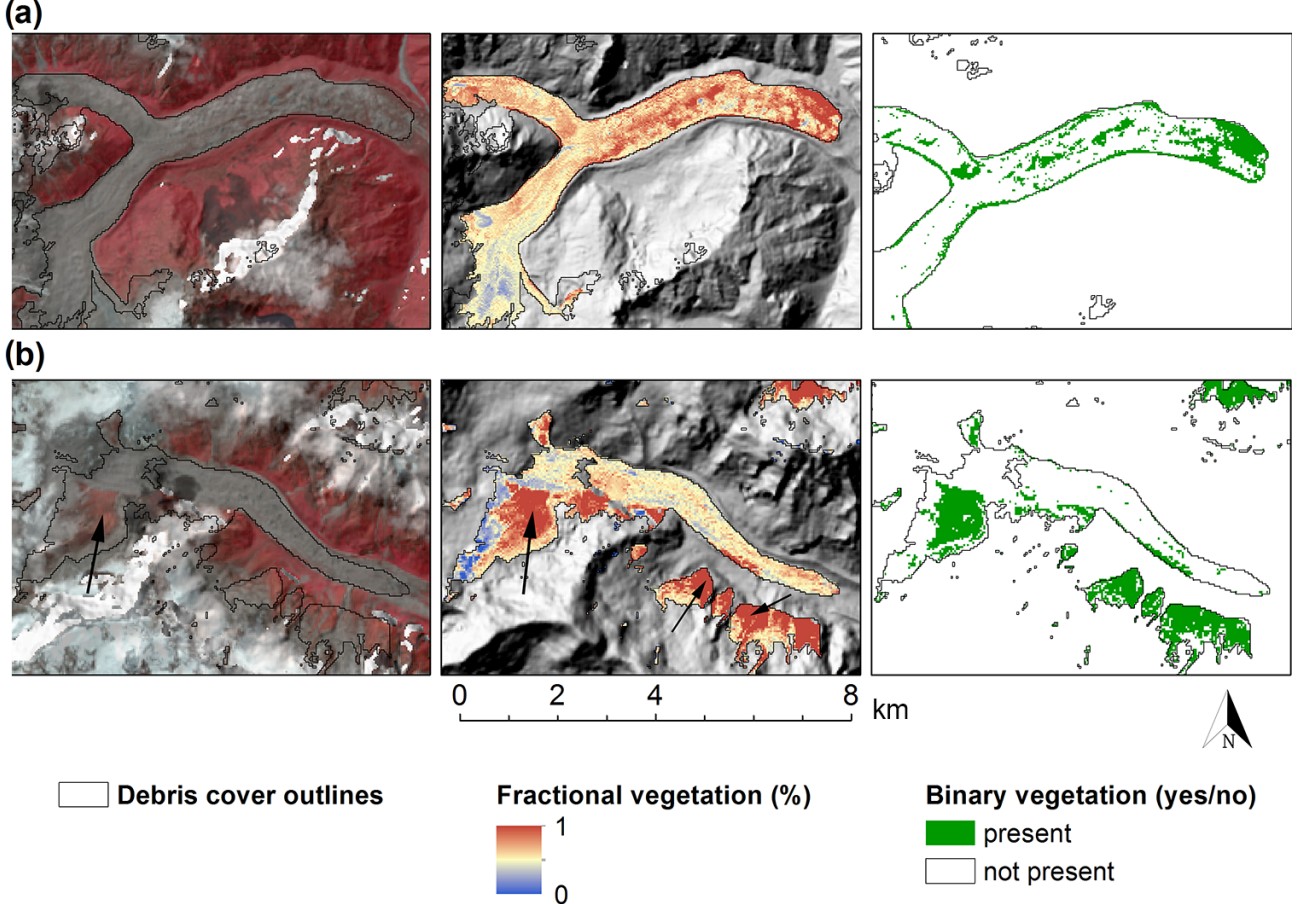

**Figure 10.** Examples of the supraglacial vegetation maps for two glaciers in the eastern Himalaya: **(a)** Kazhenpu Glacier; **(b)** Labeilong Glacier. Left panels show the Landsat 8 OLI colour composite (bands 5, 4, 3) draped onto a shaded relief map from the ALOS DEM. Middle panels show fractional vegetation, and black arrows point to identified errors (bare land and/or healthy vegetation) in the SDC dataset. Right panels show the pixels containing more than 65 % fractional vegetation.

less than 8° were associated with stagnant ice at the terminus regions of debris-covered glaciers over the Himalaya. With respect to glacier aspect, we found that the maximum pond coverage occurs on slopes with an eastern orientation (22.5 to 67.5°, 15.6 % of the pond area) and south-eastern orientation (67.5 %–112.5 % TS23, 14.2 % of the pond area), with less pond incidence (∼ 9 %) on northern-facing slopes (Fig. 12c). Although the differences in pond incidence in the different aspect bands are only within 4 %, this seems to support the fact that southern- and eastern-facing slopes receive more insolation, thus favouring ice melt and formation of ponds.

### 3.7 Supraglacial pond and vegetation distribution over the large domain

Here we present the large-scale patterns of pond and vegetation occurrence on debris-covered glacier tongues over the Himalaya domain with respect to topo-climatic variables averaged and binned at $1 \times 1°$ ($\sim 111$ km) (Fig. 13).

Binned supraglacial ponds and vegetation over the Himalaya domain exhibit clear spatial patterns (Fig. 13a and b). With regards to geographical location, the pond coverage in the western Himalaya is rather homogenous (ranging from 0.1 % to 1.5 % of the debris-covered areas) and is more pronounced and variable in the eastern Himalaya (2.4 % to 4.3 % of the debris-covered area) (Fig. 13a). Pond incidence exhibits a strong positive correlation with longitude ($r = 0.82$, $p < 0.01$) and a strong negative correlation with latitude ($r = -0.72$, $p < 0.01$) (Table 7). Supraglacial vegetation incidence is less pronounced in the north-western part of the domain (Fig. 13b) and increases significantly with longitude ($r = 0.40$, $p < 0.10$) and decreases with latitude ($r = -0.28$, $p < 0.10$) (Table 7). The surface trend analysis of pond and supraglacial vegetation incidence shows that these increase in the east–west direction at the rate of $+0.23$ % and $+0.72$ % per degree longitude, respectively.

Pond occurrence is positively correlated with average temperature ($r = 0.40$, $p < 0.1$) and with precipitation ($r = 0.53$, $p < 0.05$). Furthermore, pond occurrence is negatively

**Table 7.** Correlation matrix for topo-climatic and geographic controls on pond and vegetation coverage based on Pearson's $r$ value. Blue shades represent positive correlations and red shades represent negative correlations. "***" denotes significant correlations at the 99 % confidence level ($p$ value $< 0.01$), "**" denotes significant correlations at the 95 % confidence level ($p$ value $< 0.05$) and "*" denotes significant correlations at the 90 % confidence level ($p$ value $< 0.10$).

| | Ponds | Vegetation | Debris cover % | Termini elevation | Temperature | Precipitation | Thickness change | Velocity | Longitude | Latitude |
|---|---|---|---|---|---|---|---|---|---|---|
| Ponds | 1.00 | | | | | | | | | |
| Vegetation | 0.21 | 1.00 | | | | | | | | |
| Debris cover | 0.05 | -0.25 | 1.00 | | | | | | | |
| Termini elevation | -0.03 | 0.24 | -0.51** | 1.00 | | | | | | |
| Temperature | 0.40** | 0.12 | 0.39 | -0.29 | 1.00 | | | | | |
| Precipitation | 0.53** | 0.29 | 0.32 | -0.53*** | 0.51** | 1.00 | | | | |
| Thickness change | -0.37* | -0.01 | -0.30 | -0.18 | -0.04 | -0.12 | 1.00 | | | |
| Velocity | 0.18 | -0.18 | 0.30 | -0.76*** | 0.21 | 0.54*** | 0.11 | 1.00 | | |
| Longitude | 0.82*** | 0.40* | 0.27 | -0.07 | 0.42** | 0.74*** | -0.42** | 0.24 | 1.00 | |
| Latitude | -0.72*** | -0.28* | -0.47** | 0.32 | -0.73*** | -0.78*** | 0.25 | -0.37* | -0.88*** | 1.00 |

**(a)**

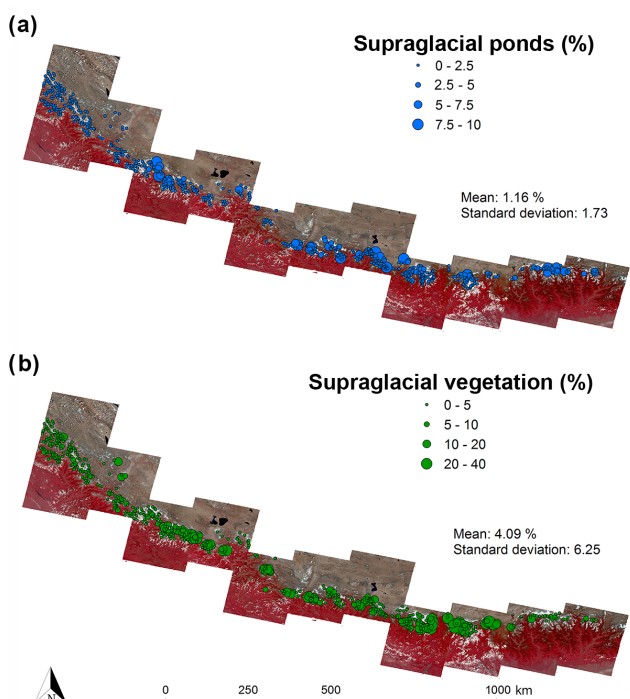

**Supraglacial ponds (%)**
- 0 - 2.5
- 2.5 - 5
- 5 - 7.5
- 7.5 - 10

Mean: 1.16 %
Standard deviation: 1.73

**(b)**

**Supraglacial vegetation (%)**
- 0 - 5
- 5 - 10
- 10 - 20
- 20 - 40

Mean: 4.09 %
Standard deviation: 6.25

N

0    250    500      1000 km

**Figure 11.** Distribution of **(a)** supraglacial pond coverage and **(b)** supraglacial vegetation, expressed as percent of each debris-covered area on a glacier-by-glacier basis for the 408 sampled glaciers.

correlated with glacier thickness change ($r = -0.37$, $p < 0.10$) (Table 7). We did not find significant correlations of pond and vegetation occurrence with supraglacial debris cover, glacier termini elevation or average glacier velocity

(Table 7). Supraglacial vegetation had a weak non-significant positive correlation with precipitation and termini elevation.

## 4 Discussion

### 4.1 Controls on mountain-range-scale supraglacial pond and vegetation distribution

The topo-climatic conditions for the occurrence of supraglacial ponds on the surface of debris-covered glaciers have been addressed in previous studies (e.g. Sakai, 2012; Sakai and Fujita, 2010), but supraglacial vegetation and its controls have rarely been addressed. Previous studies showed that both ponds and vegetation tend to develop on stagnant, low-angle slopes of the debris-covered tongues (Sakai and Fujita, 2010; Reynolds, 2000; Quincey et al., 2007). Furthermore, we would expect to find more ponds and lakes on debris-covered glacier tongue glaciers CE11 situated at lower elevations, which experience increased temperature and therefore enhanced surface melt. However, our analysis on a glacier-by-glacier did not yield significant spatial trends in pond and vegetation occurrence with respect to these controls. This implies that at the mountain-range scale the distribution of supraglacial features may be governed by more complex factors, such as geomorphologic, glaciologic and climatic patterns. Here we discuss the occurrence of supraglacial ponds and vegetation in light of regional topo-climatic conditions averaged over $1 \times 1°$ cells.

A first observation is that supraglacial debris covers a larger part of the glacierized areas in the central and eastern Himalaya compared to the western extremities (Fig. 13c). Supraglacial debris decreases linearly from the south-west to north-east, and it is significantly correlated with latitude

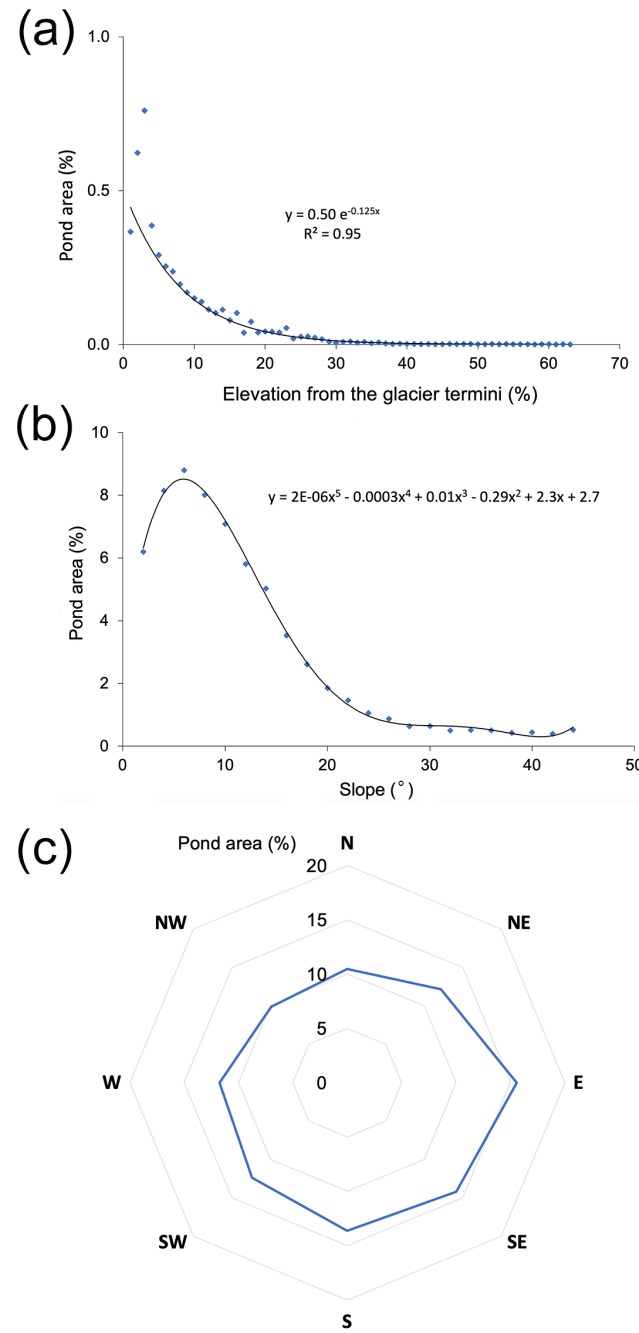

**Figure 12.** Plots of supraglacial pond coverage summarized over **(a)** elevation bands expressed as percent above terminus, **(b)** slope expressed as 2° bins and **(c)** glacier aspect expressed as 45° bins.

($p = 0.47$, $p < 0.05$). At the same time, the elevation of the debris-covered glacier termini increases northwards towards the Tibetan Plateau ($+354$ m per degree latitude) and to a lesser extent from west to east ($+114$ m per degree longitude) (Fig. 13d, Table 7). The increasing trends in both pond and vegetation coverage towards the eastern Himalaya noted earlier (Fig. 13a, b) are consistent with the presence of lower

glacier termini and higher rates of debris in the eastern part compared to the western part. Overall, debris-covered glacier tongues descend to lower elevations in the central and eastern Himalaya regions ($\sim 3700$ to $4400$ m) compared to the western part ($\sim 4700$ to $4900$ m). Our results show that glacier termini elevation exhibits only a very weak negative control on pond occurrence (Table 7) and a slightly larger but not significant control on vegetation coverage. It appears that the elevations at which supraglacial debris is found do not significantly influence pond occurrence vegetation growth on these tongues.

Development of supraglacial vegetation (mostly shrubs) has been noted on stagnant, thick debris-covered tongues in various areas of the world (Xie et al., 2020; Tampucci et al., 2016). Increasing trends in supraglacial vegetation in other glacierized areas such as the Alps are a consequence of climatic change (Vezzola et al., 2016). As supraglacial vegetation typically only develops on stagnant surfaces that are no longer undergoing substantial gravitational reworking, its presence may also constitute an indication of glacier inactivity and later stages of decay. The increased vegetation occurrence towards the eastern Himalaya (Fig. 13b) observed in this study coincides with a clear west-to-east pattern in negative glacier surface elevation changes based on Shean's et al. (2020) dataset (Fig. 13e). Glacier surface changes become increasingly more negative towards the east at the rate of 0.02 m per degree longitude. Spatial patterns in glacier surface thinning and the resulting mass balance is consistent with the eastward increase in both pond and vegetation incidence observed in this study. In this study, however, the direct dependence of supraglacial vegetation on glacier thinning patterns is rather weak (Table 7).

In addition, the eastward decrease in average glacier velocities ($-0.2$ m a$^{-1}$ per degree longitude and $-0.1$ m a$^{-1}$ per degree latitude), based on the trend analysis of 2013–2015 datasets from Dehecq et al. (2015) (Fig. 13f), shows a tendency for stagnating debris-covered glacier tongues towards the north and towards the east. Stagnant glaciers were reported for the northern parts of the central Himalaya (Scherler et al., 2011) and were attributed to topographic differences, i.e. low slope angles on the northern slopes of the range promoting development of stagnant ice. Such patterns are in contrast with more rugged, steeper terrain of the southern slopes, which favours more dynamic glacier environments (Scherler et al., 2011). The stagnating trends coupled with a higher percentage of supraglacial debris correlate with the higher incidence of vegetation towards the east (Fig. 13b), supporting the idea that debris cover of sufficient stability favours plant colonization (Fickert et al., 2007). Such patterns point to a potential transition of debris-covered glaciers in certain areas towards vegetated glaciers as noted in other studies (Fickert et al., 2007). It has been noted in recent studies that supraglacial ponds can enhance local ablation rates by up to 3 times (Brun et al., 2016; Miles et al., 2018; Irvine-Fynn et al., 2017). The slightly more negative

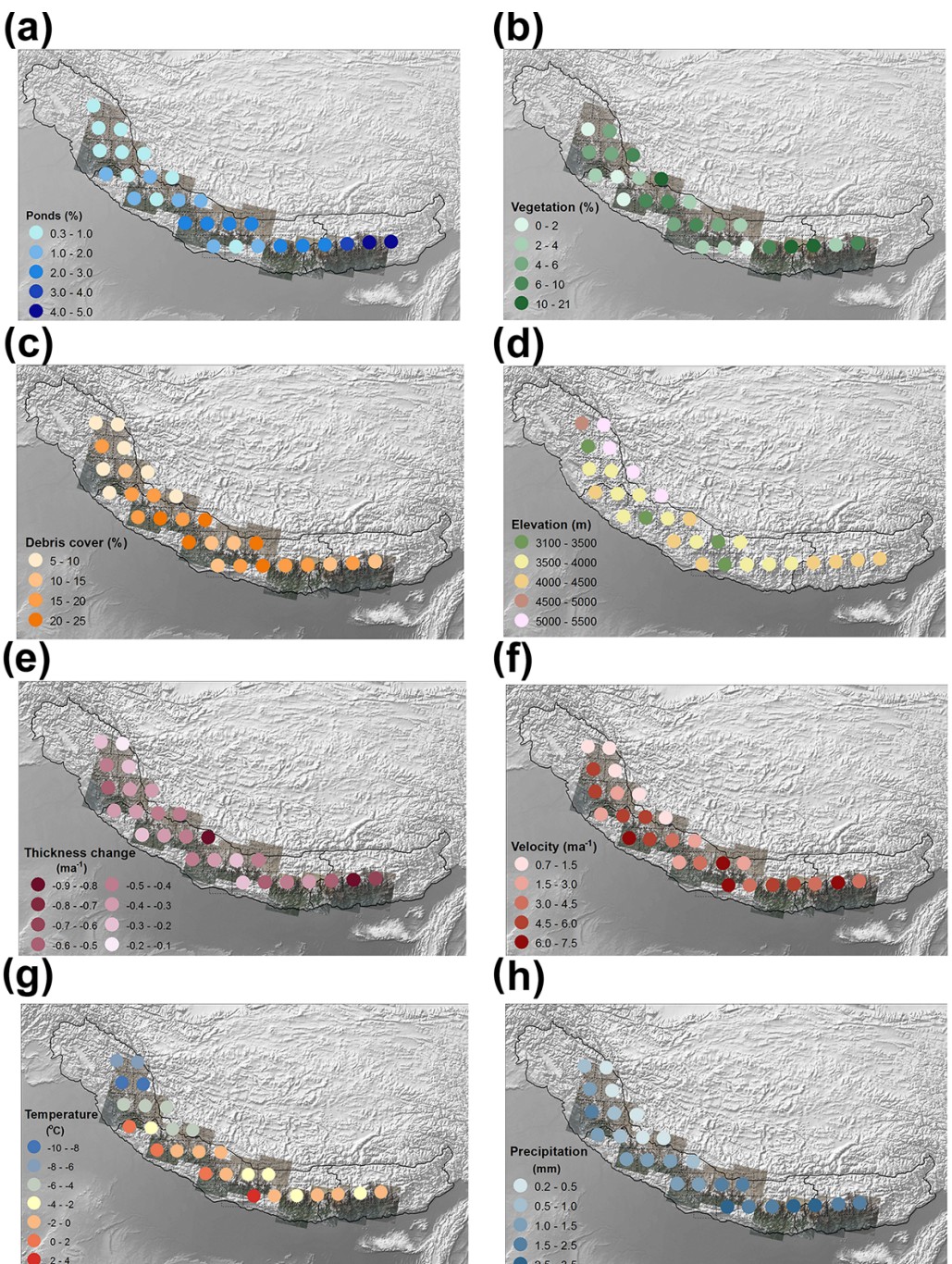

**Figure 13.** Plots of **(a)** LMM-derived ponds, **(b)** LMM-derived vegetation, **(c)** debris cover expressed as percent of the glacierized area, **(d)** minimum elevation of debris cover, **(e)** average temperature from ERA5-Land (October 2015), **(f)** total precipitation from ERA5-Land (October 2015), **(g)** thickness change trends 2000–2018 from Shean et al. (2020) and **(f)** average velocity trends 2013–2015 from Dehecq et al. (2015). All variables were averaged over the glacierized areas and gridded over $1 \times 1°$ grid cells.

mass balances and lower surface velocities towards the east may indicate the transition of debris-covered glaciers to inactive debris-covered glacier tongue or a rock glacier in this part of the Himalaya (Jones et al., 2019 TS24; Monnier and Kinnard, 2017).

Climate factors (i.e. higher temperatures and precipitation) induce more dynamic environments and could favour increased surface melt and pond formation (Herreid and Pellicciotti, 2020). In the case of the Himalaya, we observed a significant south-west-to-north-east decreasing trend in gridded

average temperatures, with a stronger decrease in the south-to-north direction ($-2.3\,°C$ per degree latitude) (Fig. 13g and Table 7). Total gridded precipitation for the same month increases in the eastern direction at the rate of $-0.06$ mm per degree latitude and significantly decreases towards the drier, colder regions of the Tibetan Plateau with a stronger gradient northward ($-0.23$ mm per degree latitude) (Fig. 13h and Table 7). On the contrary, the warmer and wetter areas of the eastern Himalaya seem to favour higher pond coverage, as also suggested in other studies (Herreid and Pellicciotti, 2020). At larger scales, it has been shown that certain conditions related to topography and lithology could offset this dependency, but at the range of the Himalaya, this climatic dependency holds. Climatic conditions and glacier characteristics in the western Himalaya are more similar to those in the Karakoram, where glaciers have undergone less shrinkage (Brun et al., 2017; Kääb et al., 2012; Gardelle et al., 2013), than those in the central and eastern, monsoon-influenced parts of the Himalaya, which exhibit higher temperatures and larger precipitation amounts.

The controls on debris-covered glacier surface evolution are a complex combination of the cumulative debris-supply, mass-balance condition, debris cover expansion, stagnation and total lowering. Studies have noted that surface types are related to the evolutionary stage of a debris-covered glacier (cf. Thompson et al., 2016), in that debris thickness variability, local topography, degree of downwasting, and glacier tongue slope are all potentially, at least partially, related to the time lapsed since debris cover formation (Sakai and Fujita, 2010; Nicholson et al., 2018). Relatedly, Herreid and Pellicciotti (2020) introduce the term of debris-covered glacier stage ranging from 0 to 1 as a percentage of the full, 2D debris cover carrying capacity of a glacier, such that if 100 % of the ablation zone is debris-covered, then the debris-covered area cannot expand further without up-glacier migration of the equilibrium line. Further analysis is needed to accurately capture the complex combination of topographic and climatic factors that contribute to the development of ponds and vegetation on supraglacial debris cover using the most recent publicly available and corrected datasets (Herreid and Pellicciotti, 2020) and a carefully quality-controlled output of the method proposed here. A full understanding of the occurrence of surface features also requires knowledge of the transient co-evolution of glacier extents and debris cover. This would allow us to quantify better the controls on pond formation and vegetation growth in specific catchments as the debris cover and glacier geometry develop over time.

## 4.2 Spatial and spectral limitations of the Landsat data

Our analysis of surface composition of the debris-covered glacier tongues is subject to several limitations related to the spectral and spatial resolution of the input Landsat data. While linear spectral unmixing is a relatively straightforward routine to implement once the endmembers and their spectra are selected, using Landsat data at 30 m spatial resolution and spectral dimensionality for spectral unmixing has its limitations. While Landsat 8 is superior to the previous Landsat missions in terms of its calibration, geometry and radiometric resolution (Irons et al., 2012), its spectral dimensionality remains an issue, particularly with respect to mapping of the various types of debris material and/or supraglacial ponds with various degrees of turbidity. Previous studies in the Himalaya (Casey and Kääb, 2012; Casey et al., 2012; Matta et al., 2017) suggest that the spectral dimensionality of these two surfaces is greater than the dimensionality of the Landsat 8 OLI bands available for unmixing. Landsat has limited spectral resolution data (7 bands available for unmixing) compared to hyperspectral data (e.g. AVIRIS, 224 bands). Both the partially constrained and the unconstrained LMMs yielded negative abundances in our study, with larger positive values ($> 3$) especially for the vegetation class. Since our fractions did not satisfy the sum-to-unity condition, normalization of the classes was necessary, which may have introduced further uncertainty in our results because some classes had higher positive values than others. However, previous studies showed that these negative values do not necessarily affect the ability to discriminate between surfaces (Klein and Isacks, 1999).

Limitations posed by the spatial resolution of Landsat data (30 m) affected the accuracy of the selected endmembers. While we used the pixel purity index to automate the selection of endmembers, we acknowledge that some mixture may still occur at 30 m spatial resolution. Furthermore, the 30 m spatial resolution does not allow us to detect supraglacial features such as ice cliffs or small ponds which can span only a few square metres. Improvements envisioned here include applying the spectral unmixing Sentinel-2 imagery, which has a better spectral, spatial and temporal resolution (13 bands in the visible to short-wave infrared, 10–20 m, 5 d revisit time) compared to Landsat (7 bands in the visible to short wave, 30 m, 16 d revisit time). This would allow for better definition of endmembers, facilitating more accurate and repeated mapping in the future.

Furthermore, the 30 m spatial resolution of the DEM does not allow us to infer the precise control of topographic factors such as slope and aspect on pond formation or a full quantification of the small-scale controls of pond incidence but only provides a mountain-range scale of the pond distribution.

## 4.3 Limitations in the endmember definition

In this study, we utilized the maximum number of endmembers ($n = 6$) allowed by the spectral resolution of the Landsat 8 OLI data (7 bands), in an attempt to capture the variability of the system and to avoid high RMSE of the model which may occur due to missing classes. The main difficulty here consisted in capturing the wide variability of the materials present across the mountain range, for example different lithologies, while ensuring a "valid" LMM. This is de-

fined as one where fractional values do not exceed 1.01 (under strict constraint rules) or 2.01 (under looser rules) and where RMSE is less than 2.5 % (Painter et al., 2009). Our choice of debris endmembers was limited to light and dark debris, and these may not cover the wide spectrum of lithology present across the Himalaya. With regards to the on-the-ground spectral characteristics of the debris material in the Khumbu region, Casey et al. (2012) showed that these vary due to the presence of various of minerals, notably distinct granitic (lighter) vs. schistic (darker) debris types with different compositions. However, spectral differences in these two classes can also be related to debris water content especially on very thin debris (as for thinly-debris-covered ice cliffs) and are associated with grain size, i.e. fine-grained sediments have a greater capacity for water retention (Juen et al., 2013; Collier et al., 2014). We also noted such differences in the spectra for wet fine debris and dry coarse debris with large grain sizes on Mer de Glace (Fig. 4b); however, the limited Landsat spectral resolution implies that we could not define separate endmembers for each. Using only two endmembers for debris cannot capture the various types of debris with different mineral and geochemical composition, nor can it distinguish between debris with various degrees of water content, which has a different spectral signature compared to dry debris (Fig. 4b). Furthermore, we could not take into consideration bare illuminated non-glacierized surfaces including nunataks, which were occasionally mistakenly included within the polygons in the SDC dataset. As a result, these areas were also associated with some high positive fractional values, which might have affected the overall RMSE of our model and particularly the sum-to-unity condition.

Although we defined the water endmember on the basis of turbid water (greyish-blue ponds), supraglacial ponds of various turbidity levels are present across the mountain range, due to various degrees of suspended sediments. The colour of these ponds can range from grey to turquoise and reddish shades in various proportions (Matta et al., 2017) to small clear water supraglacial ponds (Takeuchi et al., 2012; Giardino et al., 2010), as observed in the field (Fig. 3e). Each type of pond has different spectral signatures, but the limited spectral resolution of Landsat does not allow us to use concomitantly both a clear and a variable turbid water endmember in the spectral unmixing. Nevertheless, as shown in Fig. 7, the majority of the turbid supraglacial ponds are connected to the exposed ice and glacier drainage network (hence larger suspended sediment), are expected to be most relevant to glacier evolution and may be of concern for outburst flood potential. Our clear water algorithm nicely picks out the small number of isolated non-turbid ponds at the terminus of the Ngozumpa Glacier (Fig. 7), highlighting the success of different endmember selection for addressing other scientific questions. With further testing, fractional water maps obtained from spectral unmixing techniques can be used to characterize the state of lakes and ponds in terms of their turbidity (Matta et al., 2017; Giardino et al., 2010), i.e.

by quantifying the fraction of a pixel covered by water, light and/or dark debris. In this regard, repeated monitoring of pond turbidity using these combined tools allows changes in suspended sediment load to be tracked over time, which are considered direct indicators of glacier wasting processes and glacier–lake interaction (Giardino et al., 2010). This aspect is not fully explored in this study but can be further investigated by combining LMMs with field spectra of ponds and lakes to characterize the various degrees of turbidity across the mountain range. Since lake turbidity is temporally highly variable and since our current dataset is a snapshot of pond density, it cannot be used to infer any variability in sediment concentration, but it provided the basis for tracking changes in glacier area, which has important applications.

Similarly, we could not define a healthy vegetation endmember whose spectral signature (not shown here) differs from that of the dry vegetation endmember we selected. However, small amounts of healthy vegetation do occur on debris-covered glaciers in the eastern part of the Khumbu domain, and these were indeed detected by the LMM (Fig. 10).

The cloud and clean ice detection based on LMM was not accurate in this particular configuration. While some isolated pixels were classified as clouds, others pixels were confounded with other types of surfaces, notably debris (Table 2). While the cloud distribution noted in this study corresponds to local meteorology, i.e. more frequent cloud cover in the eastern Himalaya until later fall months compared to the western part (Thayyen and Gergan, 2010), we are less confident in the actual estimations of the cloud cover areas, so we do not wish to over-interpret these. Applying algorithms such as Fmask (Zhu et al., 2015) to mask the clouds resulted in misclassification of the entire glacierized surface as cloud, which is a well-documented issue (Stillinger et al., 2019), so we could not mask the clouds prior to the spectral unmixing.

Likewise, clean ice was poorly classified, mostly likely due to its poor representation in the dataset (i.e. limited number of clean ice pure pixels). While our results hint at the presence of ice to some extent, we are not confident about these results. Some pixels correspond indeed to location of ice cliffs which were perhaps exposed at the end of the ablation season (cf. Fig. 9); others correspond mostly to clean ice patches at the upper limit of supraglacial debris which were included in the input data, which dated from previous years, or seasonal snow. While we chose our images at the end of the ablation season, post-monsoon the snow cover is usually minimal, but early snowfalls can occur. Other features such as the ice sails (Evatt et al., 2017) may not be extracted at the spatial resolution of the Landsat imagery, since these features often span only several square metres. At the same time, the LMM algorithm in its current parameterization cannot detect ice cliffs dusted with fine debris, which have a lower albedo than clean ice (Naegeli et al., 2015). Targeting exposed but dusted ice features within the debris cover in addition to clean ice would need some refinement of

the algorithm using Sentinel-2 imagery with better spectral resolution and better parameterization (Kneib et al., 2020), optical thresholding of band ratios using high-resolution imagery (Anderson et al., 2021) and/or feature detection based on OBIA (Kraaijenbrink et al., 2016; Watson et al., 2017a; Mölg et al., 2019).

## 4.4 Uncertainty due to the thresholds applied to fractional maps

Selection of the thresholds used to classify the fractional maps to obtain the final maps of each surface is another source of uncertainty in our method. Previous spectral unmixing studies (Hall, 2002; Rittger et al., 2013) justified using a threshold of 0.5 for the classifying fractional maps for various types of surfaces, while they also tested thresholds as low as 0.15 (Rittger et al., 2013). While we applied a threshold of 0.5 and 0.65 to our water and vegetation classes, respectively, for the other classes the fractional thresholds were ultimately determined using visual inspection, which introduced a certain degree of subjectivity into our study.

## 4.5 Quality of input SDC dataset

Due to the spectral limitations of Landsat, in this study we applied the unmixing only to the debris-covered areas of glaciers to reduce model complexity. Therefore, model performance is to some extent subject to the quality of the input dataset. At the onset of our study, the only global database of supraglacial debris was the SDC dataset (Scherler et al., 2018), and although Herreid and Pellicciotti (2020) provide updated supraglacial debris outlines, these were not available at the onset of our study and are not currently incorporated in the standardized RGI dataset. Debris outlines in the SDC dataset constitute a multi-time stamp dataset, based on data spanning 1998 to 2015, while our Landsat data were based primarily on 2015. This may introduce uncertainties in the calculation of pond coverage. For example, we assumed that any changes at the termini of the debris-covered areas would have occurred within these older outlines, since surge-type glaciers and hence apparent glacier advance are rare or nonexistent in the Himalaya region, contrary to the Karakoram (Sevestre and Benn, 2015). However, recent studies have reported an upward expansion of the debris cover in the Himalaya (Xie et al., 2020; Thakuri et al., 2014; Bhambri et al., 2011b; Kamp et al., 2011), which we do not account for here. As such, in these areas, our pond density may be underestimated, and this would need a more in-depth analysis and the availability of multi-temporal supraglacial debris datasets. Furthermore, our study revealed some important issues with the input SDC dataset used to constrain the spectral unmixing, particularly the inclusion of patches of healthy vegetation and bare bright steep terrain. The spurious vegetated areas present within the debris cover outlines (Fig. 10b) may have affected to some extent the quality of the spectral

unmixing, i.e. the non-negativity and the sum-to-unity conditions, because it produced large negative and positive fractional vegetation values. We were able to identify theses as being healthy vegetation on non-glacierized terrain. On the other hand, some of the high percentage of supraglacial vegetation in some of the eastern parts is attributed to errors in the input supraglacial dataset, and we are hesitant to overinterpret the vegetation analysis. However, we note the potential of the fractional vegetation maps for identifying mapping errors in the SDC dataset. Because these abnormal values served to identify errors in the existing SDC dataset, they constitute a valuable tool to correct and refine these global databases.

## 4.6 Wider applicability of the method

In this study we demonstrated the transferability of a method developed on a single region for the year 2015 (Khumbu) by applying it to a Landsat 8 OLI scene from a different area (Lahaul–Spiti) for the same season (post-monsoonal) but a different year (2016) and validating the ponds with PlanetScope data. In the light of the spatial and spectral limitations of Landsat data discussed above, the applicability of our approach for multitemporal analyses requires careful considerations. When transferring methods from one scene to others, illumination differences and shadow effects across the scenes need to be resolved, particularly if the scenes are not acquired on the same date. In this study, we attempted to minimize these effects by applying atmospheric and topographic corrections and implicitly assumed that the set of endmembers defined for the Khumbu could be applied to the entire Himalaya. However, in some areas, some spectral differences may remain, leading to confusion between the water/light debris/ice classes and hence some overestimation of the pond coverage, particularly in some areas of the western Himalaya. While these pond areas require further quality control prior to their inclusion in regional datasets, they are within the uncertainties reported at other sites, for example Langtang (Steiner et al., 2019). Furthermore, if the approach is used over the same area for multi-temporal pond or vegetation analysis, the geolocation accuracy of the Landsat can be a concern, because the pixels can be slightly misaligned from acquisition to acquisition, resulting in potentially very different compositions and unmixing results. This needs to be mitigated by co-registration of the scenes prior to unmixing and performing the change analysis. Further uncertainty is introduced in our study by the fact that for certain areas of the Himalaya, Landsat cloud-free and snow-free scenes were not available for the year 2015, and we used scenes from 2014 and 2016 (cf. Table 1). We assumed that surface conditions were similar but acknowledge that pond areas are dynamic and can change from year to year. Furthermore, this study does not account for the seasonality of supraglacial ponds but provides a methodological basis for their identification.

## 5    Summary and further work

In this study, we estimated the spatial distribution of surface characteristics on debris-covered glaciers (various types of debris, clean ice, supraglacial ponds and vegetation) at the subpixel scale using 30 m fractional maps obtained from a spectral linear mixing model. We tested the approach in the Khumbu region comprising eastern Nepal and parts of China using Landsat 8 OLI imagery and then applied it over the entire Himalaya to evaluate its performance over a larger domain. Pléiades and Planet high-resolution imagery was used to assess the endmember selection and to validate the mapped supraglacial pond areas using OBIA techniques. Our key findings can be summarized as follows.

- We demonstrate the use of Landsat spectral unmixing in determining the surface properties of debris-covered glaciers, which holds great potential for mapping the dynamic changes in surface conditions at a regional scale. While we present a method that holds promise for effectively partitioning the surface properties of debris-covered glaciers, we recommend that future analysis of the potential drivers and controls on the observed surface types and their regional variation revealed by this method be carried out on a further-quality-controlled dataset to avoid over-interpretation of any errors within the datasets used.

- We show that spectral signatures derived from the Landsat 8 OLI imagery and cross-checked using high-resolution Pléiades images can be applied at the mountain-range scale provided that all images are atmospherically and topographically corrected to reduce differences in illumination patterns and that images are acquired around the same date. While the limited Landsat spectral resolution did not allow for a very fine definition of the wide spectrum of all the different debris lithologies and ice types present on debris-covered tongues across the study area, LMM successfully distinguished among broad categories and convincingly reproduced independently mapped supraglacial pond areas. Overall, we consider the spectral unmixing method presented here a promising approach to add to the suite of tools that are valuable in analysing the dynamic surfaces of debris-covered glaciers.

- One of the major contributions of the current study is that we produced a supraglacial pond inventory for the entire Himalaya for the year 2015, based on spectral unmixing of coarse-resolution and freely available Landsat 8 OLI satellite imagery. We consider that this approach can provide more detail and thus outperform other analyses of supraglacial pond identification and classification performed on similar Landsat data for the same period but based on normalized difference water indices (Shugar et al., 2020) or

manual delineation (Chen et al., 2021). The method and results are comparable to mapping quality from higher resolution, allowing improved analysis of multitemporal change in pond incidence and size in a future study. The dataset of supraglacial ponds is available in the public domain via the Zenodo data repository (https://doi.org/10.5281/zenodo.4421857).

- Regional trend analysis of gridded data indicates that higher average temperatures and more abundant precipitation have a strong influence on pond development and to a much lesser extent on supraglacial vegetation occurrence. Higher glacier thinning rates coupled with lower average glacier velocities are consistent with pond incidence and seem to favour the development of supraglacial vegetation. The extent of the supraglacial debris and the elevation of the termini exhibit a weak control on supraglacial pond coverage and a moderate control on supraglacial vegetation.

Future developments to overcome the current limitations of this study include the use of more sophisticated non-linear mixing models, which would allow us to discriminate materials of interest in more detail. Work is ongoing to make the unmixing step approach fully automated by integrating it within scripting routines (Bunting et al., 2014), so that it can be applied in the future to derive supraglacial pond outlines at multi-temporal scales and monitor pond development over time. Given that these surface ponds are ephemeral and change rapidly, automated multi-temporal-scale mapping is highly desirable to track their evolution over time in various regions. The analysis presented here complements and expands the existing proglacial lake databases for the year 2015 by providing supraglacial pond extents. With continued advances in satellite data in the near future, the methodology developed here provides avenues towards achieving large-scale, repeated mapping of supraglacial features.

*Code availability.* Atmospheric and topographic corrections were performed using the ARCSI routine, embedded in the freely available, python-based RSGISLib software available freely (Bunting et al., 2014). The code for batch processing of the Landsat 8 OLI images for the entire Himalaya can be provided upon request. Post-processing of the spectrally unmixed Landsat 8 OLI maps was done using the Python module ArcPy from ESRI ArcGIS. The steps for loop processing (normalizing the fractional raster files, classifying the surfaces and extracting the composition of the debris-covered glaciers from the fractional maps) can be provided upon request.

*Data availability.* Landsat 8 OLI data used in this study can be obtained at no cost from the USGS EarthExplorer (https://earthexplorer.usgs.gov/, last access: TS25). All versions of the NASA SRTM Global 1 arc second DEMs are available from the Earthdata platform (https://earthdata.nasa.gov/, last access: TS26). All versions of the ALOS Global Digital Surface Model, in-

cluding the one used in this paper, are available from https://www.eorc.jaxa.jp/ALOS/en/aw3d30/index.htm (last access: TS27). Datasets of supraglacial ponds and vegetation, along with the fractional maps, are available via the Zenodo data repository (https://doi.org/10.5281/zenodo.4421857, Racoviteanu et al., 2021).

*Supplement.* The supplement related to this article is available online at: https://doi.org/10.5194/tc-15-1-2021-supplement.

*Author contributions.* AER conceived the idea, designed the spectral unmixing experiments, led this work, obtained and processed the Landsat and the high-resolution images, and wrote the paper with input from co-authors. LN provided the Pléiades imagery, discussed the research strategy and helped select endmembers based on field expertise. NFG supervised the study and provided geomorphology expertise. All authors contributed to writing the paper.

*Competing interests.* The authors declare that they have no conflict of interest.

*Acknowledgements.* This research received funding from the European Union's Horizon 2020 research and innovation programme under the Marie Skłodowska-Curie grant agreement no. 663830. Lindsey Nicholson was supported by the Austrian Science Fund (FWF) grant P28521. Access to Pléiades imagery was provided through the Österreichische Forschungsförderungsgesellschaft (FFG) project "High-resolution spaceborne studies of mass balance processes on glaciers of the Khumbu Himal, Nepal" (GlHima-Sat). We acknowledge the BritInn Fellowship Programme which funded Adina E. Racoviteanu's work visit to the University of Innsbruck to develop this research with Lindsey Nicholson in 2018. We are grateful to the United States Geological Survey and to the Planet API programme for providing free access to Landsat and RapidEye imagery. We thank Lorenzo Rieg and Christoph Klug of the University of Innsbruck for processing of the Pléiades DEMs.

*Financial support.* This research has been supported by the NAME OF FUNDER (grant no. GRANT AGREEMENT NO). TS28

*Review statement.* This paper was edited by Daniel Farinotti and reviewed by Marin Kneib and one anonymous referee.

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

## Remarks from the language copy-editor

## Remarks from the typesetter

TS29 Please ensure that any data sets and software codes used in this work are properly cited in the text and included in this reference list. Thereby, please keep our reference style in mind, including creators, titles, publisher/repository, persistent identifier, and publication year. Regarding the publisher/repository, please add "[data set]" or "[code]" to the entry (e.g. Zenodo [code]).

TS30 Please provide the page range or article number.

TS31 Please provide the page range or article number.

TS32 Please note that the TC update has been inserted.

TS33 Please provide the page range or article number.

TS34 Please note that the ESSD update has been inserted.

TS35 Please provide the page range or article number.

TS36 Please provide the page range or article number.

TS37 Please check DOI.

TS38 Hall et al. (2002) is not mentioned in the text.

TS39 Please check the page range.

TS40 Please provide place of publication.

TS41 Please provide the volume.

TS42 Please provide the page range or article number.

TS43 Please check the page range.

TS44 Please provide the page range or article number.

TS45 Please provide the volume.

TS46 Painter et al. (2003) is not mentioned in the text.

TS47 Please provide the page range or article number.

TS48 Please provide all editor names.

TS49 Please provide place of publication.

TS50 Please provide the page range or article number.

TS51 Please confirm.

TS52 Please note that the TC update has been inserted.

TS53 Please provide the last name.

TS54 Please provide the page range or article number.

TS55 Please provide the volume.

TS56 Please check the page range.

TS57 Please provide the page range.

TS58 Please provide the page range.

TS59 Please provide the page range or article number.

TS60 Please provide the volume and page range.

TS61 Please provide the page range or article number.

TS62 Please check name.

TS63 Please provide publisher and place of publication.