# Peer review of "Surface composition of debris-covered glaciers across the Himalaya using linear spectral unmixing of Landsat 8 OLI imagery"

_The Cryosphere, 2020_

## Author Response (AR1)

Dear Reviwers,

Please find below our full replies to Reviewers 1 and 2. Please note that Reviewer1 comments were based on the original submission (which had not undergone changes after Editor comments), and Reviewer 2 comments were based on the second corrected version we submitted after the comments from the Editor. Therefore, the line numbers referred to in each of the reviewer comments differed slightly.

We thank both Reviewers for their detailed reviews and excellent suggestions, all of which we addressed in the revised manuscript. Please find below our replies to Reviewers, in bold text.

RESPONSE TO REVIEWER 1

**Thank you again for the thorough review, and for raising pertinent points related to the methodology used, their transferability and the validation of the results. We have addressed all the 5 main points as well as the detailed minor comments in the revised manuscript. Also, we have taken into account the suggestions for improving the clarity of the writing and the representation of the existing literature. The manuscript has undergone a substantial review and multiple re-writing, so some of the paragraphs have changed considerably. We have marked this in the line-by-line comments\*.**

**General comments**

**RC: Choice of endmembers and validation:** my understanding from the manuscript is that the Pléiades (and RapidEye) images used here were used as a quality control of the surfacetype of some Landsat pixels. For this to work, you would need to make sure that the Landsat pixels are composed only of one surface type, or at least quantify the different surface types. This can be done fairly easily with some manual delineation from the Pléiades images, however one thing to be careful with here is the coregistration of the Pléiades and Landsat 8. Somewhere in the text is mentioned that the position of Landsat 8 is accurate within 50 m, which may translate to significant surface type changes, i.e. the surface characterized using the Pléiades image may not be the same as the Landsat one if the images are not correctly aligned. Proper alignment of Pléiades/RapidEye and Landsat (using cross correlation techniques for instance) should be ensured and demonstrated in the paper.

**AC: We appreciate and fully agree with this concern about the quality control of the surface types. We have checked for the x/y shifts in the Pleiades and RapidEye data with respect to Landsat before co-registration. These were minimal (< 1 m), but we have however co-registered these images using COSI-Corr, and reported the RMSE in the revised manuscript (section 2.2)**

**With regards to the stated Landsat positional accuracy of 50 m, this is a global measure and does not come from our assessment. Since this may not apply here, it was removed.**

**With regards to the comment "Landsat pixels are composed only of one surface type", this is of course one of the limitations of using Landsat. We point out in the revised manuscript that there may still be a degree of "mixture" in the pure pixels due to the resolution of the data, and we will acknowledge this as a source of uncertainty in the revised manuscript.** We **clarify that spectral signatures were not extracted from the Pleiades image but directly from Landsat using the pixel purity index routine in ENVI. We now explain more thoroughly this routine in the revised manuscript (section 2.5.1)**

RC: Focus regions: the method is applied to the entirety of the Himalaya, with a focus on three regions meant to represent the climate variability across the domain. The Lahaul Spiti is very far to the West and the Khumbu and Bhutan are relatively close, in the eastern part of the range. To really represent the climatic gradient as described in the manuscript, I strongly recommend adding one or two study regions in between Khumbu and Lahaul Spiti domains.

**It is true that the distribution of the regions was not even. This is because we sampled the known climatic regions, i.e., from west to east: a) the monsoon transition zone; b) the central Himalaya and c) the eastern extremity of the Himalaya (these are now shown and labelled in Fig 1). In the revised manuscript we have now added another region in the central Himalaya. We have also added gridded climate data to our analysis, to support our choice of climatic regions. However, we estimate that because of the location of these additional scenes in the central Himalaya, any differences in the composition of the debris-covered tongues might be due to interregional variability in lithology, for example.**

RC: Generalization of the method to all of the Himalaya: the method was calibrated and validated for one specific location of the Himalaya and only for one Landsat 8 image. Further checks are needed to demonstrate the transferability of the method to the whole Himalayan range. I recommend the authors to validate the surface composition maps they obtain for at least 1 (better 2 or more) other site and Landsat 8 image. Given the free availability of Rapid Eye imagery using a Planet academic licence, this would be easily achievable.

**In the revised manuscript ,we have now applied and validated the method for a different site / Landsat 8 scene- the Lahaul Spiti in the monsoon transition zone, for an additional year (2016) for which we obtained high quality Planetscope data (3 m). We validated the pond areas using PlanetScope and the same technique as for the Khumbu Pléoades image (OBIA) As suggested, we have also searched the Copernicus data archive for Pléiades imagery for the year 2015 ; unfortunately there are no suitable images. Therefore, we have applied and validated the method for the year 2016 in this area.**

**We additionally validated the pond results at other site (Langtang in the central Himalaya) for the year 2015 based from SPOT7 analysis for ~ the same date as our Landsat imagery (Steiner et al., 2019).**

**This has been added to the methods, and the data sources were added to Table 1.**

**With regards to the transferability of the method: the idea here was to test whether a single spectral signature extracted in an area with a variety of lithology (i.e., Khumbu) can be applied over the entire range, and to evaluate its uncertainties and limitations associated with that. We have clarified the focus of our paper in the Introduction, and have discussed at length the limitations of the method in the discussion section 4.6.**

**RC: Controls on supraglacial ponds**: this is an interesting point and one of the, if not the, main outcomes of this manuscript. However, the analysis conducted here is very simplistic, especially considering the work from past studies, and further analysis is needed here to show the significance of such results. It is difficult to see anything in the related figure 11. I would suggest conducting a more detailed analysis of the controls, especially by partitioning glaciers in elevation bands since the ponds are very variable already at the glacier-scale (this is obvious comparing Khumbu upper and lower sections for instance). The 'slope' derivation is unclear - does it relate to longitudinal surface gradient? (Quincey et al., 2007; Miles et al., 2017; King et al., 2020). Consideration of surface depressions/topography (Benn et al., 2017; Miles et al., 2017; King et al., 2020; Salerno et al., 2016) and velocity (Miles et al., 2017) would also be welcome.

**We thank the reviewer for raising these very interesting points, and we agree that a detailed analysis of the topography, geomorphology etc. on pond incidence as suggested, such as looking at elevation bins, is interesting. The overdeepenings and slope conditions that favour the formation of lakes has been addressed in several localized studies (Pandit and Ramsankaran, 2020; Sakai and Fujita, 2010; Reynolds, 2000; Linsbauer et al., 2016). Accurately evaluating these conditions at regional scales requires better constraint of glacier thickness for debris-covered areas as well as continuous coverage of high-resolution elevation data, which is not yet available. A quantification of the pond formation conditions over the entire domain is beyond the scope of the current study. However, at the suggestion of both reviewers, in the revised manuscript we have now additionally undertaken and analysis of mass balance, climate and glacier velocity trends to better understand the controls on pond incidence at regional scale. Please refer to the new discussion section 4.1 and Figure 13 in the revised manuscript.**

**We also wish to clarify that:**
   (1) **The pond database has not undergone manual corrections. Existing lake or debris cover databases, including the ones cited in this paper, typically undergo multiple iterations and improvement to produce a fully quality-controlled pond dataset, which would allow a full analysis of controls, Here, our intent is to show the potential of the method to decompose the surface composition and notably the pond coverage. For this reason, we are not exploring high resolution depressions/topography; this can be attempted in a future study at local or regional scale.**
   (2) **As the pond extent is temporally variable, here we only offer a snapshot of pond distribution for one date in a particular year, at the end of the ablation season. A more**

**complete analysis to add further process-understanding would necessarily require investigations (such as small-scale depressions/topography) that go beyond the scope of the paper in our opinion, because we wish to focus on showcasing the potential of the method.**

(3) **The 'slope' derivation here is not the one used by Quincey et al 2007, but rather the the average downglacier slope of the debris-covered section only. For the revised manuscript we binned the slope raster in 2-degree bins.**

**RC: Use of the SAM method:** The use of the SAM in this manuscript raises a few questions: Based on these different points, I feel that the SAM does not bring much added value to this manuscript, but instead is an additional method that adds confusion to the results. I therefore suggest removing it entirely, unless it was indeed used to select endmembers, in which case, two sentences about this in the methods should be largely enough (the 'endmember selection' part is already very detailed).
Why use it over the whole Landsat image if the focus is on debris-covered glaciers?
Why use it only for the Khumbu domain if the aim is to provide maps at the scale of the Himalaya? The main advantage of this approach stated here is that 'it is relatively insensitive and albedo effects'. This is because this method looks at the relative differences between the spectra, which is also the case with linear spectral unmixing when it respects the 'sum-to-unity' constraint. The advantage of the SAM over the LMM is therefore not clear. In part 3.1, L 357, you say that 'The SAM method is presented here only as an additional verification on the endmembers chosen'. This is in contradiction with the
presentation of the SAM in the methods.

**The SAM was an intermediary technique which performed better at cloud classification (hence its application over the entire image) - but since we have added an extensive analysis to our paper, we have removed the entire SAM section in order to focus only on the LMM.**

**Line-by-line comments**

Title

Title: specify 'linear spectral unmixing'

- **Done, title now reads "Surface composition of debris-covered glaciers across the Himalaya using linear spectral unmixing and Landsat 8 OLI imagery**

Title: replace 'multi-sensor' with 'Landsat 8'. Pléiades & RE imagery are only used for calibration & validation at a very specific location. Not used for the unmixing.

- **Replaced, see above**

Abstract

L 13: The study area corresponds to the Himalaya, remove 'Hindu Kush'
- **Done**

L 14: 'covered with'

- **Fixed and rephrased to : "The ablation area of glaciers often features a highly heterogeneous debris mantle comprising pond"**

L 14: specify 'rock debris'

- **We have used "debris mantle" here**

L 17: Add accent on Pléiades - change throughout text

- **Checked and corrected throughout text.**

L 18: specify Landsat 8 - add throughout text whenever required

- **We have checked and replaced with "Landsat 8 OLI" throughout the text**

L 22: The surface composition maps are still 30 m. What do you mean by 'finer classification'?

- **This entire section has been rephrased (to account for reviewer 2 comments as well) to: "Here we use linear spectral unmixing of Landsat 8 OLI images (30 m) to obtain fractional abundance maps of various surfaces (debris material, clean ice, supraglacial ponds and vegetation) present within the debris cover across the Himalaya around the year 2015"**

L 24: i would not qualify the missing 19.7% as negligible

- **The results for the entire Himalaya show 7.6 % total (clouds and clean ice) and 1.2% as missing, not 19.7%. We have added all these precents to clarify. This now reads** (l 26 – 29 **in the revised abstract):**

  **"[…] debris-covered glacier zones comprised 60.9 % light debris, 12.8 % dark debris, 5.6% clean ice, 4.5 % supraglacial vegetation, 2.1 % supraglacial ponds, small amounts of cloud cover (2 %) and unclassified areas (1.2 %)."**

L 25: This might need to be retoned - see comments on the results regarding the seasonal variability of supraglacial ponds.

- **Agreed. We clarified that we report the pond percentage at the date of the imagery and that this study did not account for seasonality. We have removed this from the abstract and have added this as a limitation in section 4.6.**

Introduction

**In order to address all the suggestions, we have completely re-written and re-arranged the introduction for more clarity, and have also added the suggested references.**

L 34: remove 'Hindu Kush'
- **Done**

L 35: suggest adding reference here - e.g. Sherler et al., 2018 and Herreid and Pellicciotti, 2020
- **Thanks. We added this on l 40-42 ; the first phrase was slightly re-phrased and now reads: "Globally, supraglacial debris cover accounts for ~ 7 % of the global glacierized area (Scherler et al., 2018; Herreid and Pellicciotti, 2020)**

L 36: rock falls – plural
- **Added the "s"**

L 42: add references to energy-balance descriptions of ice cliffs. E.g. Sakai et al., 2002; Reid and Brock, 2014; Steiner et al., 2015; Buri et al., 2016a & b
- **These had been cited on l 43 in the previous version but we have revisited this because l 42 did not mention specifically ice cliffs but rather melt under debris. To address both this suggestion and the suggestion of reviewer 2, we have added 2 references (Østrem, 1959; Nicholson and Benn, 2006) and also moved (Reid and Brock, 2010) here:**

  **L 45 - 47 : "First, they influence the surface energy receipts of the supraglacial debris surface and the efficiency with which atmospheric energy can be transferred to the underlying ice and cause glacier ice ablation (Østrem, 1959; Nicholson and Benn, 2006; Reid and Brock, 2010)."**

L 44-45: these references don't really fit here. You could keep Reid and Brock, 2010 for melt under debris.
- **See above, moved Reed and Brock 2010 to the previous line (l.48)**

Add other references to melt under debris and melt of ice cliffs & ponds.
- **We added more references, the list is now : (Ragettli et al., 2016; Miles et al., 2016; Sakai, 2002; Buri et al., 2016; Steiner et al., 2015)**

The link with Foster et al., 2012 and Miles et al., 2020 is not clear here.
- **We had given this as examples of models used to infer melt under debris - but we agree it may not be clear, and is also unrelated to the point. We have therefore removed these.**

L 51: Shugar et al., 2020a -> 2020. Change also in references. Check throughout text.

- **We have changed this. Initially we had cited the datasets as per the NSIDC DAAC recommendations, separately from the paper.**

L 54: 'properties of the supraglacial debris' - not clear. Replace with 'number/area of ponds'
- **We agree with this comment; what we meant here is that quantifying the properties of debris (for ex. slope, thinning rates etc) is needed to predict future formation of ponds (and not just the number and area).**
- **We rephrased this to (l 60 to 62 in the revised manuscript)"**
**"Quantifying the number/area of supraglacial ponds and their evolution (Miles et al., 2017; Liu et al., 2015; Watson et al., 2016) is important for assessing which ones might represent conditioning factors for hazards (Sakai and Fujita, 2010; Reynolds, 2000)".**

L 55: i.e. -> and
- **Done; see above**

L 57: There are many more studies looking at the evolution of supraglacial ponds, including some already cited in this manuscript (Miles et al., 2017; Watson et al., 2016; Liu et al., 2015 …). Add more references here or specify the statement.
- **Our statement here referred to the conditions conducing to the formation of the lakes (gradient angle etc.), and not the actual evolution of lakes, hence we have given two main studies as reference** (Sakai and Fujita, 2010; Reynolds, 2000)**. We have clarified and changed the statement as above, and also removed this statement: "Presently, such analyses have only been undertaken at single glaciers or small regions in the Nepal and Bhutan Himalaya, for isolated years only"**
- **Instead, we rephrased to (l 66 - 69): "Our understanding of the intraregional and regional difference variability in glacier mass balance, including debris-covered glaciers in the Himalaya have improved in the over the last years (Dehecq et al., 2019; Brun et al., 2017; Shean et al., 2020). The role of glacier morphology in controlling glacier behaviour and changes has been also been demonstrated in recent studies (Salerno et al., 2017; Brun et al., 2019)."**

L 60: debris cover -> debris-covered
- **Done**

L 60: the references for transition of dcgs to rock glaciers is largely incomplete. Other relevant references that could be added: Jones et al., 2019; Knight et al., 2019 and related literature
- **Jones et al 2019 was already cited in the submitted manuscript; we have checked and completed the reference list which now reads: (Shroder et al., 2000; Jones et al., 2019; Knight et al., 2019; Monnier and Kinnard, 2017).**

L 62: references?
- **Added (Scherler et al., 2018; Herreid and Pellicciotti, 2020)**

L 63: Shugar 2020b -> 2020. Check throughout text.

-   **Done**

L 63: Wangchuk and Bolch, 2020: this is more a methods paper than a regional inventory.
-   **We agree, and have removed the reference.**

L 63: add Chen et al., 2020
-   **Done**

L 64: 'not consistent' - this is very vague. Specify.
-   **This was rephrased to (l 79 - 81) in the revised manuscript:**
    **"While these global datasets represent an important development in advancing the understanding of the distribution of debris-covered glaciers at a large scale, they can suffer from the use of inconsistent methods and different temporal coverage between and/or within regions."**

L 64-66: this is repeated from the previous sentence. Remove
-   **Text has been partly removed and incorporated in previous phrases, which now reads (l 86 – 88):**
    **"Supraglacial ponds and ice cliffs are not represented either in these supraglacial debris cover databases or in the updated, publicly available regional glacier lake inventories (Wang et al., 2020; Shugar et al., 2020; Chen et al., 2020), which tend to focus primarily on the representation of proglacial lakes and their decadal changes"**

L 67: methodologies. Use plural.
-   **Change made**

L 67: is the methodology really the problem here? My feeling is that the main issue with mapping these relatively small features comes from the resolution of the sensor used to map them more than the method - see Watson et al., 2018 for comparison of sensors to map ponds with NDWI. Also in Kneib et al., 2020, we decided to use a NDWI instead of an LSU approach to map the supraglacial ponds.
-   **In the revised version, this entire section has now been reworked.**
-   **A few remarks: We all agree that spatial (and furthermore the temporal resolution) of Landsat is an issue. In our original manuscript, we had referred to the resolution of the sensor in the methodology as limitation in the data sources.**
-   **With respect to methodology, we have clarified that despite methodological developments for mapping ice cliffs and ponds using high resolution imagery, a robust and transferable method does not yet exist.**
-   **L 109 -112 now read: "Despite methodological developments, a robust and transferable method for mapping ice cliffs and ponds in a systematic manner using these high-resolution datasets does not yet exist and current methods remain computationally-intensive"**
-   **With regards to NDWI:  We added references to studies which used NDWI in l 95 -96: (Watson et al., 2018; Gardelle et al., 2011; Miles et al., 2017; Kneib et al., 2020; Liu et**

**al., 2015; Wessels et al., 2002; Narama et al., 201. While this index is still a useful method for mapping lakes, our comparison in this study shows that Chen et al. and Shugar et al. 2020 lake databases overall do not map the small and/or transient supraglacial ponds; besides, it was not the focus of their study (D. Shugar - personal communication). While Kneib et al 2020 used NDWI successfully, it was beyond the purpose of our study to apply yet another modified NDWI ; rather, we focus on exploring an little utilised method.**

L 68: remove 'only'
- **Done**

L 69: Google Earth is a tool, not a set of data. Usually, it's Landsat images that are shown there.
- **We fully agree Google Earth is definitely NOT a dataset and we had not stated so. We merely stated that a lot of studies digitize manually in GE. The phrasing may have been confusing, so we have shifted this up and revised it to:**
- **l 97- 99: "Supraglacial ponds and ice cliffs have been mapped using a combination of manual digitization on high-resolution multi-spectral imagery (1-3 m) or directly on Google Earth (Brun et al., 2018; Watson et al., 2018; Watson et al., 2017a, 2016; Steiner et al., 2019) …"**

L 69: Other relevant references for mapping multispectral images: Miles et al., 2017; Steiner et al., 2019; Kneib et al., 2020, Kraaijenbrink et al., 2016, Anderson et al., 2019, among others.
- **We have added these and many more references. We have completely revised the entire pages 3 and 4 of the introduction**

L 70: Other relevant references for mapping with topographic models: Herreid and Pellicciotti, 2018; Westoby et al., 2020
- **Added – see above**

L 73: add Steiner et al., 2019 for ice cliffs in Langtang. For ponds, see Liu et al., 2015.
- **Added - see above**

L 85: Object Based … add capital letters
- **Done**

L 78-89: Incomplete. In this paragraph you need to cite Scherler et al., 2018 and Herreid and Pellicciotti, 2020 - the current 2 global datasets for on-glacier debris-cover extents, and where they fit in terms of methods.
- **We have revised this as noted above, and have added the discussion of Herreid and Pellicciotti (2020) and Scherler et al (2018). The first paragraph, now on p3 (l 86 – 94) summarizes the advances and limitations of regional debris cover and lake databases.**

- **These pages re-organized and re-written. We have also shifted the phrase about SAR and machine learning to the paragraph concerning the methods to map ponds and cliffs using multi-spectral imagery (now on l 106 -107)**

L 90-114: this part lacks organization and the reader is lost. L 108-110 should go in the previous paragraph. L111-114 does not tie well here and should go higher with the description of applications of spectral unmixing.
- **The pages were entirely re-organized and re-written as mentioned above**

L 101: define spectral unmixing here in a few words.
- **Done, we have moved this phrase from methods to here:**
- **L 129 to 126:**
  **"Spectral unmixing routines, initially described by Atkinson (1997; 2004) and Foody (2004), allow decomposition of a given pixel into constituting materials, providing their fractional abundance and thus generating a more realistic representation of complex surfaces (Keshava and Mustard, 2002)…"**

L 105: We also used spectral unmixing to map ice cliffs in a recently published study (Kneib et al., 2020)
- **Thanks for this reference. At the time of paper preparation and submission this study was not published. We have now added this.**

L 111: The Xie et al. references do not quantify supraglacial features (cliffs or ponds) but are focused on the debris-covered area. Wangchuk and Bolch, 2020 use Sentinel imagery, not Landsat. L 111-114 is unclear and could be removed or at least rearranged.
- **We have removed this reference (Xie et al., 2020) from here but we mention  is mentioned later on in the manuscript in the context of the minimum slope of debris cover occurrence.**

L 118: teste -> test
- **corrected**

L 118: on -> over
- **Changed**

L 119: you need to demonstrate the transferability of the approach. See major comment.
- **This has been demonstrated in the revised manuscript, see the answer to major comment and the "validation" section.**

L 127-128: Transferability to open-source software is not addressed in the manuscript.
- **We agree. The transferability to open-source software could not be achieved for this study; we hope to address this is in a future study. We have therefore removed this statement.**

**Methods and data sources**

L 141: add Brun et al., 2018; Kneib et al., 2020
- **done**

L 141-142: Are these changes in glacier area? Specify if so
- **added**

L 146: There is more variability than 7-8 %. See numbers from and refer to Watson et al., 2017, who looks at the full Khumbu region, and Kneib et al., 2020 for Khumbu glacier.
- **We have revised this, and added the numbers and references:**
  **L 154 – 156 "Supraglacial ponds cover ~ 0.3 to 7 % of the glacierized area in the Khumbu based on high resolution Pléiades data (Watson et al., 2017a; Kneib et al., 2020; Salerno et al., 2012); ice cliffs cover between 1 and 9.2 % of the glacier area (Brun et al., 2018; Watson et al., 2017a; Kneib et al., 2020)."**

L 149: i do not think that the decimals are needed here - unless they correspond speci cally to the bounds of the Landsat images that were used?
- **We removed the decimal degrees but have added "~" because these are now rounded**

L 151: add Maurer et al., 2019
- **Thank you, we have added this reference**

L 153-158: The difference between the different regions is a bit unclear. I would insist on this idea of monsoon gradient (Bookhagen and Burbank, 2010).
- **We rephrased to (l 158– 165):**
  **"To examine and highlight regional differences in the composition of the debris-covered glacier tongues, we sampled four sub-regions selected across monsoonal gradients as defined in the literature (Bookhagen and Burbank, 2010; Thayyen and Gergan, 2010): the Lahaul Spiti region in the "monsoon-arid" transition zone, the Manaslu and Khumbu regions in the central Himalaya and the Bhutan region in the eastern Himalaya (Error! Reference source not found.)"**

L 163, 165 & 171: specify that the Pléiades and RE data were only used for the Khumbu region
- **Done; we rephrased the paragraph, added new data and made it more concise**
- **We added the PlanetScope image which was used for an additional validation:**
  **"A PlanetScope ortho-tile from Oct 19th, 2016 (3 m spatial resolution, 4 multi-spectral bands) was used in Lahaul Spiti area to validate the ponds resulting from unmixing the 2016 Landsat 8 scene for this region (Error! Reference source not found.)."**

L 165-166: 'so we … Landsat data'. Not necessary, remove.
- **removed**

L 170: need-> needed

- **Corrected**

L 170: the images are not entirely cloud-free.
- **Changed to: "in order to minimize cloud and snow cover occurrence.."**

L 171-173: It is actually very important that all the images are from the post-monsoon and i would recommend insisting on this, since even for different years you would expect similar surface conditions. This is especially true for ponds (Miles et al., 2017)
- **We clarified this on l 174 – 179:**
  **"We selected scenes from the post-monsoon period only (September to November) in order to minimize cloud and snow cover occurrence (Bookhagen and Burbank, 2006). In addition, Landsat scenes across the domain were selected around the same date as much as possible to minimize seasonal differences in surface conditions, notably seasonal changes in pond occurrence (Miles et al., 2017)"**

L 176: images per acquisition -> images (there is only on Pléiades acquisition). Similarly, remove 'fall acquisition' (L 178)
- **Done, this was shortened**

L 179: specify snow-free in the debris-covered part
- **Added, as mentioned above**

L 180: reference for ERDAS?
- **We added the software version and reference (ERDAS, 2010) the best we could ; there are no standard ways to cite ERDAS.**

L 182: image parts -> scenes
- **Changed**

L 182: using -> with
- **Changed**

L 183: 4, 3 (space missing)
- **Space added**

L 183-184: Have you considered correcting the Pléiades image to surface re ectance? This would give you an idea of what the spectral values are there for in an image for which you can determine the composition well. I am also surprised by the use of RapidEye image with Pléiades images, as if they were equivalent (the RE image is almost not mentioned in everything that follows). The spatial resolution is indeed very different (also the spectral), and the RE image is corrected to surface refectance while the Pléiades are not. If you consider the RE image to be enough, using Pléiades sounds like an 'overkill' since the RE images are freely available on Planet.com with an academic licence.

- We appreciate Reviewer's suggestion. We initially had not considered it necessary to derive surface reflectance for the Pleiades since this was not used to extract surfaces signature. Rather, we have extracted the surface signature from the Landsat scene itself (a well-established protocol described in Keshava and Mustard 2002) to explore the applicability of our method other areas of the Himalaya where we might not have suitable high-resolution scenes at a particular date/time.
- Also, we do not consider the Pleiades imagery an "overkill" in this case because these had already been processed for a previous study (Rieg et al . 2018), and these images are of high quality.

L 187: the RapidEye images were resampled? Why and to what resolution?
- **No, the images were not resampled to any other resolution, so the reviewer must have misunderstood. This is just a required step in any grid processing, e.g., the mosaicking step in ERDAS LPS, where the resampling method needs to be specified. To avoid confusion, we have revised this to:  "The individual image scenes were mosaicked to a single image using nearest neighbour at 2 m spatial resolution."**

L 192: elevation data used here is also remote sensing data - the tiles of 2.2 & 2.3 need to be clearer
- **We have shortened this section and moved the elevation data to section 2.3 as it was used for the topographic corrections.**

L 192: This part is confusing and mostly unnecessary. Just specify what data was used, for what reason (vertical accuracy and less voids) and what you extracted from it.
- **This was fully rephrased and shortened (see the revised section 2.3).**
- **Also, we have added a new section describing the topo-climatic variables used for the new analysis (section 2.7)**

L 193: against what did you evaluate the 2 datasets? A check could be to test against the Pléiades DEM. Also, for this study the absolute elevation values are not so much of interest, as long as they represent the topography well.
- **An evaluation of the two DEMs is beyond the scope of the study. We chose the ALOS DEM based on the vertical accuracy, and data voids. Using the ALOS DEM provided better shadow rendering when performing the topographic corrections of the Landsat 8 images, and this was now included in section 2.3 "Atmospheric and topographic corrections"**

L 193: specify time period covered by these datasets.
- **This was added : "2006 to 2011"**

L 196: If the goal is to make a thorough comparison of the 2 datasets (which i do not think is needed), you will want to compare quantitatively the data void area
- **Reviewer is correct in that we do not aim at a thorough comparison of the two DEMs.**

L 202: remove 'of interest'
- **Removed**

L 205-208: not needed here.
- **Ok, these were removed**

L 219: Why such low elevation?
- **We made a typo here, it is wavelength, not elevation, this was rectified: "we used level 1.5 data at reference wavelength of λ = 500 nm for the three sites in eastern Nepal located within our study area.."**

L 202-203: 'which provided…'. Remove. Already stated before.
- **removed**

L 229: 'debris   and/or ice cliffs, ponds…'
- **Change made**

L 234-234 & 236: Shouldn't it be 'linear unmixing models'? Also, in the manuscript you use LMM but also spectral unmixing in an equivalent way. I would recommend sticking with one term and using it consistently.
- **Thank you for this comment, which we addressed. We recognize that various formulations are used interchangeably in the literature. We found that the correct and most often used term is "spectral unmixing using the linear mixing model" and then the algorithms used are referred to as "linear unmixing" (steps related to endmember selection, etc..), as presented in Keshava and Mustard 2002.**
- **We chose to refer to this as follows: first we refer to spectral unmixing as the broad technique in the introduction and in section 2.5, and then we use the term linear mixing model (LMM) throughout the manuscript when we refer to the actual steps.**

L 243: 'for variable illumination conditions': explain better or remove whole sentence.
- **This is related to the amount of shading etc.. We agree that it is perhaps not relevant for a discussion here so it was removed**

L 243-245: why?
- **We are unclear what the reviewer refers to here. If this is related to the fully constrained model that require perfect knowledge of the system, this is simply never the case that we can fully quantify all the components on a system. We have added this to clarify :**
  **L 266- 268: "Models that comply with both conditions (called "fully-constrained models") but are difficult to achieve because they require perfect knowledge of the system, which is rarely feasible. Furthermore, fully constrained models have been shown to produce unrealistic fractions in poorly defined areas or areas of low illumination (Cortés et al., 2014)."**

**Please also note that we have fully revised the entire section to make it more concise and clearer**

L 245: 'fully-constrained' - remove space
- **Done**

L 249-253: See major comments.
- **The SAM section was removed**

L 256-270: Was the endmember selection done within the debris-covered area or using the whole scene? I suspect the whole image. How did you distinguish clean ice from snow? These two surface types can be di cult to distinguish from one another.
- **We have explored both options. Ultimately, to simplify the system, we have chosen the endmembers from the DC glacier areas only to simplify the system and obtain the fractions of materials over the DC. We did not choose snow vs ice, we only used a clean ice class**
- **We clarified this in the revised manuscript (l 276 -277)**
- **Note also that the entire paragraph was re-written**

L 257: ROIs -> actually just single pixels.
- **This had already been removed in the re-submitted version of the manuscript after main editor comments.**

L 257: MNF is only used once and therefore does not require an acronym. Same thing for PPI (L 259) and SMAAC (L 261)
- **These (and many other acronyms) had already been removed in the re-submitted version of the manuscript after editor comments; it seems that reviewer used the original submission, which indeed contain the acronyms (these were removed after Editor comments)**

L 258: What are then the results of this MNF? And this SMAAC?
- **For the MNF (acronym was removed), we clarified on l 276 :**
  **"..we performed a forward Minimum Noise Fraction Transform on the Landsat scene (Green et al., 1988), which consists in a linear transformation of the data based on principal component analysis, used to estimate noise in the bands. All bands had eigenvalues > 1, so we determined the dimensionality of the Landsat data as n = 7"**
- **For SMAAC (removed in the revised manuscript though), we found a good correspondence between our selected classes and the automated Sequential Maximum Angle Convex Cone routine in ENVI. We only used the SMAAC for as double checking the validity of our endmembers, and we did not use them in the actual analysis. As such, the lines referring to SMAAC were removed.**

L 265: remove 'ice'
- **Done ; the paragraph was fully revised**

L 267-268: this does not make sense, why take clean ice and cloud endmembers then?
- **Because, when we did, there were large model errors due to missing end members (see our explanation at the end of section 2.5.3).**
- **We have tried various combinations, including 3 classes and 4 classes, which would simplify the system but introduced large errors.**
- **We removed the following statement which was perhaps confusing : "We based this on that fact that images were mostly cloud free, and we would not expect clean ice to occur in large quantities on the debris surface"**

L 270: i understand that you are using the Pléiades image to check qualitatively the surface type of the Landsat pixels. This could be made a bit clearer in the text (the exact use of the Pléiades image). Furthermore, this raises a few questions (see major comment):
- **This was added in revised sections 2.2 and 2.5.1**
- **While image shifts were negligible (see below) we co-registered both Pléiades and Planet scenes with respect to the Landsat (see replies to major comments).**

L 256-277: endmembers may vary from scene to scene, depending on the spectral characteristics (not likely for Landsat), the illumination, the geology (for the debris pixels). Assessment of their transferability is required to validate mapping across all the Himalaya (see major comments).
- **We agree and this was one of the original questions for this study, i.e., can the same endmembers be used for the entire Himalaya domain and what are the uncertainties associated with it? As a response to one of the major comments, we assessed the performance of this parameterization in other regions (Lahul Spiti etc).**

L 269-270: how many pixels does this make then? 6? You could consider showing them in one of the gures (Fig. 1 for example. Same thing for the validation pixels, but maybe this will be too much)
- **No, these are not 6 pixels but 6 classes, each class had a number of single pixels, 10 - 38 pixels per class**
- **We had added this on a new figure (Fig. 2) because Fig 1 was revised to show the climate regimes and all the domains and validation sites and it was already very busy**

Did you coregister the Pléiades/RapidEye with the Landsat image?
- **Yes – see the answers above. We had added this to section 2.2, l 203 - 209 where we give the shifts after co-registration.**
  **"We co-registered all high-resolution images and the corresponding Landsat 8 OLI images using the Co-registration of Optically Sensed Images and Correlation (COSI-Corr) routine (Leprince et al., 2007) implemented in ENVI 5.5 Classic (L3Harris Geospatial, Boulder CO). For the Pléiades image, after co-registration with 20 tie points and a second-order polynomial transformation (RMSE = 1.3 m), image displacements were -0.16 m in the E/W direction and 0.12 m in the N/S direction. The Planet RapidEye and PlantScope scenes were coregistered on the Landsat 8 OLI with 15 and 10 tie points**

**(RMSE = 5 m and 1.6 m, respectively), yielding offsets of ~1.1-1.7 m in the E/W direction and 0.09-0.5 m in the N/S direction after coregistration. These offsets were below the spatial resolution of both scenes (5 and 3 m, respectively).**

For this to be valid, you need to make sure that either the pixel has only one surface type, or to quantify the surface types within the pixel

- **We have addressed this in the revised text. While we applied the automated PPI routine, which does minimize the mixed pixels we acknowledge that some pixels may still not be 100% pure because of the resolution of the Landsat. We acknowledge this as one of the limitations in section 4.2.**

L 275: The data from Casey et al., 2012 would be more relevant (also for the Khumbu area)
- **added**

L 275-277: show the spectra from Matta et al., 2017; Naegeli et al., 2015; 2017; Casey et al., 2012 in Fig. 3b.
- **We have not used these quantitatively, only for visual comparison, so we consider that 2 spectra figures from our own work are sufficient here, the reader can refer to those papers.**
- **We have rephrased to (l 291 – 293):**
  **"We visually compared these spectral signatures with those previously acquired in the field on Mer de Glace in the French Alps using an SVC HR-1024 spectrometer (350 nm to 2500 nm) (Racoviteanu and Arnaud, 2013) as well as with supraglacial debris spectra from other papers (Naegeli et al., 2015; Naegeli et al., 2017; Casey and Kääb, 2012)"**

L 280-281: spectral unmixing -> linear spectral unmixing (or LMM) to be used consistently
- **Checked and fixed, we used LMM**

L 282: this evolves through the manuscript, sometimes 'SDC v.1 dataset', sometimes 'SDC', sometimes 'SDC dataset'... i suggest using just SDC consistently.
- **This was checked and fixed throughout the manuscript, we have used "SDC dataset"**

L 282: Landsat 8 and Sentinel-2
- **Added**

L 283: There is only one Scherler et al., 2018. Correct throughout text and also in the references.
- **Done**

L 283-284: not particularly relevant, remove.
- **Removed**

L 287: What about expanding debris-covered areas (Kamp et al., 2011; Thakuri et al., 2014; Xie et al., 2020; Bhambri et al., 2011…)?
- **Thank you, this is an excellent point and we have added it to the discussion of the limitations of the study in section 4.5:**
- **"However, recent studies have reported an upward expansion of the debris cover in the Himalaya (Xie et al., 2020; Thakuri et al., 2014; Bhambri et al., 2011; Kamp et al., 2011), which we do not account for here."**

L 290-291: How was this automated check and repair performed?
- **This was done with the "repair geometry command" in ArcGIS**
- **We have added this on l 240: "We fixed these errors in the SDC polygons using the Repair Geometry command in ArcGIS v10.8., in order to, we "filled" the holes so that they were included in the SDC polygons..."**

L 291: What about the other areas?
- **We overlooked correcting this line. This was also applied over the whole domain.**

L 300-301: how was this evaluation performed?
- **This does not apply anymore because the SAM section was removed from the revised manuscript**

L 302-308: this is mostly repeated from before. Remove/reorganize.
- **We have removed the SAM but have kept the explanation about how we obtained the normalized rasters and we removed duplicates at the end of that section.**

L 310-311: what happens if a pixel satisfies two different thresholds?
- **Where this arises, those pixels were considered ambiguous and "unclassified"**

L 315: 'fner classification map': the maps are still 30 m resolution and only one surface type is selected in the end?
- **This was rephrased to "30 m fractional maps", please see revised text**

L 321: remove space
- **Done**

L 331-332: it would help to show these 'validation' pixels on a map
- **We agree. As stated above, we made a separate figure for this (Fig. 2)**

L 333-335: reference?
- **Added the reference for the ENVI feature extraction on l 349: "..using OBIA techniques (Blaschke et al., 2014) implemented in the ENVI Feature Extraction Module (HarrisGeospatial, 2017)."**

L 337: How? Confusion matrix? Final pond area?

- **Manual corrections of final pond areas as mentioned on other papers (Watson et al 2017).**

L 339: Watson et al. (2018).
- **"." was added**

**Results**

L 341: see comments above. I suggest removing this whole part.
- **Agreed, and we have removed this**

L 343: in figure 4, how were the shadows mapped?
- **These were output of the topographic correction in ARCSI (but Fig 4 is now removed); however, the detail about the shadow mapping was added in the Methods section**

L 343: 'well': qualitative statement, di cult to interpret
- **Paragraph was removed so this does not apply anymore**

L 352-353 & 356: if this is the case, this should appear in the methods
- **Paragraph was removed so this does not apply anymore**

L 359: this part is very descriptive and qualitative. Shorten and merge with 3.3.
- **This was shortened and the part relating to model uncertainty was moved to the discussion section**

L 362-365: move to discussion.
- **Paragraph was removed so this does not apply anymore**

L 366: RMSE
- **changed**

L 367-368: remove sentence (repeated)
- **Done**

L 370: Is 'normalized fractional maps' not the equivalent of 'partially constrained'? In both cases the coe cients of the unconstrained model were normalized? This would then be in contradiction with what is stated above?
- **Theoretically yes, but we could not obtain the sum-to-unity for either model; therefore, we normalized the outputs using established methods so that fractions added to 1 for each pixel. When the model works perfectly, normalization would not be needed, but as mentioned here, we did have negative fractions and positives > 3.**

L 376-377: The names of the glaciers do not appear in the figures, which makes it very di  cult for anyone who's not familiar with that area to follow.
- **This was an overlook on our side, we had intended to label them. This is now fixed.**

L 384: de  ne Kappa coe  cient. It is the only occurrence of this coe  cient, is it really relevant?
- **The entire section was redone and we now report precision, recall and Fscore because new research (Foody, 2020) shows that the Kappa coefficient is not a very adequate measure. Fscore is the recommended metric so we have used this.**

L 387-388: give an estimate of the cloud-covered area
- **We do not have estimates of the cloud cover. Those reported in Landsat metadata are irrelevant because these might not be over the DC tongues). This is a qualitative statement and we have marked it as such, since we selected cloud-free images as much as possible.**

L 388-390: discussion.
- **Moved to section 4.4 in the discussion**

L 384-400: this section is repetitive, especially because the accuracy values are the same than the producer's accuracy. I suggest using the Dice coeffcient (Dice, 1945), which
takes producer and user accuracy into account in one metric (add it to Table 2). This would make this part easier to read.
- **As we noted above, the accuracy analysis was entirely re-done and re-written. We have looked into Dice's coefficient but this works to measure area overlap- and we are working with pixels. However, we are repotting Fscore, which is similar to Dice's and suitable for our data (# pixel counts)**

L 405: debris-covered
- **Corrected**

L 415: correctly -> usually
- **Changed**

L 421-422: this is mostly discussion.
- **Section was re-written**

L 428: rather than climatic patterns, i would think this to be related to the local meteorology.
- **Agree, this was revised.**

L 436: Specify Fig. 8b for Labeilong Gl.
- **Done**

L 438-441: discussion.
- **Agreed, moved there**

L 449: it is really difficult to make out the ponds in figure 6. Should it not be figure 9 instead?
- **Reviewer is correct, this should have been Figure 9. We have corrected this and moved this figure up (please note note the order of figures has changed).**

L 455-457: it would make sense to use a confusion matrix with a Dice coefficient instead. Part of this paragraph would t better in the methods.
- **See above, we have re-done the accuracy analysis (see sections 2.5.3 and 3.2)**

L 457-459: Not relevant, this is for snow and this paragraph is about water. Plus, this belongs to the discussion.
- **What we meant here was what thresholds were used in other studies. (methodology). We have moved it to discussion**

L 473: It would be interesting to compare these results with the results you would obtain with an NDWI-based approach, following the same calibration-validation scheme than for the spectral unmixing
- **While this would be interesting, it would add yet another analysis component to the paper. It is not the goal of our study as this has been done elsewhere already but perhaps can be addressed in a subsequent study.**

**Discussion**

**Please note that the discussion section has undergone major restructuring and additions.**

L 481: debris-covered
- **done**

L 484-486: you mention this distinction between light and dark debris as coming from the geology. Could this not also be related to the debris water content? Especially if the debris is very thin (as for the thinly debris-covered ice cliffs), i suspect that this could play a role?
- **We agree and we have added this on along with some references and point the reader to Fig 4 which shows the spectral differences in dry and wet debris:**
  **"With regards to the on-the ground spectral characteristics of the debris material in the Khumbu region, Casey et al. (2011) showed that these vary due to the presence of various of minerals, notably distinct granitic (lighter) vs. schistic (darker) debris types. We defined both a light and dark debris endmember to capture the variability notes in the cited study, but we are aware that these may not cover the wide spectrum of lithology present across the Himalaya. Tests using a single debris endmember resulted in large RMS errors, indicating a missing or poorly defined class. Furthermore, spectral differences in these two classes can also be related to debris water content especially on very thin debris (as for thinly debris-covered ice cliffs) and is associated with grain size, i.e. fine-grained sediments have a greater capacity for water retention (Juen et**

**al., 2013; Collier et al., 2014). We also noted this difference in the spectra for the wet fine debris and the dry coarse debris large grain size on Mer de Glace (Fig. 4b)…."**

L 492: the analysis
- **added**

L 492-493: remove 'chosen … image'
- **Done**

L 494: reference? Mention that in the post-monsoon the snow-cover is usually minimal (unless there are early snowfalls, which can happen)
- **This was based on our observations. We have added the phrase about early snowfalls**

L494-497: Note that there can be, especially in the post-monsoon, very bright cliffs. This is true after a light snowfall when the snow sticks longer to the ice than to the rock, but i
have also seen a lot of cliffs with clean ice, especially in the post monsoon. Your Fgure 2b is a good example, you can also have a look at Kneib et al., 2020, Fgure 1 - in this paper
we used 2 thresholds to map ice cliffs: one for the clean ice and one for the dirty ice. Some of this clean ice could also come from ice sails (Evatt et al., 2017) - there are some of
these in the upper part of Khumbu, and they are common on glaciers in the western part of the range. The main limitation for these two features is obviously the size, and the
mapping will be limited to the largest features.
- **Thank you for these interesting thoughts, we have incorporated these in the revised manuscript in the discussion (section 4.2 with regards to limitations to the spatial and spectral resolution of Landsat)**

L 499-500: Mapping ice cliffs with a 30 m resolution sensor is not realistic, and the results would not be representative. I suggest removing this sentence.
- **We agree, and have stated this clearly /removed most of the formulation**

L 503-504: more than the method or the spectral resolution, the limitation will be the spatial resolution. References to studies focused on ice cliff delineation would be welcome.
- **We clarified this and added references**

L 505-517: Ponds are di cult because they are very variable from one season to the other and from one year to the next. The area covered by ponds should be minimal in the
post-monsoon (e.g. Miles et al., 2017). This point should be highlighted in the discussion, with references to studies looking at pond variability (Liu et al., 2015; Miles et al., 2017;
Watson et al., 2016). It would also be interesting to compare the numbers you get for other regions with other studies (Liu et al., 2015; Miles et al., 2017; Watson et al., 2016;
Kneib et al., 2020) focusing on other glaciers.
- **See our answer to major comment**

L 510-511: explain this statement.

**This was revised in the new discussion section (see section where we address the climatic differences among the regions with ERA5-Land data**

L 514: in on -> in
- **Removed "in"**

L 515: suggest: angles -> gradient. Also, how is this calculated? Specify in methods.
> **We changed this to gradient.**
> **The slope analysis has been entirely been revised as we are now presenting pond incidence per 2-degree slope bands rather than slope gradient per glacier (see section 3.6), because the glacier-by-glacier results were not conclusive in the previous version of the manuscript.**

L 516: reference?
- **Not sure that the reviewer suggest here. This is based on our own results/discussion**

L 517: reference?
- **idem**

L 520-521: it is not clear why the bright non-glacierized area would not be mapped as light debris instead of vegetation.
- **We think that bright non-glacierized areas are to some extent covered with low shrubby vegetation, not just "bare terrain", or it is also possible the rock debris is more lichen covered than on glaciers. The latter is a speculation and we have no proof of this.**

L 530: simplify sentence
- **The phrase was separated in two now**

L 532: acronym already introduced in introduction. Since only used twice, acronym is probably not necessary
- **agree; this has already been removed in the submitted version after editor comments**

L 535-545: this should go in the methods & results.
- **We have moved this to methods (section 2.6)**

L 536: debris-covered
- **Corrected**

L 536-537: what does the size of the glacier have to do with the turbidity of the ponds?
- **The phrase was not in the right place, it belonged to a later paragraph, we have revised this. Matta et al found in their study that lakes with turquoise -grayfish water**

**were generally larger (> 1 km$^2$) than the blue lakes and they were also close to the glaciers (hence larger suspended sediment)**

L 540: How did you derive the slope? Explain. Seeing that the pond coverage is so variable even at the scale of one glacier, and so is probably the slope, my suggestion would be to look at the results in terms of elevation bands (or other glacier partitioning - possibly based on slope?)

**In the initial version of the manuscript, slope was derived as the average of the pixels in the debris cover part only. As noted by reviewer, this did not yield conclusive results because of the variable pond coverage at glacier scale. Therefore, in the revised manuscript, we revised this analysis to address this using binned elevation, slope and aspect. For example, we calculated the slope gradient per pixel and have reclassified it to 2-degree slope gradients and summarised the pond incidence within each slope bin.**

**As requested by reviewer, we have also looked at results (pond incidence) in terms of the elevation bands, i.e., we normalised 100 m elevation bands above glacier terminus, and reported the pond area within each band. We present this in the revised Fig 11a. The slope analysis is presented in Fig 11b, and the pond incidence per aspect in 45-degree bins in Fig 11c.**

**Details on methodology were added in section 2.7**

L 529: A lot of this paragraph should go in the methods + results.
**We agree and have moved this to these sections**

L 554: Have you looked at the relationship (for ponds and vegetation) with debris stage? (Herreid and Pellicciotti, 2020).

**This is an excellent suggestion, we have considered this, and we agree it would be interesting to explore this. Herreid and Pelliciotti (2020) introduce four new metrics describing the evolution of a debris covered glacier on the basis of the 'debris-cover carrying capacity' of a glacier. While all the datasets are available on Zenodo (https://doi.org/10.5281/zenodo.3866466), their study uses a corrected debris cover dataset as we explained in the introduction, which would imply re-doing the whole analysis. In a future study we plan to follow up on this study using the updated debris covered dataset and a quality-controlled pond and/or vegetation dataset to perform a multi-dimensional analysis of potential causal/driving factors.**

**We have added a paragraph explaining the outlook for a future study in the discussion section (section 4.1, l 563 – 570).**

L 563: errors in the SDC
- **Change made**

L 565: it would be interesting to look at the changes of ponds and vegetation from east to west more in details
- **We are not investigating [temporal] changes in either at this point. Does the reviewer mean "variability"? If so, this is already discussed in the manuscript and it is addressed now with the detailed analysis**

L 566: 'cannot be examined here in detail' -> 'is beyond the scope of this study'. I disagree, i think the analysis can be taken a bit further with the available data. It would actually add a lot of value to this manuscript.
- **We agree and the analysis was largely expanded (see sections 3.6 in the results and 4.1 in the discussion), in reponse to Revieers major comments 3**

L 575-581: this is not convincing. One problem is that the fraction of water will also be lower at the pond margins - and since Landsat 8 has a relatively low resolution, this will be the case for most pond pixels. You also do not present any results on this topic. As such, this paragraph can be removed.
- **We have removed these lines.**

L 576: 'fraction of a pond pixel covered…'
- **Change made**

L 584: this is not the focus here since the delineation was applied only to ponds within the debris-covered area. Also, the main difference noted between the different datasets is the mapping of the supraglacial ponds, while it is noted that there are no major differences for lakes outside the glacier areas. Therefore i would not mention the lakes outside the glacier boundaries but focus on the mapping of the supraglacial ponds, which is still a relevant discussion point. Finally, one problem that arises when applying your approach to off-glacier lakes will be the endmembers you used, since the turbidity of the lakes, but also their depth, will be quite different from those of the supraglacial ponds.

**In Figure 13 we had presented a test of this using a prior parameterization of LMM using clear water endmember (see Fig 3 spectra). While we do not expand of the discussion here, we feel it is important to mention that LMM can be applied using the clear water endmember outside the glacier areas as well. However, we are not emphasizing this aspect. Also, we present this for comparison purposes with the existing databases which do not differentiate between supraglacial ponds and lakes outside glaciers.**

**This was revised in section 3.4 now called "Application for regional non—glacier lake databases" :**
- **"While supraglacial ponds are the focus of this study, we mention that LMM can be parameterized by replacing the turbid water endmember with a clear water endmember, which have different spectra (Fig4a). LMM can be applied by masking the debris cover glacier areas to obtain lake extents outside the glacier areas across the mountain range.**

**This is beyond the purpose of this study, but we provide an illustration of such an output for the terminus of Ngozumpa Glacier (Fig 7). ..”**

L 591: this 'outperformance' is only true for supraglacial ponds (at least in Figure 13)
- **We have clarified this**

L 599: note that the outlines shown from Chen et al., 2020 have obviously been manually delineated.
- **Revised, this now reads: "..but based on normalized difference water indices (Shugar et al., 2020) or manual delineation (Chen et al., 2020);**

L 600-601: you need to mention the pond variability, which could explain some of the differences here.
- **This section relates to uncertainties in the LMM model. Uncertainties in pond variability are related to their spectral variability (turbidity), and this is integrated in the discussion of endmember selection in section 4.3**

L 609: in -> to
- **Not sure what reviewer is referring to here. In our view, " overestimation of pond coverage [.....] in some areas" seems correct**

L 613: this needs to be proven
- **We have removed this statement was it was vague but we refer to the Xie et al (2020) study**

L 615-616: what corrections do you have in mind? So this is not a final product?
- **Here we referred to heavily vegetated areas in the SDC which were errors, and which were removed when testing for the Khumbu area. We did not remove them over the entire range because the LMM picked it up automatically and this is a useful output which can be used to improve the SDC. If the reviewer refers to the final corrected SDC product, this has not been tackled in this study but it provides a basis for it.**

L 630-631: Where are these results? Are they of any use? If yes, they should be discussed in more details

- **This is shown in Fig.3 . We do not go into detail about the spectral variability of the various vegetation types but merely note that we define the vegetation class as "dry vegetation" (see fig 3)**

L 631-632: in Kneib et al., 2020 we also used a light and dark debris endmember
- **We have added this. At the time of preparation/submission of the paper, that study had not been published and we were unaware of it.**

L 639: were they removed or were they not?

- **These were not removed, as per our comments above. This was not feasible at the entire Himalaya range to do this on a manual basis, but the LMM results can served as a basis. This whole section integrated other sections and was re-written, please refer to the revised discussion section**

L 642-646: It seems that the use of this Scherler et al., 2018 SDC triggered a lot of small issues and it occupies a large part of the methods, results and discussion. No inventory will ever be perfect but do you think that your results could have been improved using the Herreid and Pellicciotti, 2020 dataset, that claims to be 'better' than the SDC you used? The main drawback of this dataset being that they used updated glacier outlines…
- **We cannot assess this. It would be an interesting study in the future, but our study was developed over 3 years using the Scherler dataset and the "automated" vegetation mapping resulting from LMM is a useful result here as a quality control of that database. While Scherler et al., also used spectral unmixing, they did not detect these vegetated areas.**

L 651: The Pléiades and RapidEye were only used for endmember selection and validation of the method.
- **We have clarified this as follows: "The Pléiades and Planet high resolution imagery were used for to assess the endmember selection and to validate the supraglacial pond areas"**
- **Note: we now refer to Planet in general (not only RapidEye) because we have added Planetscope in addition to RapidEye**

L 653-655: This has not been proven
- **This is addressed in response to major comments. We test the applicability, or the performance of the algorithm, ie., can we use the same end members etc..**

L 656-659: my understanding is that Shugar et al., 2020 used a mosaic of Landsat images to map the lakes, which means that only the persistent large lakes would be mapped anyways. So this problem is not related to the NDWI. The NDWI approach may not be perfect, but some studies have demonstrated that it works fairly well (Miles et al., 2017; Watson et al., 2018). I am not convinced by this point and would recommend a comparison of your results with a NDWI-based approach (following the same calibration scheme as for the spectral unmixing).
- **See our comments above, we present a comparison to NDWI based on 2 inventories for a small area.**

L 668: remove 'is of interest'
- **Done**

L 689-690: remove last sentence.
- **Done**

L 702-703: this is a key item and one of the main results of this study. It will be useful to have a link in the article.
- **We have now added the Zenodo link in the article (section 8- Data availability)**

L 975: remove, appears twice.
- **Fixed**

**Tables**

Table 2: use same number of decimals in the whole table
- **There is no need to do this; we already follow standard conventions. Otherwise if we use 2 decimals some classes have 0.00 and if we use three, then have to write 0.020 etc..**

Table 6: explain in caption what 'manual' and 'automated' spectral unmixing refer to. All these outlines are from this study - no need to specify.
- **We have removed this in the revised manuscript and we are just presenting a single output, labelled as "fractional", and we explain this**

**Figures**
Figure 1: caption: give description of images used for Himalaya map.
- **These are the Landsat scenes from Table 1, we have added this to the figure caption; the figure caption was re-written to incorporate other changes (added the Himalaya regins, the sub-domains and the test sites etc..)**

Figure 2: acknowledge pictures' photographer(s).
- **We had already confirmed the copyright to TC upon submission, these were main author's pictures, we have now added this in the caption.**

Figure 3: caption: for panel b, add references. Add titles for the 2 panels. Panel a: plot x axis as wavelengths and use the same x axis for panel a & b. Why are the values so low for the debris in panel b compared to panel a?
- **Re: titles, panel titles are not necessary, and we prefer not to clutter the figures with them, as we explain them in the caption.**
- **Re: debris reflectance: these are different sites, so we expect variability in reflectance due to lithology. In panel b, it is expected that fine wet debris to have a low reflectance, this would correspond to a "dirty ice" class.**
-
Figure 5: only one legend needed. Same thing for scale & N arrow. Increase visibility of legend
- **done**

Figure 6: I cannot see any letters in the maps. Only 1 legend, N arrow & scale (if same scale) needed. Consider adding a panel where you zoom in on a glacier tongue to see the ponds/vegetation/ clean ice better. No clouds in this image then?

- **Re-done. As we mention in the text, there were no clouds in this part of the image**

Figure 7: the Pléiades has a very 'red' appearance - i suggest adjusting band composition to make it look more like what the human eye would see and doing it consistently in the other   gures. Usually supraglacial lakes do not have a crescent shape.

**- We prefer to use a false colour composite here because the lakes are much more visible compared to a true colour composite (which we now use for other figures in the revised manuscript, Fig. 1, 2 and 6 for example**
**Re: crescent lakes, we meant ice cliffs (only) and circular for ponds. We have corrected this.**

Figure 8: I cannot see the black arrows mentioned in the caption.

- **This was fixed, missing errors were added**

Figure 9: the comparison of the lakes is not clear. It would help to have one of the datasets fully transparent, with just the outlines. The debris-cover outlines box is hard to see in the legend.

- **We have re-done the figure with the above suggestions; we have also added another validation site (based on PlanetScope for Lahaul Spiti area)**

Figure 10: sq.km. -> km2

- **this figure was removed**

Figure 11: Hard to see anything. Change colors. Plot in different panels ponds and vegetation. Try log scale (for panel b at least that could be useful). sq.km. -> km2.

- **We have replaced this figure with plots of pond incidence per elevation, slope and aspect bands (Fig. 12) (see comment above)**

Figure 12: describe what the background images are. Only one scale and one N arrow needed. It is di  cult to make out anything. Try zooming in a bit? The red-green combination in panel b is not ideal.

- **We have added the description of the images – however, this is the same as the ones in Figure 1. With regards to colour, we have used the same colour scheme for the ponds and vegetation (as well as other classes) throughout the manuscript so we prefer not to change it here. We consider vegetation in panel b is well visible.**

Figure 13: Consider increasing background transparency to make the outlines stand out. It's di  cult to see the purple outlines. Also for the light blue ones. Consider increasing line width.

- **We have attempted all these changes and the figure should be clearer now. This is now Fig.7 in the revised manuscript**

**We thank again the reviewer for the thorough review of our paper and the detailed and valuable line by line comments. These have all been addressed in the revised version of the manuscript. Please find our replies and changes in bold text.**

**General comments**

**With respect to the 3 important issues identified by the reviewer, we area please to say we have addressed all of these in detail, as follows**:

In terms of structure, the manuscript does not always follow a logical flow. There are parts of the results that are more fitting for the methods, and complete new analyses and several figures introduced in the discussion. I therefore suggest the authors to restructure quite substantially. I also feel there is often a mismatch in the distribution of details among different components, especially in the methods. Some parts are described overly detailed, while other (often important) parts of the methodology are dismissed with a single sentence. Please refer to my line-by-line comments below where I identify several of these issues. I would, however, primarily suggest the authors to carefully reread their manuscript with this in mind.

**Thank you for these suggestions. We have reworked and reframed the paper to present a more balanced level of detail of the various aspects of the methods applied and the associated strengths/limitations. As requested, we also moved the analyses presented in the Discussion to the Results as requested and we made all the other improvements to the structure suggested within the line-by-line comments. Furthermore, we have added an entire sub-section on the methods used for the analysis of the top-climatic controls on pond incidence (section 2.7 in methodology and 4.1 in the discussion section in the revised manuscript)**

The methods are both developed and validated for a single and relatively small subset of the entire domain over which they are applied. This is of course not ideal, particularly since the full domain is roughly 2000 km wide and considerable differences are to be expected over this large area. This could, for instance, be differences in lithological and morphological composition of the debris due to differences in geology and climate, atmospheric differences that could affect image corrections, differences in overpass time (i.e. solar zenith angles) etc. It would be very strongly recommended to seek further validation of the upscaling performed in the paper using additional high-resolution imagery outside the Khumbu region. Preferably far away, e.g. in Spiti Lahaul. Since acquisition of RapidEye by Planet Labs, academic access to the images is free. Additionally, almost all high-res satellite data (i.e. SPOT, WorldView, GeoEye, Quickbird and Pléiades) is accessible to European/Canadian researchers directly from archive (or even for tasking by submitting a small project proposal).

**We expanded our data set by adding two more pond validations outside the Khumbu region as requested (Lahaul Spiti glaciers and Langtang glaciers) – this is a total of three pond validation sites. We also added one more domain to explore the composition of the debris (Manaslu region) in addition to the existing three (Khumbu, Lahaul Spiti and Bhutan); this now amount to 4 sub-domains across the monsoonal gradient. However, we note that as per the definition of the regions in the large databases (see also Brun et al. 2017, Dehecq et al. 2020), two of the sub-domains (Khumbu and Manaslu) fall within "central Himalaya", and one in the dry monsoon transition zone (Lahaul Spiti). These sub-domains and the pond validation sites are now shown on Figure 1 along with the three climatic regions (western, central and eastern Himalaya). As correctly pointed out by reviewer, we anticipate that variability in the composition of glaciers in the same climatic zone could be due to differences in lithology, among other reasons.**

**With respect to the comment about differences in lithological and morphological composition of the debris, we are fully aware of these differences. We are also aware that, because we could not capture the full variability in lithologies in the study region using Landsat data, there may be uncertainties. With respect to atmospheric differences that could affect image corrections, as described in the manuscript, we tested the Dark Object Subtraction approach used in the ARCSI routine against ground data from two sources (CAMs and Aeronet), and the atmospheric profiles were then derived for each scene. We consider these to be the less important sources of errors but nevertheless these uncertainties are fully discussed in the revised manuscript.**

**With respect to high resolution data, we have searched the ESA data for this year and unfortunately there are no Pleiades, SPOT or WV scenes for this year for any of the sites where we wish to validate so we relied on Planet. It is beyond the purpose of our study, timewise, to submit a research proposal.**

As mentioned above, the paper is method-focused and as such presents only (very) limited process-related analysis. Particularly for publication in The Cryosphere, I think it is important to include a more advanced analysis, and provide a better and more elaborate discussion in this regard. This would improve the paper and more clearly indicate to the readers the potential of the method as a basis or input for subsequent cryospheric/hydrological analyses. Currently the main focus lies with supraglacial ponds, and in principle this is fine, but the current analysis using a simple linear regression of glacier-wide aggregates is very limited and certainly not state-of-the-art. I am also uncertain about the validity of using linear regression in this case, and if the authors were to continue using this method they should assess and clearly indicate the assumptions that are made about the data and its distribution when applying this technique. I suspect there is considerable non-linearity in the relations between pond/vegetation and glacier characteristics, and other machine learning techniques could therefore be better suited here, for example Random Forest Regression. Furthermore, past studies have shown different elevation bands to have very different concentrations and distributions of supraglacial ponds (e.g. Kraaijenbrink et al., 2016; Miles et al., 2017; Ragettli et al., 2016), and the analysis at the glacier scale cannot incorporate these important specifics. I would therefore strongly suggest

the authors to, instead of looking at entire glaciers, perform a lumped or distributed analysis of some sorts. I also think there are several additional variables that are worth exploring. Topographic ones, such as aspect, but there is also data about individual glacier change that would be valuable to link to (Brun et al., 2017; Dehecq et al., 2019; Shean et al., 2020). Finally, it would also be interesting and relatively straightforward to employ a more quantitative approach to the climate arguments presented by the paper, for example by including climatologies derived from ERA5 reanalysis data to the supraglacial pond analysis. Implementing things would allow to quantify many of the now qualitative statements, which would greatly bene t the message and value of the paper.

To summarise. I believe the manuscript displays an interesting, largely unexplored approach that could provide a valuable contribution. However, (i) the structure of the manuscript requires some reworking, (ii) validation outside the Khumbu region is necessary, (iii) a more rigorous analysis is required with respect to the supraglacial ponds.

**We thank the reviewer for these interesting ideas for further analysis. As we stated in the paper, given the limitations of the Landsat data, our intent with this publication was to focus initially on demonstrating the method, and therefore we were cautious not to overinterpret our data.**

**At the suggestion of the reviewer, we have undertaken the analyses of the pond incidence with respect to (a) glacier thickness changes from Shean et al., (2020); (b) glacier velocity from Dehecq. et al., (2015) and (c) climate data (temperature and precipitation) from ERA5-Land. In addition to debris cover and minimum elevation. We welcomed the reviewer's suggestion to aggregate the data rather than to present the data glacier-by-glacier. We have performed the analysis of the pond incidence by elevation bands above glacier terminus, as well as binned slope and aspect. For the thickness change, glacier velocity and climate data, we aggregated the data by 1° x 1° grid as used in other studies, for example Dehecq et al. (2020), so as to provide consistency with the existing datasets mentioned. We included two figurea in the revised paper (Fig. 12 which replaces the previous glacier-0by-gacier figure and Fig. 13 for the analaysis on the pond and vegetation controls)**

**With respect to the use of machine learning algorithms (i.e., RF), we consider this to hold promise, but be beyond the scope of the current paper, which is already quite substantial in length. We prefer to recommend a full analysis with machine learning involving a quality-controlled dataset to remove any outliers or errors, which may be done in future versions of this dataset.**

Line-by-line comments:

L13. The presented study does not encompass the Hindu-Kush, so I would suggest to remove it.
- **Done**

L13. "cover mantle" -> remove either cover or mantle. I would suggest mantle.

- **"cover" was removed**

L18. Landsat -> Landsat 8 OLI
- **done**

L20. "We develop", this implies that you developed the spectral unmixing technique yourself. Rephrase.
- **Agreed. We rephrased to " Here we apply linear spectral unmixing methods.."**

L22, L26. Use "classifications" instead of "maps"
- **we have completely rephrased the abstract and this line now reads:** "The spectrally unmixed surfaces are subsequently classified to obtain maps of composition of debris-covered glaciers across sample regions"

L22. "finer classification maps", how fine?
- **This was removed/rephrased as above (we did not mean finer, we meant subpixel)**

L22-26. Also mention more clearly in the abstract that you focus on the debris-covered part (as classifed by Dirk Scherler) only.
- **Done, see line 20 in revised manuscript, however since this is an abstract we did not include the reference for the paper in the abstract**

L24. What does negligible mean here exactly, and if it is negligible, why were all these classes included?
- **This was clarified and lines 25 -27 now read: "…at the end of the ablation season, debris-covered glacier zones comprised 60.9 % light debris, 12.8 % dark debris, 5.6% clean ice, 4.5 % supraglacial vegetation, 2.1 % supraglacial ponds, small amounts of cloud cover (2 %) and unclassified areas (1.2 %).**
-
L35. Again, suggest removal of "mantle"
- **The comment on l 13 suggested keeping "mantle" rather than removing it; here reviewer recommend to remove it. In this case, we have kept "presence of a debris mantle" to make it consistent with l 13**

L36. Would be good to include (Evatt et al., 2015) here
- **Agreed, this was added.**

L39-41. No reference for this? (e.g. Nicholson and Benn, 2006; Østrem, 1959)?
- **These were now added**

L45. Pro- glacial -> pro-glacial. Also, why supraglacial without hyphen and pro-glacial with hyphen? Please be consistent.
- **We have checked and only use "proglacial" throughout the manuscript**

L47. Pro-and -> pro- and
- **Added space**

L58. Intraregional and regional differences and variability in rates of glacier change have become reasonably clear over the last years (e.g. Brun et al., 2017; Dehecq et al., 2019;

Shean et al., 2020)

- **We agree with this, we meant there are still gaps in the understanding of the specific role of debris covered features at reginal scales. We have incorporated the suggested change and also added more references related to the importance of glacier geomorphology in mass balance so this now reads : "Our understanding of the regional variability in glacier mass balance of both clean and debris-covered glaciers in the Himalaya has improved over the last years (Dehecq et al., 2019; Brun et al., 2017; Shean et al., 2020), and the role of glacier morphology in controlling glacier behaviour and changes has been also been demonstrated in recent studies (Salerno et al., 2017; Brun et al., 2019)."**

L63-67. Include (Herreid and Pellicciotti, 2020; Scherler et al., 2018) here.
- **Done; this is now included in l 71 - 73:**
  **"Until recently, previous efforts to map debris-covered glaciers focused primarily on their extent rather than the surface characteristics, with a focus on creating global databases of supraglacial debris (Scherler et al., 2018b; Herreid and Pellicciotti, 2020)."**
- **On L 67 – 68: Also, next phrase was clarified that the cited studies were done at regional scales: "At regional scales, debris cover extents were mapped using a combination of digital elevation models.."**

L68. "Object-oriented" à "object-based". Object-oriented image analysis (OOIA). Object-based image analysis (OBIA).
- **Change made, it was indeed an error.**

L73-74. Second part of sentence need to be rephrased.
- **We do not see the issue. The phrase has been changed in the revised manuscript.**

L89. "Planet" is not a satellite, but a company. Pléiades is written with an accent aigu on the e. There is also SPOT, Worldview, GeoEye.
- **Agreed. Changes made. We did not mention all the possible commercial satellites, we stated "etc., This now reads (l 95 – 97): "The increased availability of high-resolution (0.5 to 5 m) remotely sensed data from Pléiades, SPOT and Quickbird satellites etc., complemented by freely available RapidEye, PlanetScope and SkySat images from Planet, has offered new opportunities for characterizing the surface of debris-covered glaciers in more detail.".**

L92. (Kraaijenbrink et al., 2016) already showed big differences between UAV-derived ponds and RapidEye-derived ponds.
- **We agree, and acknowledge that their study as was a small-scale study. We added this on lines 112 -113 : "For example, large differences were shown between UAV-derived ponds and RapidEye-derived ponds (Kraaijenbrink et al., 2016)."**

L92. "archive Landsat series" -> "the Landsat archive"
- **Done**

L93. still?

- **Removed, perhaps not needed- it was meant that in spite of increased high-resolution data, medium resolution data is still a valuable resource. Rephrased to: "Even with the increased availability of high-resolution imagery, medium resolution data from archive Landsat series (30 m spatial resolution.."**

L94. The Landsat archive indeed spans five decades, but the 30 m data (TM, ETM, OLI) only four. Landsat 4 was launched in '82 if I recall correctly.
- **Thanks for this. We have corrected it.**

L94. I would not necessarily call this a drawback, as it can be advantageous for some applications
- **we replaced this with "limitations"**

L95-96. "which…sensor". This is not a discriminating factor between full-pixel vs sub-pixel techniques, as they both utilize the same data picked up by the sensor.
- **Agree, this entire paragraph has been rephrased, see lines 114 to 125 in the revised manuscript**

L96-100. I cannot follow the logic here. First the authors mention little emphasis on spatial variation of pixel values and pixel neighborhoods, i.e. suprapixel, but provide examples that focus on the pixel internals, i.e. subpixel. Rephrase and/or explain better.
- **The reviewer has misunderstood here. We have not mentioned suprapixel, but rather "super-resolution mapping', which includes sub-pixel techniques (Atkinson (1997; 2004) and Foody (2004)). This might be too much technical detail for the reader so we have simplified it, and the whole paragraph has been thoroughly revisited and re-written. See l 110 -125 in the revised manuscript.**

L105. Exploited -> explored
- **Change made**

L112. "allow" -> could allow
- **This phrase was removed in the revised manuscript**

L124-127. This is a bit out of place here, and should be expanded and moved to discussion.
- **We agree, this was removed as it was already touched upon in the discussion.**

L126. If the goal is to transfer the method to open source software, why has the procedure been built in ENVI in the first place? Throughout the methods there are a lot of (proprietary) ENVI algorithms and tools involved, which counters this statement.
- **This phrase was removed. We had indeed intended to use the open source RSGISLIB to apply the approach over the entire Himalaya but due to changes in the implementation in this particular software, and resulting difficulties, this could not be achieved so we relied entirely on ENVI and focused on the methodology**

L134-137. David Shean's work should be added here (Shean et al., 2020)
- **We do not see the relevance of this work in this context, as here we were referring specifically to studies done over the Khumbu. However, this was cited on l.67 in the revised manuscript with regards to advances in glacier mass balance at mountain range scale.**

L139-140. Rates of change of what exactly? Area, volume, debris-cover? Clearly specify this
- **This was change in area. We have clarified this now.**

L139. Use SI throughout. % per year -> % a
- **done**

L151, L153. Quotes for A, B, C are not necessary
- **These were removed, and the letters are now integrated in the Fig.1 caption only**

L160-L161. Reads as if Landsat is considerably worse than Sentinel-2, and clearly not the first choice. Remove or rephrase.
- **This was fully revised in the new version of the manuscript to incorporate suggestions from reviewer 1 as well.**

L165. Verb should be plural
- **Verb on l 165 was already plural ('…were needed'). No change made**

L163-164. Although I understand this choice, it is rather tricky to assume that the debris surface is similar from year to year around the same time. This should be better acknowledged.
- **We agree that there are limitations, and unfortunately there was no choice in order to have a continuous area over the domain. We have acknowledged this as an additional source of uncertainty (l.179 in the revised manuscript) :**
- "We acknowledge that this choice may introduce some uncertainties due to the temporal difference, which we discuss later (section 4.6)."
  **and again in the discussion section.**

L171. Is "Pléiades 1A" the name of the satellite or the sensor?
- **It is the sensor indeed, but it is referred to in the documentation as "satellite sensor" https://www.satimagingcorp.com/satellite-sensors/pleiades-1/ Left as is.**

L181. Mentioning "Planet satellite" is a bit odd here. Furthermore, RapidEye is a constellation of five satellites.
- **We agree, RapidEye is a constellation operated by Planet. We have removed "Planet satellite" and just left "RapidEye level 3A analytic ortho tile"**

L182. What is the geodetic accuracy of this L3 ortho tile? These preprocessed products often have orthorectification issues in high relief terrain. How was this solved/accounted for?
- **We address this by assessing co-registration shifts with Landsat8 imagery using CosiCorr, as response to the major comments (see below)**

L183-185. I would stick to pure data description here and not hint at the methods already using this sentence.
- **We have mde these changes.**

L186. Did you also consider the high-resolution HMA dem? Why, why not? (Shean, 2017)
- **Yes, we did consider the HMA DEM at 8 m when testing the methods over the Khumbu. This did not make a difference in the topographic controls. Furthermore, the HMA DEM over the**

**entire domain contains considerable data gaps and it was not adequate at large scale. The HMA DEM would be suitable only at glacier scale where the debris-covered area does not contain gaps, but this is not the case for entire HMA**

L193-194. Here a whole analysis (which is introduced in the discussion) is dismissed with a single sentence. It should be properly outlined here in a separate section. Also see comments for the discussion.

- **Here we are mainly specifying the data sources and what we obtained from it, in line with Reviewer 1 comments. We have created a separate section (2.7 in the revised manuscript) for where we list the data sources used for the topographic, climatic glacier mass balance and velocity controls on pond incidence, as a response to major comments.**

L199. Remove "easily" and replace "high-mountains" with "study area"

- **Done**

L217-218. It is very tricky to assume that these parameters can simply be transferred to the other scenes that are thousands of km away and from different times of day, dates and/or years. It is, as the authors write in L208, a procedure that should be performed on an image basis. This should be better acknowledged here, and potential limitations should be clearly indicated. To my opinion, this also strongly endorses the importance of additional validation of the applied spectral unmixing results for areas outside of Khumbu (see main comments).

- **We appreciate the reviewer's concern about the uncertainties associated with the imagery. We have investigated this issue in depth, and can clarify that :**
a) **We have checked and ensured that the Landsat scenes were acquired around the same time of the day (05 UTC time +/- some minutes)**
b) **Solar azimuth varied from 138 to 156, with most scene consistently at 143 degrees**
c) **solar zenith varied from 34 to 51 with most scenes around 30 degrees**
d) **The AOT was derived individually for each scene automatically in ARCSI based on a Lookup Table**
e) **We have used the STDSREF product as it produces a standardised surface reflectance product which tries to correct for global and local viewing and solar geometries.**
f) **Tests were done in Australia for example, to find the optimal AOT value and it was found that using a fixed AOT of 0.05 estimated the fractional cover [of vegetation in that case] within 10% of their true value when the true value of AOD is less than 0.1 (Gillingham et al., 2013). We have discussed this issue with colleagues who have used the ARCIS routine extensively, including in the Himalaya, and concluded that we can generally assume that these areas have quite clean atmospheres so it is therefore not unreasonable for the AOT value to be very low (0.05) (P. Bunting, personal communication).**

**We have amended and detailed this information in the following lines:**

**L 175 – 181: "In addition, Landsat scenes across the domain were selected around the same date as much as possible to minimize seasonal differences in surface conditions, notably seasonal changes in pond occurrence (Miles et al., 2017). All chosen images were acquired around the same time of the day (05 UTC time), with similar solar azimuth (~143 degrees) and zenith angle (~30 degrees). This is important to ensure that differences in surface conditions were minimal. Where the 2015 images had too much cloud or snow, we selected images for**

**the same season in 2014 and 2016 (Table 1). We acknowledge that this choice may introduce some uncertainties due to the temporal difference, which we discuss later (section 4.6). "**

**L 208 – 208: "ARCSI allows for global and local viewing and solar geometries using physically-based illumination and reflectance corrections based on topographic data (Shepherd and Dymond, 2003), a specified atmospheric profile, an Aerosol Optical Thickness (AOT) value and sensor geometry,. These settings are important for minimizing differences in surface conditions among the various scenes."**

**L 223-225: "In the Himalaya, we can generally assume relatively clean atmospheres and thus consider that low AOT values are reasonable (P. Bunting, Aberystwyth Univ., personal communication, Feb. 2021). Our choice of a constant AOT value in high environments is in line with findings from other studies (Gillingham et al., 2013; Matta et al., 2017).**

L221. Remove "basic"
- **Done**

L229-231. Italics are not necessary here
- **Removed the italics**

L240. which -> that
- **Ok, changed**

L252. I happen to know what a MNF transform does, but the large majority of readers of TC probably do not. It should be better explained and also discussed why this is necessary. It is also not clear to me whether it was used just to determine the dimensionality, or also to reduce noise by discarding MNF bands and/or to decorrelate the OLI bands (Meer and Jong, 2000). Proper references for this procedure are also necessary.
- **Point well taken, we have now described and referenced the MNF routine in the revised version of the manuscript (section 2.5.1, l 274 – 277): ".. we performed a forward Minimum Noise Fraction Transform (Green et al., 1988), which consists in a linear transformation of the data based on principal component analysis, and is used to estimate noise in the bands. All bands had eigenvalues > 1, so we determined the dimensionality of the data as n = 7 which were used for the subsequent steps"**

L253. "Pixel purity routine", "the n-D visualizer" are very much ENVI terms and will not ring a bell with the readers. Since the endmember selection procedure is crucial for the entire analysis it should really be explained in full detail. Why were these tools used, to what effect, and what are the pros and cons. Also, it should somewhere be stated which version of ENVI was used, and whether it was ENVI classic or not.
- **Because the pixel purity index is proprietary to ENVI, we have devoted some lines to explain in detail how they work in the revised manuscript, with the adequate references and pros and cons. We have used ENVI 5.5 for the LMM and Envi Classic for running Cosi-Corr; these were indicated and referenced in the revised manuscript (see the revised section 2.5.1)**

    **"We used the unsupervised pixel purity index routine in ENVI to find "pure" pixels in an automated manner. This routine outputs a data cloud where the value of each point indicates the number of times each pixel was marked as extreme, thus representing pixels with the highest occurrence in the image. We optimized the pure pixel extraction using various**

**numbers of iterations (20,000 to 50,000) with thresholds ranging from 2 to 3 (i.e., two to three times the noise level in the data) until all "pure pixels" were detected. Larger thresholds identify more extreme pixels, but they are less likely to be "pure" endmembers. "Pure" pixels were identified on the Landsat 8 OLI scene as corresponding to six endmembers: clean ice, dry vegetation, clouds, light debris, dark debris and turbid water (**Error! Reference source not found.**). These were checked against co-registered Pléiades and RapidEye false colour composites in the Khumbu in order to minimize any occurrence of "mixed pixels".**

L259-L263. I do not understand the flow and logic between these sentences. Please restructure.
- **We have moved some of this to the discussion of uncertainties (section 4) and simplified it because some of the studies mentioned were based on hyperspectral data**

L264. Not really "areas" if it is only one pixel. Also, picking one pixel does not mean it is not a mixel. Picking one pixel "reduces the chance of a mixed spectral signature in the region of interest of each endmember".
- **This was revised and re-written and now reads: (l 273 – 275): "These were checked against co-registered Pléiades and RapidEye false colour composites in the Khumbu in order to minimize any occurrence of "mixed pixels"**

L265. How do you account for spatial discrepancies between the OLI, Pléiades and RapidEye data? I have not read anything with respect to co-registration of the di erent scenes. In such a multi-sensor study co-registration is a crucial component of preprocessing, since otherwise it is not guaranteed that the images line up correctly. This would greatly impact the endmember selection and validation procedures and could undermine the entire study. Even after co-registration there will be errors that should be considered and acknowledged.
- **At the suggestion of both reviewers, we now show the co-registration of Pleiades, RapidEye and Landsat using Cosi-corr in the revised version of the manuscript. The initial x,y shifts between the images were less than 1 m, less than the spatial resolution of Pléiades and RapidEye. This was reduced further after the co-registration.**
- **We summarize this in the methods section (2.2):**
  **" We co-registered all high-resolution images and the corresponding Landsat 8 OLI images using the Co-registration of Optically Sensed Images and Correlation (COSI-Corr) routine (Leprince et al., 2007) implemented in ENVI 5.5 Classic (L3Harris Geospatial, Boulder CO). For the Pléiades image, after co-registration with 20 tie points and a second-order polynomial transformation (RMSE = 1.3 m), image displacements were -0.16 m in the E/W direction and 0.12 m in the N/S direction. The Planet RapidEeye and PlantScope scenes, were co-registered with 15 and 10 tie points (RMSE = 5 m and 1.6 m, respectively), yielding shifts after co-registration of ~1.1-1.7 in the E/W direction and 0.09-0.5 in the N/S direction. These offsets were below the spatial resolution of all scenes (2 - 5 m)."**

L265. "false colour composites". Also, the band numbers are used often for all sensors, but they are not defined anywhere. Please add the bands, their no. and their spectral characteristics, e.g. wavelength and bandwidth/FWHM, to the dedicated table (Table 1).
- **done**

L271-272. From my experience, turbid water can (at least in VIS) still look quite different from pond to pond, depending on the type of suspended sediment. From blueish (glacial silt) to reddish. How is that accounted for?

- **We are aware of this from our own field observations (see Fig. 3 in the revised version) and this was nicely described in (Matta et al., 2017). Picking a single pond type was a difficult choice because we were limited to the spectral resolution of Landsat, which does not allow capturing the various spectrum of turbidity. We defined our water endmember as "grey-bluish", based our choice on visual observations and outcome of Matta et al 2017 study over the entire Himalaya. They showed that 45 % of the ponds in Nepal are largely grey, with 27% blue (27%) and turquoise (25%) and 3% were classified with mixed colours. Over the entire Himalaya they report 52% of ponds to have grey waters and 24% with mainly blue waters, etc. We are aware that this will not capture the variability of the spectra as showed in the Matta et al (2017) paper based on field studies, and have discussed this as at length as a limitation of our study in the discussion section 4.3).**

L287. "area" -> "an area"
- **Done**

L310. What is meant by "finer classification map"? What resolution, how was this done, to what purpose, how does this affect the analysis? This is crucial information that should be explained in detail. Also I think "map" can be removed as just "finer classification" surfaces. For me, map has the connotation of being a spatial display of something with the primary purpose being presentation.
- **This was rephrased to (l 313 – 314) : "For further analysis, we require maps of the surfaces rather than just a numerical value of area, so we classified the 30 m fractional maps by applying a threshold α to produce binary maps for each class".**

L313. "from the Khumbu" -> "that were derived for the Khumbu domain"
- **Correction made**

L313. I am not sure whether this strictly falls under upscaling, since the spatial support (i.e. Landsat pixel scale) remains the same. What about applying/extending/extrapolating/inferring?
- **We agree, and have chosen "inferred". However, these phrases were removed from this section and incorporated elsewhere (l 132 -133 in the introduciton)**

L313. Composition does not seem the right word here. Classification?
- **We use the word "classification" for the algorithm but we aim here (as expressed in the tile) at quantifying what type of materials the DC tongues are composed of. Left as it is.**

L316-317. I find this too much detail. When something can be reproduced similarly in a plethora of ways and di erent software packages, it is not about the tools for the job, but purely about the method and approach. Also, the Python module of ArcGIS is called ArcPy, not ArcPython. And strictly speaking it would be simply Python scripting using the ArcPy module to invoke ArcGIS functionality.
- **Agreed - we have removed this. We mention arcPy now in the Code availability section.**

L323. How is this iterative procedure performed exactly? How do you select new endmembers. Using the n-dimensional visualizer, or the PPI, or something else?
- **This is now more thoroughly explained in section 2.5.1. We did not select new endmembers but rather chose the pixels which we considered pure as well as tried various combinations of the classes from 3 to 6 classes, and various iterations of the PPI. Please refer to section 2.5.1**

L324. These are not a lot of ground truthing points, to be honest. It is also important to know how these points were determined. It is somewhat vaguely stated that these are "well-distributed on several tongues", but it is not clear how the points were generated/identified. To obtain a fair classification accuracy measure it is crucial that the validation points are not manually digitized, but randomly selected within the entire domain of the Pléiades/RapidEye images. To get a (more) even number of points among classes that strongly vary in size, a stratified random sample should be taken. This section requires more clarity about the exact procedures used to perform the accuracy analysis.

- **We agree that these are not "ground truth" to be precise, but we take them to represent ground truth based on the high-resolution data. Ideally we would like to have GPS points from the field, perhaps for a future study.**
- **While this can be improved using a stratified random sample, we chose to focus validating the results at other sites, and conducing and in-depth analysis of the topographic, climatic, mass balance controls etc. on pond occurrence. We note that other studies (e.g., Anderson et al 2020) used 93 (vegetated /non vegetated) pixels, for example. We consider that validating with high resolution data at several sites is more important here and we focused on this aspect.**

L328. OBIA is mentioned before, but never properly referenced using for example (Blaschke et al., 2014). I also find the description of the OBIA procedure to be quite lacking.What settings were used exactly? How did the image segmentation work? Was there any postprocessing done on the objects, e.g., splitting/merging? How were the lakes classified, manually or automatically using a decision tree approach? What was the accuracy of the OBIA classification? Without this information it is impossible for the reader to estimate the validity of the derived data for validation purposes.

- **We referenced and have expanded this in the revised manuscript. See section 2.6 in the revised manuscript.**

L333. Remove "might have occurred"
- **Done**

L336-353. It is not completely clear to me why the SAM procedure was included, since the remainder of the manuscript focuses almost solely on LMM results. This paragraph mentions that the SAM results were used to test endmember choices, but it is not clear how this is done. (And this should be included in the methods, not the results section). I would suggest to expand this section and clearly describe to what purpose it was implemented, or remove the SAM entirely from the manuscript if that does not compromise other parts of the study.

- **We agree and have removed the SAM analysis entirely**

L351. Is that is -> is that it
- **Corrected**

L362. Abbreviation for root mean square error should be RMSE, not RMS.
- **Changed made throughout manuscript**

L361-363. Why did it have lower average RMSE? Was this due to a specific class mainly, or overall. How was the class-by-class performance difference? Maybe the average worse performer, performed better in more 'important' classes? Please elaborate,

- **We have addressed this in the revised accuracy assessment (section 3.2 in the results section of the revised manuscript). We now present class metrica, ie. using the class-by-class precision, recall and Fscore metrics, described in detail in the methods (section 2.5.3)**

L363-365. Two times roughly same sentence here.
- **This section has been revised and the duplicates were removed.**

L372-376. I find most of this to be more fitting for the methods section. Also, how were these seemingly arbitrary threshold values determined?
- **For the ponds, thresholds were chosen using validation against our OBIA outlines from Pléiades, and the area of Khumbu Glacier only from Watson et al.2017 study.**
- **For all other classes, thresholds were chosen by using visual comparisons against high resolution imagery. We also followed other studies, which have used the 50% threshold (i.e., if there is greater than 50% water coverage this is classified as water). For the other classes, the threshold was lower (i.e., a pixel covered by as little as 30% debris was classified as debris). We acknowledge there is some subjectivity in this, and we acknowledge it in a new discussion section 4.4 in the revised manuscript.**

L372-376. Since multiple classes can be attributed to the same pixel using this multi-step thresholding of the fractional results, the order in which these threshold classifications are combined into a final product matter. That is, what will be the final class of a pixel when it falls within the thresholds for multiple fractional layers? It is not clear to me how this is done exactly.
- **Unfortunately, when we used 6 classes to unmix the pixel, a simple standard threshold did not work, so we had to rely on manual interpretation. We used mutually exclusive conditions so a pixel could not be assigned to the same class. We have clarified this in the text and discussed the limitations in section 4.4.**

L380. A remote sensing classification accuracy of 75% is frankly quite low (see e.g. Foody, 2008; Foody and Atkinson, 2002). For me, it really gives rise to the thought how other classification procedures might fare on the same data. Would a simple minimum distance supervised classification perform better or worse? Is it really beneficial to use this technique? Since the accuracies are low, particularly for the debris classes, I expect a very thorough and complete discussion about the limitations and capabilities of the method in comparison with possible other classification approaches.
- **We clarified in the revised text that the overall accuracy is not the best metric because this is it lumps all classes, and includes the poorly identified classes such as clouds and ice. While the overall accuracy of the model is 75% accuracy is not ideal, the accuracy is higher for the water and vegetation classes (> 80 % recall, >90% precision), which are better accuracy metrics. One of the scope of this study is also to assess the performance of LMM for mapping individual classes, and not to assess the LMM as a whole.**
- **Lower accuracies ~ 60 -70 % have been reported in more sophisticated deep learning approaches (Robson et al., 2020). There is certainly room for improvement, but we attribute the overall accuracy of the model to the inherent complexity of the debris cover system coupled with uncertainties in the parameterization of the LMM, and we have marked this as such.**
- **While other approaches such as NDWI for mapping these ponds might be envisioned, we stress here that in this paper we aimed at understanding roughly the composition of the glaciers in one algorithm, and if other methods do exist, such as supervised classification, they**

**have surprisingly not been used to achieve mapping of the supraglacial features at regional scales.**

L385-386. I do understand the argument that there is a link between the occurrence of a class and the classification accuracy.
- **This was phrased awkwardly; we have removed this.**

L387. If it is heavily dusted, then it is not clear ice, right? That is, exposed ice != clean ice
- **We agree, phrasing was confusing. This entire section has been re-written to make it easier to follow, also at suggestion of reviewer 1. We had initially hoped to use the clean ice endmember to identify pixels which might contain ice cliffs. Some of these ice cliffs can be are "exposed ice" at the end of the ablation season (as correctly pointed out as well by reviewer 1 as well; others are dusted with debris. In order to capture this variability, we would have needed to define a clean ice and a dirty ice end member. We did explore this option as well (results not presented here), but due to the spectral similarity between some dark debris classes and dirty ice, this led to mixing and results were inconclusive. Therefore, we have chosen the clean ice member because there were areas at the upper limit of the DC which were included in the SDC, and we wanted to identify these as validation of the performance of the LMM for clean ice.**

L400-404. It might be good to add uncertainty ranges to these percentages, given the moderate classification accuracies.
- **We agree this would be desirable but we cannot estimate the uncertainties (+/-)**

L415-416. "5.6% of the debris area". This is quite arbitrary because this number completely depends on the quality of the SDC dataset.
- **We report all the percentages with respect to the areas based on the SDC. If there are biases, then the biases are the same since we refer to the same area. The uncertainty here lies rather in the poor identification of the clean ice. It just indicates that some clean ice does exist within the SDC outlines, which we confirmed by visual inspection.**

L420-421. If clouds are not present in the validation region, how were you able to assess the accuracy and confidently extend that to the other landsat scenes?
- **Clouds were not present within the Pleiades scene, but are present in small amounts on some of the glaciers in the eastern part of the Landsat scene visually. They can be detected using a band 5 threshold (not presented here). We do not overinterpret this because clouds were poorly identified in the LMM.**

L424-426. There are several of these climatic 'speculations' in the manuscript, which could be substantiated by including some climate data (see also main comments)
- **This was addressed with the analysis of the climate data (see the new Discussion section 4.1)**

L425. Of course it has to do with climate to some degree, but satellite images are snapshots and there is just a degree of luck involved regarding cloud cover. I would suggest not to over-analyze this.
- **We agree, the cloud mapping is not robust in this study so we have toned this down.**

L444. "latter two" is a bit odd here, since after the two that is being referred to there are other things still mentioned. Suggest rephrase.

- **Replaced with "Ngozumpa and Khumbu"**

L449. "OBIA image segmentation". I would use either OBIA or image segmentation. This depends on whether you manually assigned the object to the water class or performed an automated procedure (an OBIA), which is not clear from the methods.
- **We have checked throughout the manuscript that we are only using "OBIA". Also, we have added a more detailed description of how the mapping was performed fir the pond validation, see section 2.6**

L454-455. Not sure whether it is fair to compare a water classification to a snow classification.
- **We are not comparing the surfaces, but the classification type. The methods can apply to any class, be it vegetation, snow etc (Rittger et al., 2013). We have clarified this.**

L459. Overestimated is one word.
- **Correction made**

L460. I am a bit puzzled by the binary pond area. Wouldn't one of the benefits of having fractional subpixel information be that one could do analysis using those fractional values. First converting them to binary information seems to undo that. Is this then really better/di erent than a supervised classification or NDWI thresholding approach?
- **As we mentioned in the text (l 314), in this paper we focus on classification rather than the precise quantification of the lakes, due to the limitations we are now describing in detail in this version. We needed the actual location of the lakes as well, in addition to the areas, which is also useful for the type of analysis that both reviewers suggested. In a multi-temporal study focusing merely on % lake coverage, a fully automated fractional approach would be desirable- however this was not our purpose in this study.**
- **Methods used to map lakes fall mainly into two categories: ''binary'' (i.e., water or not-water) on normalized indices, such as Normalized Difference Water Index (NDWI) and empirical methods, used in current global (Shugar et al., 2020) or local-regional lake inventories (Watson et al., 2018; Wang et al., 2020); and "fractional" maps based on spectral unmixing (Painter et al., 2009). These were nicely summarized and compared by Rittger et al., 2013 in the context of snow mapping.**
- **In the binary algorithm, generally, pixels covered with greater than 50 % of a specific type of surface (e.g vegetation, rock etc.) are assigned to that class (Hall, 2002). In this study we use a fractional approach, but we use these maps for classification purposes as in Hall et al, 2002 to obtain maps of each surface – but the method would still fall under "fractional". Using the term" binary" introduced confusion so we have revised this and we are only reporting "fractional" with a threshold.**

L461. OBIA analysis = object-based image analysis analysis
- **Corrected**

L467. "Good agreement" is subjective, needs to be quantified.
- **this was rephrased/re-written (see section 3.3)**

L478. Maybe add a line or two that helps to substantiate this presumption?
- **We have rephrased this and shifted it to the discussion section 4.3 l 648 – 650: "Our choice of debris endmembers was limited to "light" and "dark" debris, and we are aware that**

**these most likely do not cover the wide spectrum of lithology present across the Himalaya**

L488. I have seen snow patches on the debris in spring in the field and on satellite imagery, but not in the early post-monsoon period. I am not saying it is impossible, I only find it quite unlikely. Isn't it clear from the rest of the Landsat scenes whether there are snow patches or not?

- **There is some snow in the upper part of the debris cover in the extreme end of the Himalaya (Bhutan) where the end of the monsoon is later (Oct-Nov). Other than this, yes snow melts sooner on the debris cover, so we do not see large snow patches.**

L500-512. As mentioned before, it would be a great addition to the manuscript to include climate data to really quantify this climate dependency instead of providing only speculation.

- **This was addressed as a reponse to general comments, see discussion section 4.1 which include the analysis of climate patterns.**

L511-512. As mentioned before, it would be a great addition to the manuscript to include data of glacier mass balance (Brun et al., 2017; Shean, 2017) and velocity (Dehecq et al., 2019) to substantiate these hypotheses.

- **Idem; however, we updated this to Shean et al. 2020 which is as far as we know a better spatially resolved mass balance dataset**

L507. "Less glacier shrinkage" over what time period?

- **This section has been changed and has been shifted to the discussion section in relation to the thickness change data from Shean et al. 2020.**

L511. Reference?

- **The previously mentioned phrase on l 511 related to the citations in the previous one (Sakai and Fujita, 2010; Reynolds, 2000), to which we also added (Quincey et al., 2007) – but the phrase was a repetition of the one above so we combined them  in l 519 – 520:**
  **"Supraglacial ponds have been shown to form on stagnating areas on the ablation areas of glaciers with negative mass balance and surface angles lower than 2° (Sakai and Fujita, 2010; Reynolds, 2000; Quincey et al., 2007). "**

L522. Reference?

- **We have cited this paper : (Vezzola et al., 2016)**.

L524-566. I find it very odd to only introduce this analysis here, in the discussion. Although it is not part of the remote sensing and unmixing methodology, the methods used here should be added to a dedicated methods section and the results to a dedicated results section. I am not completely opposed to introducing figures in the discussion section, but introducing three new figures with results there is a bit odd. I would suggest to carefully reconsider the discussion and put any methods/results related parts in the correct sections.

- **The description of methods has been moved up and expanded as section 2.6 in the revised manuscript;**
- **The results and the figure related to it (now Fig 11 and Fig 12) were moved to the corresponding Results section (3.6).**

L531. What constitutes a debris cover glacier tongue in this case? How does removing small tongues help to remove bare land patches. This part requires clarification. Also, 1 km2 is not big, but certainly not very small: 79566 of 95537 glaciers in Asia are smaller than 1 km2.

- **As noted in other studies (Brun et al., 2019), there is no standard definition of a debris-covered glacier is in terms of the % debris cover. The cited study used a threshold of 19% to separate the debris cover, based on 100 randomly selected glaciers. In the SDC dataset, some of the debris covered pixels are not actual DC tongues but isolated rock debris or just bare land, so applying an area threshold was essential prior to performing the analysis. We agree with reviewer that a lot of glaciers in the Himalaya are small, but this argument holds for clean glaciers, not debris covered glaciers. In a previous paper (Racoviteanu et al., 2015) we have shown glaciers in the Himalaya averaging ~ 1 km2 are mostly clean glaciers, compared to debris-covered glaciers which average 15 km2 in that particular region. We feel that our area threshold is justified and is not over restrictive.**
- **However, in the current manuscript we focused on the elevation and gid by grid analysis rather than on the glacier-by-glacier analysis**
- **Furthermore, most studies conducted over the entire HMA have adopted the approach, for example Brun et al., (2017) restricted their analysis to all glaciers > 2 km$^{2.}$**

L532. So larger glacier tongues have more turbid supraglacial ponds?
- **This has been shown in Matta at al. (2017) paper but we have removed this statement and discussed the issue of turbidity in the context of limitations in section 4.2.**

L538. "For ex."? Why not just the broadly accepted "e.g."
- **Changed**

L542. I am not very surprised that average glacier values do not show strong correlations since the supraglacial pond density is highly variable over a single glacier. It would probably be better to look at elevation bands, as other studies have also done (e.g. Ragettli et al., 2016).
- **See our reply to general comments #3; we have performed a thorough analysis by elevation, slope and aspect bins as well as averages by 1 x1 degree grid. See the corresponding discussion section 4.1**

L551. Quantify "in general"
- **Removed (we now use numbers)**

L551. Seems more than 20% on the figure.
- **This does not apply anymore as this figure has been removed from the revised manuscript**

L556. Again, what is meant by "in general"
- **removed and quantified**

L559-561. I do not find this surprising: (i) looking at the scatter plots I highly doubt whether the assumptions that are made for linear regression are valid here, (ii) the signal is strongly subdued by looking at glacier-average values. Other machine learning approaches that can robustly deal with non-linearity might work better here, e.g. Random Forest.
- **As we noted before, we consider random forests to be potentially promising, but beyond the scope of the current paper, which is already quite substantial in length. Linear regressions are**

**still standard and widely used methods. A full analysis with machine learning may be done in a future study.**

L562-566. I am not sure whether Figure 12 and this small description add much to the analysis in its current state.

- **We have revised and kept this as Figure 11 in the revised manuscript as it shows the glacier by glacier % pond and vegetation, and we use this figure as support to further discuss trends by elevation bands and by grid (section 4.1).**

L524-566. Overall, I find this analysis quite lacking in rigour and novelty. With a few adaptations I think a much more interesting and valuable analysis can be performed (see main comments).

- **We have addressed this in response to general comments and have performed a thorough analysis using climate, velocity etc…(see section 4.1)**

L574-575. It should be acknowledged here that the lake turbidity is temporally highly variable and, also given the uncertainties of the classification method, the satellite snapshots might therefore be di cult to use for this purpose. Spatial accuracy of the Landsat OLI data will also be a concern, as from acquisition to acquisition the pixels will be slightly misaligned, resulting in potentially very different 'mixel' compositions and unmixing results. This effect will be particularly strong for the relatively small ponds that are almost always adjacent to the spectrally very different debris pixels. This argument of course not only applies in this case, but also for the applicability of the entire approach with respect to multitemporal analyses. These limitations should be clearly stated and discussed in the discussion section.

- **Thanks for these remarks, which we have now incorporated in the revised discussion section 4, specifically 4.2 with respects to spectral limitations of Landsat with respect to variability in turbidity, and section 4.6 with respect to transferability of the methods.**

L586. "outperforming". I do not think that purely based on visual inspection of a 5 x 5 km subset of one of the major glacier tongues in the validation region of this study, which is a minute subset of the entire dataset, one can draw the conclusion that this method outperforms the other approaches. To make such claims there has to be some level of quantification and an assessment of much larger area.

- **We agree that a larger scale accuracy /comparison has not been performed.**
- **By "outperforms" we meant that our approach detects supraglacial ponds while in the cited datasets have substantial omissions errors, and in some cases patches of ice were classified as ponds (this is based on qualitative visual comparison by and author). We have revised the phrasing so that it is not misleading to the reader.**

L599. To my opinion, automated scalability to large regions is also an important limitation to consider.

- **We agree and we want to clarify that this can be possible when the approach will be embedded in open-source software. We refer to this in the Summary and further work (section 5, lines 780 - 787):**
  **"Future developments to overcome the current limitations of this study include the use of more sophisticated non-linear mixing models, which would allow to discriminate materials of interest in more detail. Work is ongoing to make the unmixing step approach fully automated by integrating it within Python-based routines (e.g. Bunting et al., 2014), so that it can be applied in the future to derive**

**supraglacial pond outlines at multi-temporal scales and monitor pond development over time**"

L603-604. This gives the impression that it would be simple to transfer the unmixing parameters to the entire Landsat archive. This is not true because of differences that exist between sensors and bands, even though sometimes these are small: MSS != TM != ETM+ != OLI. For each sensor separate endmember selection will have to be performed and for older images this will not be trivial, given the lack of high-res calibration/validation data. I am not saying it is impossible, but these lines should be honest about the ease of transferability and the application of the method to historical imagery.

- **To clarify, we did not mean that the same bands are used for older sensors ; we did state in our methods that Landsat 8 is superior to the older missions in terms of spectral band width, calibration, geometry and radiometric resolution.**
- **We do acknowledge that various solar illumination in complex terrain result in widely varying spectral responses which affect image processing operations, including sub-pixel routines – and we try to mitigate that by performing the topographic corrections. We do not expect Landsat 5 and 7 to perform exactly the same, but we remind the Reviewer that sub-pixel methods were originally developed for Landsat TM. We believe that limitations in the software used outweigh the limitations of the particular sensor.**
- **We agree that applying this method requires calibration depending on the sensor; however, we do not believe that totally different endmembers should be selected for older images, because the surfaces within a DC glacier would be the same. While the spectral characteristics of the surfaces will differ, their spectral curves do not change, i.e. vegetation has a particular spectrum, etc. Here we presented a proof-of-concept method.**

L608. As mentioned before, I would like to see this confidence validated for a region outside the Khumbu with additional high-res imagery.

- **Addressed ; see our replies to general comments, we have validated the method for pond areas in 2 additional areas (one using our own mapping on PlanetScope and the other by comparison with pond areas based on Spot7 from Steiner et al, 2019).**

L611. What is meant by "some post-classification corrections"? How will these be determined without validation?

- **This depends of course on validation, but corrections are also very often done manually using visual interpretation, this is often the case for example for debris-covered glacier mapping which the reviewer is probably familiar with.**

L626. I have not read this before and couldn't find it. I was under the impression that only turbid water was considered as endmember. Again, be strict about separating methods, results and discussion and do not introduce new methods in the discussion.

- **We have revised the manuscript and made these changes, and the results are now presented in the Results section; only 1 figure is now introduced in the Discussion section (Fig. 13) to support our discussion on controls on pond and vegetation incidence.**

L632-635. I cannot follow the logic here. Please rephrase.

- **The SAM analysis was removed and the phrase was revised.**

L636. reference for the bad performance?

- **This was based on our testing of FMask algorithm embedded in ARCSI prior to using the Landsat 8 images as well as a stand-alone tool. It has been shown (Stillinger et al., 2019) that cloud-masking algorithms such as FMask suffer from confusion between cloud/ice/snow/other surfaces.**

L650-652. Successfully applied but not validated on accuracy.
- **This was addressed ins response to general comments, we have now validated the pond mapping at a total of 3 sites in the revised manuscript**

L654. Important to mention, though, is that commercial high-res imagery was required for proper endmember selection. Also, I think detail alone is not the sole criterion on which performance should be assessed. Usability, scalability, ease of use, speed of implementation are all factors to consider.
- **We agree that all these metrics need to be considered in the performance of an algorithm. While we used commercial satellite imagery to check the end member selection, this step is not necessarily needed. The endmembers were chosen from the Landsat image itself, which is also a standardized procedure (as described in Keshava and Mustard 2002, etc..) ; the use of high resolution was an extra step we have undertaken. This can also be achieved using field data, or any high-resolution image including those available freely from Planet (as we have done for the areas outside the Pléiades extent). In other words, the choice of endmembers did not depend on commercial satellites -it was an option, and we chose to use in this study.**

L657-659. I don't think this was confidently demonstrated in this study. Also what is meant by historical and more recent here? All images that were used are from ~2015.
- **This was removed as this entire section was re-written. We have removed the term "historical and more recent", which perhaps created confusion**

L658 "imagers" -> "images"
- **fixed**

L660-662. Yes, this seem to be true. But would just calculating a long-term NDVI composite and thresholding based on that not results in a much simpler approach that is as effective?
- **This is a justified question. While time series of NDVI based on Landsat was used in a recent study (Anderson et al., 2020) to map subnival vegetation across the Himalaya, the cited study also needed to use a threshold to map the vegetated areas, and did not provide maps of supraglacial vegetation but rather they focused on the greenness index. The accuracy of the vegetation mapped with NDVI using Landsat imagery in GEE in the cited study was 79.6% so we do not believe that NDVI would be superior to the sub-pixel analysis. In this study we sought to explore the suitability of subpixel analysis to classify supraglacial vegetation at the same time as supraglacial ponds etc. in a single algorithm.**

L665-666. Rephrase sentence, grammar incorrect.
- **This was removed as the context was incorporated elsewhere (in the results)**

L675. "other python-based routines". Remove "other", the ENVI approach was not a Python one. Capitalize "Python", it is a name. Why just Python-based routines, it can probably be achieved using various programming languages? I would suggest to change this to "routines using open source software"
- **Thank you for this suggestion, change made to "Python-based routines"**

L685. Complement -> to complement
- **We did not change this. Both formulations are correct : "help complement" or "help to complement" and we prefer the former which is more commonly used ("to" is optional)**

L687-688. I find this an odd last sentence. Would fit better somewhere in the introduction.
- **Agreed, this has been moved and re-worked in the Introduction, and the last phrases have been re-worded**

L697. "in ArcPython" -> "using the Python module ArcPy from ESRI ArcGIS"
- **Change made**

Fig. 3. It would be good to map use the actual wavelengths on the x-axis for panel A.
- **We included the wavelengths in Table 1. Left as is sicne sometimes we refer to Landsat bands in the text of the manuscript.**

Fig. 5. Both my printout and zoomed-in PDF have too low resolution of the grids, and details are not visible. Font size on the legend is also very small. Would be better to convert it into a full page 3x2 format.
- **The following changes were made : the legend has been redone (only one legend for all), only one scale bar, so the text is bigger.**

Fig. 6. Similar comment as for fig 5. The small details are not discernable due to resolution/size issues.
- **The figure has been redone to fix this**

Fig. 7a. What are the white blobs on the eastern moraine?
- **These are areas of topographic shadow, which were masked out (optionally) as output from ARCSI so these can be removed from the unmixing. We have added this in the legend and also clarified that topographic shadows were not included as a class.**

Fig. 9. The legend mentions transparent Pléiades outlines, but these details are not visible without zooming in a few 100%. Illegible on my (not bad) printout.
- **The figure has been redone with slightly different colour scheme and thicker lines to address this.**

Fig. 9, L1169-1170. This is something for the results section, not for a figure legend.
- **We moved this to the results**

Fig. 10. This figure does not add much to the analysis, in my opinion, and could easily be combined with figure 11.
- **This has been removed**

**References added and/or mentioned in the revised manuscript**

Anderson, K., Fawcett, D., Cugulliere, A., Benford, S., Jones, D., and Leng, R.: Vegetation expansion in the subnival Hindu Kush Himalaya, Global Change Biology, 26, 1608-1625, https://doi.org/10.1111/gcb.14919, 2020.

[revised manuscript text omitted]

---

## Referee Report (RR1)

**General Comments**

The revised manuscript by A. Racoviteanu and co-authors 'Surface composition of debris-covered glaciers across the Himalaya using linear spectral unmixing and Landsat 8 OLI imagery' has improved a lot from its original version and I would like to congratulate the authors for all the work they put in, one can tell that this was no minor undertaking. The resulting manuscript is very nice and interesting to read, and the results come out much stronger than in the previous version. The authors have addressed all my comments, and most importantly the more general ones, in a very rigorous way. I am now convinced of the value and robustness of the method and results that the authors present.

The manuscript has undergone major changes and I still have a number of minor comments/suggestions to improve its readability. Line numbers indicated correspond to the revised manuscript (not the track-changes document):

**Title**

L2: 'and' -> 'applied to' or 'of'

**Abstract**

L22: 'eastern' -> 'central' according to Fig. 1

L25: Would be nice to specify here the proportion of debris-covered glaciers as it is the focus of the study

L26-27: Numbers do not add up to 100%

**Introduction**

L68: 'been' repeated

L79-85: While this is very interesting, I am not sure that it is very relevant here. I would suggest not mentioning it here but maintaining it in the discussion.

L83: 'a revised dataset'

L104: These studies are not really 'more recent' but rather just looking at smaller domains with finer resolution

L109: Zhang et al., 2021 (**https://doi.org/10.3390/rs13071313**) possibly relevant here.

L136: 'vegetation the mountain range' - something missing here

**Data sources and methods**

L166-167: 'selected' repeated twice

L191: Figure number missing

L197: suggest specifying 'This RapidEye scene consists of orthorectified…'

L217-218: 'numerical inversion *of* the surface'

L238: suggest specifying 'In this study the outlines in the SDC dataset…'

L244: remove 'see' before Delafontaine et al., 2009

L258: 'therefore' appears twice

L272: Sections 2.5.1 is well written and detailed. Very nice to read.

L317: refer here to section 3.1 for the actual values of these thresholds

L326-327: Unless I am mistaken, no need to spell the acronym of RMSE here (done before)

L339: 2TP + *FP* + FN

L340-342: the difference between recall and precision is not very clear here.

L347: specify that this is 6 to 7 glacier *for each site*

L358-360: Very good that you specify this here. In my experience, it is actually quite difficult to fully automate an OBIA approach…

L375 topo-climatic

**Results**

L379: I think here it is still missing that pixels which satisfy 2 different thresholds are categorized as 'unclassified'

L444: Add name of the lakes you refer to in corresponding figure

L475: patterns

L484: I would rather suspect (from experience of fieldwork in the region in September/October) that this is due to early snowfalls rather than late ablation season

L487-490: this belongs to the discussion

L495-498: discussion

L509-512: These are interesting results in their own way – would actually be interesting to see. You could consider adding a supplementary figure for this?

L512-515: discussion

L519, 525: Show this exponential decay in the corresponding figure along with its coefficients

L520-521: Discussion

L523-524: 'slope gradient' is wrong here, it should be slope as the gradient of the slope is actually the second derivative of the topography. There are several occurrences of this in the text, make sure to correct all of them.

L530: specify '*glacier* aspect'. Remove 'slope'.

L532-536: this is mostly discussion

**Discussion**

L539: A lot of this paragraph, as well as figure 13 should appear in the results. This will need to be reformulated.

L554: I do not understand this as eastern Tibet is actually in the south of the range, especially when considering a 1x1° grid

L594-595: These trends appear to be very small and I am wondering if they are really relevant…

L603-604: No need for such a justification, suggest removing this whole sentence.

L615-620: This makes sense but am not sure it is appropriate here. It reads more like a 'response to reviewers'. Suggest removing to stay concise.

L630: 'Landsat' repeated.

L639: '*square* meters'

L663: 'we are aware …' comes back several times in your discussion and I personally do not like it. I would suggest remaining objective and removing it.

L668: Here you mention that your approach can help track the changes in lake turbidity. I would also insist on the fact that it can help track the changes in pond area, which is also a great outcome.

L696: '*square* meters'

L696: some of these ice sails are actually probably large enough to encompass several Landsat pixels and could explain some of the bare ice patches.

L728: Suggest simplifying title to 'Wider applicability of the method'.

L729: 'demonstrated'

L736: can -> could

L736: remove 'we acknowledge that'

**Tables**

Table1: add 'Band 1' for the Landsat 8 OLI bands

**Figures**

Figure 1: What are the plain lines? Add reference for these regional/climatic outlines

Figure 2: Turquoise over red is difficult to see and could be a problem for color-blind readers. Nice that you shoed the ground truth points. I was wondering if there would be a way to show the different classes? It would be interesting to have a zoomed-in view of one of the glaciers to show these different classes.

Figure 6: It is hard to distinguish the LMM ponds from the OBIA ponds. Suggest taking different colors

Figure 12: This figure could be very much improved. It is currently very raw and simplistic. There is space to add additional information. For example, I suggest showing the exponential decay you mention by

fitting and exponential to your points. Since we are talking of bins, wouldn't horizontal lines or bars make more sense than points? For c), specify in figure that this is glacier aspect.

Figure 13: It is difficult to see any correlation with such plots. It could be worth increasing the size of the dots in a first step. However, if you really want to show a correlation, it would be interesting to do so in an x-y plot.

---

## Author Response (AR2)

**Reviewer 1 General Comments**

The revised manuscript by A. Racoviteanu and co-authors 'Surface composition of debris-covered glaciers across the Himalaya using linear spectral unmixing and Landsat 8 OLI imagery' has improved a lot from its original version and I would like to congratulate the authors for all the work they put in, one can tell that this was no minor undertaking. The resulting manuscript is very nice and interesting to read, and the results come out much stronger than in the previous version. The authors have addressed all my comments, and most importantly the more general ones, in a very rigorous way. I am now convinced of the value and robustness of the method and results that the authors present.

The manuscript has undergone major changes and I still have a number of minor comments/suggestions to improve its readability. Line numbers indicated correspond to the revised manuscript (not the track-changes document):

**Thank you very much for another thorough read of the manuscript and for the final comments and suggestions. These are very much appreciated and will improve the manuscript further.**

**Title**
L2: 'and' -> 'applied to' or 'of'
*Agreed, we have replaced "and" with "of"*

**Abstract**
L22: 'eastern' -> 'central' according to Fig. 1
*Correct. This hadn't been updated. It is corrected now.*

L25: Would be nice to specify here the proportion of debris-covered glaciers as it is the focus of the study
*True. "glacierized" should be "debris-covered area", as per table 5 – somehow this got changed to glacierized. This was corrected and the number matched as per Table 6. We did not work with full glacierized areas.*

L26-27: Numbers do not add up to 100%
*Thanks for pointing this out. The number for dark debris got mixed up in the multiple rounds of edits, it was 23.8%, not 12.8% as per Table 6. This has been corrected and they add up to 100.*

**Introduction**
L68: 'been' repeated
*Repeated word deleted*

L79-85: While this is very interesting, I am not sure that it is very relevant here. I would suggest not mentioning it here but maintaining it in the discussion.
*We consider that this frames our study as it points out strengths and weaknesses in current datasets, so we prefer to retain this here.*

L83: 'a revised dataset'
*"a" added.*

L104: These studies are not really 'more recent' but rather just looking at smaller domains with finer resolution
*We replaced "more recent" with "other"*

L109: Zhang et al., 2021 (**https://doi.org/10.3390/rs13071313**) possibly relevant here.
*Thank you. This was added and the phrase was slightly re-written since it only pertained to SAR and OBIA or machine learning. It now reads: "Synthetic Aperture Radar overcomes the limitations of optical remote sensing in areas with frequent cloud cover (i.e., the eastern Himalaya), and has been used to map supraglacial ponds and track their dynamics (e.g. Strozzi et al., 2012; Wangchuk and Bolch, 2020; Zhang et al., 2021)."*

L136: 'vegetation the mountain range' - something missing here
*"across" was missing. We have added it*

**Data sources and methods**

L166-167: 'selected' repeated twice
*Corrected*

L191: Figure number missing
*Added*

L197: suggest specifying 'This RapidEye scene consists of orthorectified…'
*Added*

L217-218: 'numerical inversion *of* the surface'
*Added*

L238: suggest specifying 'In this study the outlines in the SDC dataset…'
*Added*

L244: remove 'see' before Delafontaine et al., 2009
*Done*

L258: 'therefore' appears twice
*Corrected*

L272: Sections 2.5.1 is well written and detailed. Very nice to read.
*Thank you for this comment!*

L317: refer here to section 3.1 for the actual values of these thresholds
*Done*

L326-327: Unless I am mistaken, no need to spell the acronym of RMSE here (done before)
*We had first mentioned it on l 210 but we had not spelled it so we moved it there*

L339: 2TP + *FP* + FN
*Thank you for spotting this! This was indeed a typo, and has been corrected*

L340-342: the difference between recall and precision is not very clear here.
*For "recall", we have added ", i.e the percentage of results correctly classified by the algorithm" and it should be clearer now.*

L347: specify that this is 6 to 7 glacier *for each site*
*Done*

L358-360: Very good that you specify this here. In my experience, it is actually quite difficult to fully automate an OBIA approach…
*Thank you. Agreed!*

L375 topo-climatic
*This was corrected*

**Results**
L379: I think here it is still missing that pixels which satisfy 2 different thresholds are categorized as 'unclassified'
*This is true, we had described this in detail in the answer to reviewer but not included changes to the revised text. Rather than putting this in the results, we think it fits better in the methods in section 2.5.2 so we added it there: "The thresholds varied by class, because any pixel contains a mixture of materials in various proportions (section 3.1). Pixels which satisfy 2 different thresholds are categorized as 'unclassified'."*

L444: Add name of the lakes you refer to in corresponding figure
**done**

L475: patterns
*"n" was added*

L484: I would rather suspect (from experience of fieldwork in the region in September/October) that this is due to early snowfalls rather than late ablation season
*This is a very good possibility. Since we do not know the exact case we have re-written as: "perhaps due to early snowfalls common in this area at this time of the year."*

L487-490: this belongs to the discussion
*Shortened and moved to section 4.3*

L495-498: discussion
*Moved to section 4.5*

L509-512: These are interesting results in their own way – would actually be interesting to see. You could consider adding a supplementary figure for this?
*We did have a figure showing these results in the previous version of the manuscript. This was removed as per recommendation of both reviewers, and was replaced with the current Fig 11. As such, we do not consider adding this back in.*

L512-515: discussion

*We moved it to 4.1, it fits well there (also merged with the comment below)*
*"At the mountain range scale, the distribution of supraglacial features may be governed by more complex factors which include geomorphologic, glaciologic and climatic patterns. The topo-climatic conditions for the occurrence of supraglacial ponds on the surface of debris-covered glaciers have been addressed in a small number of studies (e.g. Sakai, 2012; Sakai and Fujita, 2010). While we could hypothesize that both ponds and vegetation tend to develop on stagnant, low angle (< 2°) areas of the debris-covered tongues (Sakai and Fujita, 2010; Reynolds, 2000; Quincey et al., 2007) and at lower elevations, which would favour increased temperature and therefore increase surface melt, we found that trends were not statistically significant on a glacier-by-glacier basis."*

L519, 525: Show this exponential decay in the corresponding figure along with its coefficients
done

L520-521: Discussion
***Re-phrased and moved to the paragraph above in section 4.1 (see text pertaining to comment on l 512 – 515)***

L523-524: 'slope gradient' is wrong here, it should be slope as the gradient of the slope is actually the second derivative of the topography. There are several occurrences of this in the text, make sure to correct all of them.
***Corrected***

L530: specify '*glacier* aspect'. Remove 'slope'.
***Done***

L532-536: this is mostly discussion
*Moved to section 4.2 **Spatial and spectral limitations of the Landsat data***

**Discussion** L539: A lot of this paragraph, as well as figure 13 should appear in the results. This will need to be reformulated.

***We have added a new section to the results, 3.7 Supraglacial pond and vegetation distribution over the large domain where we present only trends in the pond and vegetation distribution over the mountain range (Fig 13 a,b) and introduce Fig 13. The introduction of the discussion section was re-worked to only discuss the controls. This should read better.***

L554: I do not understand this as eastern Tibet is actually in the south of the range, especially when considering a 1x1° grid
***We removed this because it was mentioned several times in the uncertainties***

L594-595: These trends appear to be very small and I am wondering if they are really relevant…
***We agree, but we prefer to present them and we have added that:***
***"However, we note that trends in glacier velocities noted here are very small and may not be conclusive".***

L603-604: No need for such a justification, suggest removing this whole sentence.
***Done***

L615-620: This makes sense but am not sure it is appropriate here. It reads more like a 'response to reviewers'. Suggest removing to stay concise.
***Remove****d*

L630: 'Landsat' repeated.
***Repetition removed***

L639: '*square* meters'
***Added***

L663: 'we are aware …' comes back several times in your discussion and I personally do not like it. I would suggest remaining objective and removing it.
***We have removed the two instances in which this appeared***

L668: Here you mention that your approach can help track the changes in lake turbidity. I would also insist on the fact that it can help track the changes in pond area, which is also a great outcome.
***Moved to the end of that paragraph and rephrased to read: "Since lake turbidity is temporally highly variable and since our current dataset is a snapshot of pond density, it cannot be used to infer any variability in sediment concentration, but it provided the basis for tracking changes in glacier area, which has further applications.***

L696: '*square* meters'
***Done***

L696: some of these ice sails are actually probably large enough to encompass several Landsat pixels and could explain some of the bare ice patches.
***We cannot prove this, but we now express it as a possibility only***

L728: Suggest simplifying title to 'Wider applicability of the method'.
***Agreed and have simplified it***

L729: 'demonstrated'
***Changed***

L736: can -> could
***Changed***

L736: remove 'we acknowledge that'
***Removed***

**Tables**
Table1: add 'Band 1' for the Landsat 8 OLI bands

*We are not sure what is meant here "add Band1"*
Each Landsat band is preceded by Band 1, 2 etc

**Figures**

Figure 1: What are the plain lines? Add reference for these regional/climatic outlines
**It was a bug in Arc in the way - this is now fixed. Reference was added, as well as labels for the regions**

Figure 2: Turquoise over red is difficult to see and could be a problem for color-blind readers. Nice that you shoed the ground truth points. I was wondering if there would be a way to show the different classes? It would be interesting to have a zoomed-in view of one of the glaciers to show these different classes.

*Thanks for pointing this out. We have changed the colours to green and yellow.*
*With regards to the classes: Since the classes are defined as pure SINGLE pixels, it would not be an interesting figure, i.e. just showing some coloured pixels zoomed in. Therefore we feel it would clutter Fig 2 without adding any important information.*

Figure 6: It is hard to distinguish the LMM ponds from the OBIA ponds. Suggest taking different colors

*We have tried all the possible colours and completely re-done the figure, and unfortunately could not find a better way to display unless we zoom in, which is not possible while retaining the overview of the spatial extent. In this case, this seems to be the best we can do, and the figure has much improved from the last version. The LMM and the OBIA match pretty well, and this is the key point here. Left unchanged.*

Figure 12: This figure could be very much improved. It is currently very raw and simplistic. There is space to add additional information. For example, I suggest showing the exponential decay you mention by fitting and exponential to your points. Since we are talking of bins, wouldn't horizontal lines or bars make more sense than points? For c), specify in figure that this is glacier aspect.

*It is already specified in the figure caption "aspect over glaciers". We changed to "glacier aspect". We also added the functions (exponential decay for  (a) and polynomial for (b).*

Figure 13: It is difficult to see any correlation with such plots. It could be worth increasing the size of the dots in a first step. However, if you really want to show a correlation, it would be interesting to do so in an x-y plot.

*We have increased the size of the dots. In Figure 13 we aim at summarizing spatial trends, and this is clear in the text and caption. We did not present it as a corelation analysis but rather as support for the discussion of the controls. At the suggestion of both reviewers, we have added a correlation matrix.*

**Reviewer 2 General Comments**

The revised manuscript by Racoviteanu et al. on spectral unmixing of debris-covered glaciers has been considerably reworked and has been much improved with respect to the initial submission. I compliment the authors on their effort.

Comments provided by reviewer #1 and by me have been largely responded to satisfactorily and the manuscript was adapted where necessary. There is better validation now; a weak analysis has been replaced with more convincing approach; the discussion about the advantages and, importantly, the limitations of the approach has become much clearer; and many of the more subjective statements and interpretations were removed.

There a still a few points that I think should be resolved before publication of the manuscript.

**Thank you for these comments, and for reviewing the paper once again and providing very useful comments. We have addressed these line-by-line comments as well as the bigger concerns.**

L96. I would not say it is freely available, as that is not entirely true.
**Planet imagery for free (of course within the area limitation) for academic purposes with an API. We do not wish to go into detail here. We have removed "freely" but this is not correct either.**

L196. Fix the underscore in the reference.
**This will be done at the very final version of the paper when the references will be converted to plain text as it is related to a glitch in EndNote which we use here to manage the references**

L237. Fix the underscore in the reference.
**Same as above**

L257. LMM is here defined as plural form ('models'). In other places in the manuscript, it is also used as singular. Be consistent. LMMs is preferable when used a plural.
**We added "s" where appropriate.**

L276-277. Consist in -> consist of
**done**

L352. Cannot really put my finger on it, but "the Langtang" and "the lahaul spiti" sounds very odd. What about using Khumbu Region, Langtang Region, Lahaul Spiti Region throughout?
**Added "region" to the for instances where this was found.**

L349. Is HarrisGeospatial really written without a space?

**See comment above with regards to l 196. This is fixed when references will be converted to plan text.**

L367. "Numerous" is quite understating. "Majority" or even "vast majority"

**We have changed to "the vast majority"**

L475. Patters -> patterns

**Fixed**

L540-L592.

I think the approach taken here is very much an improvement over the previous analysis, which I think was not very convincing. However, I do not really understand why it was chosen to do the analysis on a 1x1 degree grid, as this makes the presented analysis still somewhat shallow. The authors state it is to have optimal comparison with other papers (Brun, Shean, Dehecq etc.), but that comparison is not really made in the manuscript. They also indicate that more detailed analyses are necessary to robustly analyze the drivers of debris-covered glacier surface properties. However, since all input data have much finer resolution or are even available at the glacier level, the authors could have presented a more robust and detailed analysis using their data. A more sophisticated approach that looks at individual glaciers seems very much feasible and not much more elaborate than what is currently presented, and could quite easily and logically be combined with the results presented in Figure 11.

**In the previous version of the manuscript, we presented a glacier-by-glacier analysis and both reviewers found the results not conclusive because of the glacier-by-glacier variability. Accordingly, we binned our data by (a) elevation and (b) 1x1 grid, with the aim to show the principle large-scale patterns of ponds and vegetation distribution rather than to try to derive causal relationships, as we feel this is too much to cover along with the method description within this paper. The 1x1 grid approach is standard to many HMA-region publications, and our paper fits within this approach to summarize spatial trends. We have added more comparisons with Dehecq, Brun etc.**

**We still contend that given the complex, time-dependent, covariance of debris-covered glacier surface properties, a deeper analysis seeking causal relationships and drivers of these large-scale patterns should be performed only on a further quality controlled dataset and in conjunction with additional datasets, simulated glacier states and, while undoubtedly interesting, this lies outwith the scope of this paper.**

The authors present trends of the different variables with latitude and longitude to evaluate regional patterns. By discussing it in light of these spatial patterns, they infer relations between the variables in a rather unquantified manner. It would be much stronger to directly compare the different variables and quantify whether they actually correlate. This would mean providing the correlations and whether these are actually statistically significant. Clearest would be to include an additional panel to Figure 13 with a correlation or trend matrix for all the presented

variables in Figure 13 (+ latitude + longitude). Also, for the now-included trends with latitude and longitude, it should at least be mentioned whether these trends are significant or not.

**We have added a correlation matrix as a separate table (Table 7).**

L600 stepper -> steeper
**Corrected**

L667 ponds -> pond
**Corrected**

L783 Python-based -> scripted
**Changed**